# D-LMBmap: a fully automated deep-learning pipeline for whole-brain profiling of neural circuitry

Zhongyu Li[1,4], Zengyi Shang[2,4], Jingyi Liu[2,4], Haotian Zhen[2,4], Entao Zhu[2,4], Shilin Zhong [3], Robyn N. Sturgess[1], Yitian Zhou[2], Xuemeng Hu[2], Xingyue Zhao[2], Yi Wu[2], Peiqi Li[2], Rui Lin [3] & Jing Ren [1] ✉

Recent proliferation and integration of tissue-clearing methods and light-sheet fluorescence microscopy has created new opportunities to achieve mesoscale three-dimensional whole-brain connectivity mapping with exceptionally high throughput. With the rapid generation of large, high-quality imaging datasets, downstream analysis is becoming the major technical bottleneck for mesoscale connectomics. Current computational solutions are labor intensive with limited applications because of the exhaustive manual annotation and heavily customized training. Meanwhile, whole-brain data analysis always requires combining multiple packages and secondary development by users. To address these challenges, we developed D-LMBmap, an end-to-end package providing an integrated workflow containing three modules based on deep-learning algorithms for whole-brain connectivity mapping: axon segmentation, brain region segmentation and whole-brain registration. D-LMBmap does not require manual annotation for axon segmentation and achieves quantitative analysis of whole-brain projectome in a single workflow with superior accuracy for multiple cell types in all of the modalities tested.

Comprehensive descriptions of neuronal connectivity are fundamental for understanding the brain's functional organization. Recent proliferation and integration of tissue-clearing methods and light-sheet fluorescence microscopy (LSFM) present unparalleled opportunities to achieve high-throughput mesoscale three-dimensional (3D) whole-brain connectivity mapping[1–4]. The increasing deluge of large imaging datasets urgently calls for analysis tools to quantitatively profile axonal projections with canonical coordinates at the whole-brain level with minimal manual labor. This necessitates an integrated workflow for automated processing of 3D axon recognition and segmentation, as well as whole-brain registration with standard brain atlases.

Recently, deep neural networks (DNNs)[5] have been widely investigated for biomedical imaging analysis. Machine-learning algorithms learn patterns from annotated images to recognize, classify and segment regions of interest such as cells, axons and brain regions. For example, by training manually annotated axons in a set of 3D cubes, TrailMap[6,7] achieves mesoscale axonal segmentation using 3D U-Net[8]. BIRDS[9] employs DeepLab V3+ (ref. 10) and SeBRe[11] applies Mask R-CNN[12] to achieve the detection and segmentation of major brain regions in single-viewed two-dimensional (2D) slices, whereas mBrainAligner[13] successfully segments major brain regions by applying 3D U-Net to downsampled whole brains.

[1]Division of Neurobiology, MRC Laboratory of Molecular Biology, Cambridge, UK. [2]School of Software Engineering, Xi'an Jiaotong University, Xi'an, China. [3]National Institute of Biological Sciences (NIBS), Beijing, China. [4]These authors contributed equally: Zhongyu Li, Zengyi Shang, Jingyi Liu, Haotian Zhen, Entao Zhu. ✉e-mail: jren@mrc-lmb.cam.ac.uk

**Fig. 1 | Overview of D-LMBmap. a**, The pipeline of LSFM whole-brain imaging, which includes tissue clearing, axon staining and microscope imaging. The brain can be imaged in both autofluorescence (488 nm) and stained-specific (647 nm) channels. **b**, Whole-brain axon segmentation. An axon segmentation DNN is trained to segment axons in thousands of 3D cubes from the LSFM brain and then the whole-brain axons can be reconstructed by a combination of segmented cubes. **c**, Brain-style transfer. For the LSFM brain imaged in the autofluorescence channel (left), a style-transfer DNN is trained to learn the image style of a brain atlas, where each LSFM image slice can be transferred in atlas style (for example,

Allen atlas), as well as preserving their original structures (right). **d**, Brain region segmentation. Major brain regions can be automatically segmented by a DNN, using either the original or style-transferred brains. **e**, Whole-brain 3D registration and projection mapping. The original LSFM brain, major brain regions and style-transferred brain are set as the input of the DNN for the whole-brain registration with the brain atlas. After that the registered brain is combined with the whole-brain axons to achieve the projection mapping and axon quantification. **f**, The software interface of D-LMBmap for whole-brain projection mapping and visualization.

Despite this, critical challenges remain in whole-brain connectivity profiling. First, current solutions for training segmentation DNNs are based on exhaustive manual annotations and can only perform well in predicting samples that are similar to the training data. Training models using manually traced annotations for complex and extensive axons in large-scale 3D whole brains or delineating multiple brain regions in thousands of slices for each sample from different batches/modalities, is very inefficient. Second, current 3D brain registration methods relying on whole-brain intensity (for example Clearmap[14,15] and aMAP[16]) or specified brain regions and landmarks (for example mBrainAligner[13]) fail to coordinate multiregional alignment optimization and whole-brain registration. Consequently, brain regions that are vulnerable to damage during sample preparation often experience misalignment or inadequate results. Deformation during registration of large brain regions can result in decreased accuracy for smaller structures inside them. Third, whole-brain data analysis requires combining multiple software packages and needs secondary development by users, such as data annotation, deep model training, whole-brain axon prediction and 3D whole-brain registration, which is tedious and technically difficult, resulting in these pipelines being rarely revisited by the community.

In this study, we develop D-LMBmap (deep-learning pipeline for mouse brain mesoscale automatic profiling) to address all the above challenges. It packages an integrated workflow containing three modules based on new deep-learning algorithms for whole-brain circuitry profiling (axon segmentation, brain region segmentation and whole-brain registration). We achieve robust whole-brain axon segmentation by building an improved nnU-Net-based deep-segmentation model and automated generation of large-scale high-quality training data. To quantify axon densities in each brain region, we develop a cross-modality 3D whole-brain registration method through a style-transfer solution and a multi-constraint strategy. Major brain regions are segmented automatically by a multiview semi-supervised network and a multiple constraint unsupervised VoxelMorph-based network[17] is designed to achieve whole-brain registration that considers the alignment of style-transferred source brain and the segmented brain structures. Our pipelines require minimal manual input and are extensible to diverse image modalities, either at the axon or whole-brain level. D-LMBmap outperforms existing methods in all three modules in accuracy, speed, generalization and ease of use.

## Results

### A complete pipeline for mesoscale whole-brain analysis

D-LMBmap consists of interconnected modules that facilitate a workflow starting from data input and resulting in axonal projection quantification and visualization in the Allen mouse brain atlas (CCFv3) (ref. 18). The main modules of axon segmentation, brain region segmentation and whole-brain registration utilize advanced DNNs. D-LMBmap is designed for whole mouse brain axonal projections labeled by anterograde tracing, but the axon segmentation module can be substituted with a soma detection module for retrograde tracing or brain activity mapping. Each module can be used independently or combined for cross-modality analysis.

Here, LSFM datasets serve as examples to illustrate the workflow and strategies in each module (Fig. 1). Tissue-cleared mouse brains with labeled axonal projections are imaged using two fluorescent channels: axon labeling (specific stain) and autofluorescence (Fig. 1a). Axon segmentation employs images from the axon-labeling channel (Fig. 1b), whereas brain region segmentation uses images from the autofluorescence channel (Fig. 1c,d). To overcome challenges in brain region segmentation and registration, we built a neural network submodule, 'brain-style transfer', which renders brain images in Allen atlas style (Fig. 1c). The whole-brain registration module integrates the outputs of both channels, generating quantified whole-brain projection intensity maps (Fig. 1e). D-LMBmap offers a user-friendly graphical user interface for comprehensive usage (Fig. 1f) and trained models are equipped for each module. It is also an open-source software with high-level application programming interfaces for customization. The graphical user interface enables selection and computation using different pretrained deep models.

### Automated axon segmentation

Training DNNs generally requires abundant data with accurate annotations. Low quality or quantity of training data can impede the recognition of axons and artifacts; however, manually annotating 'foreground axons' and 'background noise' is extremely labor intensive. The morphology, density and contrast of both vary greatly across different brain regions and the image variability is particularly pronounced across different experimental batches. Manually labeling complex and dense axons in 3D space, particularly at the whole-brain level, poses great challenges. To tackle this, we develop automated axon segmentation for whole-brain projection mapping. Large-scale and diverse annotated datasets for training are generated by automatically annotating 3D image cubes and creating derivative artificial cubes by data augmentation.

First, D-LMBmap asks users to select small 3D cubes (150 × 150 × 150 voxels) containing predominantly artifacts or axons from the whole brain (Fig. 2a). With just one click, users can select the desired region and D-LMBmap automatically extracts the 3D cube of the predefined size. This manual selection of 3D cubes is the only step that relies on user input and after that D-LMBmap automatically annotates the selected 'artifact' and 'axon' cubes. 'Artifact' cubes represent areas with no axons, visually represented as black within the algorithm (Supplementary Fig. 1a). For voxel-level annotation of the 'axon' cubes, we employ a binarization and skeletonization workflow (Fig. 2b and Supplementary Fig. 1b). D-LMBmap first binarizes the cube and extracts the axons with a set of image-processing techniques, including Gaussian filtering, Gaussian difference and thresholding. D-LMBmap then connects the adjacent fragments of the binarized axons through dilation. Axon center lines are extracted for skeletonization and dilation ensures unified thickness in the annotation, resulting in automatic 3D annotation of axons. Unlike manual annotation in individual 2D slices[6], which leads to repetitive annotations in adjacent slices, D-LMBmap avoids redundancy in the z-stack annotation.

The annotated 3D training cubes generated at this step only represent simple examples of axons or artifacts and they cannot fully capture the intricate reality of complex axons and backgrounds. To enhance the diversity, complexity and quantity of the training pool without additional effort, we develop multiple data augmentation strategies (Fig. 2c). Alongside common augmentation techniques in image segmentation such as random rotation, scaling, Gaussian

**Fig. 2 | Automated axon annotation and whole-brain axon segmentation.** **a**, A 3D whole brain and the selected 3D cubes containing sparse axons, dense axons and artifacts, respectively. **b**, The automated annotation workflow for 3D cubes with 'pure' axons. It contains two steps, adaptive binarization (second row) and axon skeletonization (third row). **c**, The introduced data augmentation strategy to improve the diversity of annotated cubes, including CutMix, histogram matching and local contrast augmentation. **d**, The workflow of the DNN for axon segmentation, including preprocessing for training cube packaging, training strategy self-regulation and network training. **e**, The comparison of axon segmentation results of example cubes containing sparse axons, dense axons and mixed artifacts and axons between TrailMap and D-LMBmap. The blue squares indicate the regions for zoom-in comparison. (Scale bar, x, y, z = 60 μm). **f**, Quantitative comparison between D-LMBmap and TrailMap under the evaluation of Dice, ClDice, Precision and ClPrecision (two-tailed paired t-test, n = 10. ClDice, P = 0.0003, t = 5.622, d.f. = 9; ClPrecision, P = 0.0005, t = 5.223, d.f. = 9; ClRecall, P = 0.0622, t = 2.129, d.f. = 9; Dice, P = 0.000095, t = 6.641, d.f. = 9). Measure of center, mean; error bars, mean ± s.d.

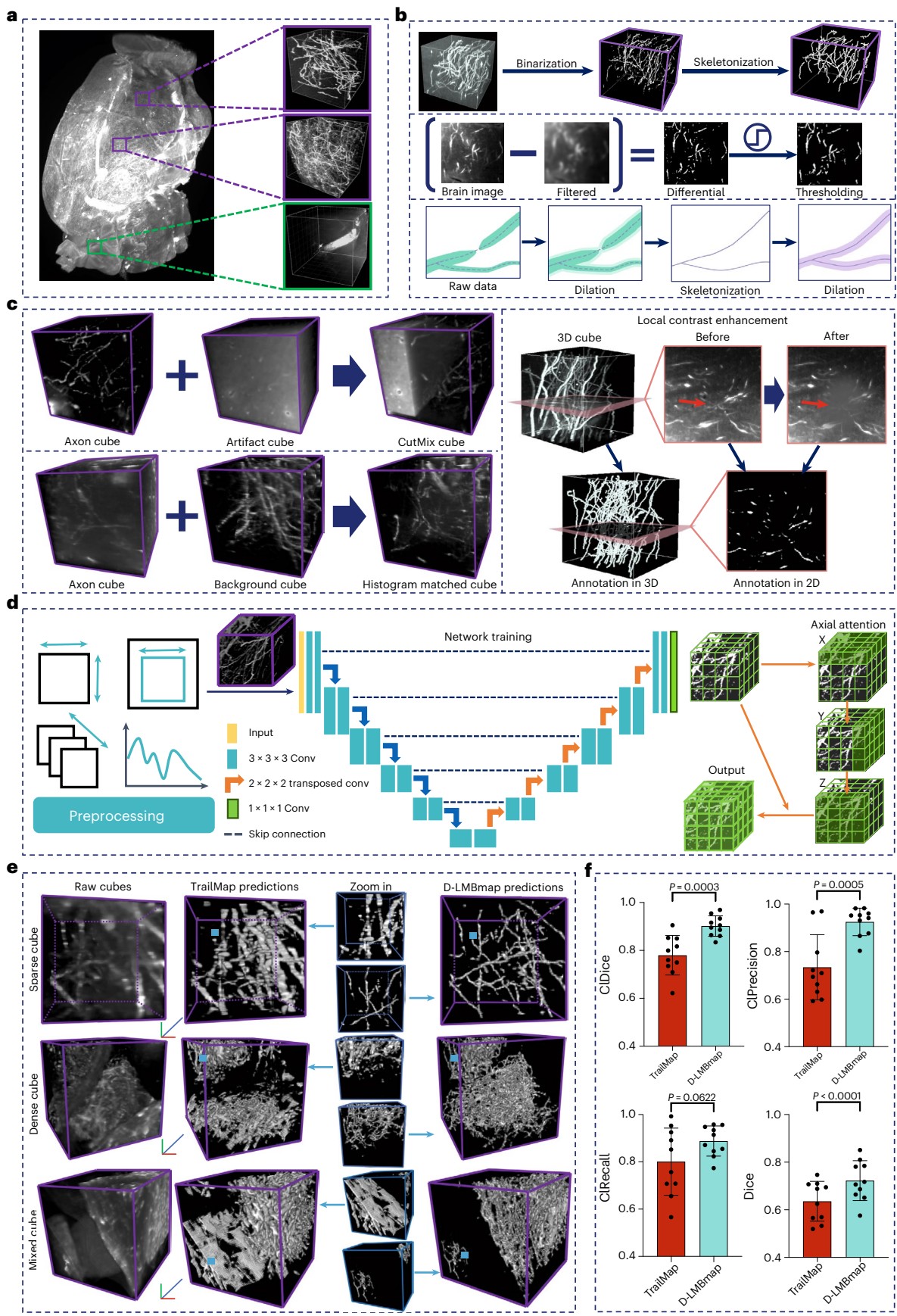

noise, blur and brightness adjustments[19], we introduce three additional augmentations to simulate scenarios found in diverse stained brain datasets. First, CutMix[20] randomly combines segments from 'axon' and 'artifact' cubes to create new cubes. Second, we employ histogram matching[21] to transfer backgrounds between cubes while preserving annotated axons. Third, D-LMBmap randomly adjusts the intensity of selected stochastic axons to augment morphologies and connectivity patterns[22]. Both the automatically annotated cubes and data augmented cubes serve as training data for segmentation DNNs, enabling the recognition of axons and artifacts across a broad range of diverse 3D cubes.

Finally, we implement nnU-Net[23], a self-configuring DNN, for axon and artifact prediction (Fig. 2d). Compared to other commonly used network architectures such as 3D U-Net[8], V-Net[24] and DenseVoxel-Net[25], nnU-Net[23] offers more-advanced 3D semantic segmentation. It dynamically configures training parameters based on dataset properties during preprocessing, including resampling, normalization and batch size. To further improve its performance, we introduce the axial attention[26] in the decoder to our nnU-Net-based network[23], preserving the tree-topological structure of axons along each axis ($x$, $y$, $z$). After training the segmentation model, axons within each 3D cube are automatically segmented and reconstructed. To analyze the whole-brain axons, a single mouse brain imaged by LSFM is divided into over 4,000 individual cubes, which are later combined.

We assessed the performance of D-LMBmap in segmenting whole-brain axons on multiple brain samples generated in different laboratories and containing various types of axons, including serotonergic[27], GABAergic, glutamatergic[28] and dopaminergic (Supplementary Table 1). In comparison with TrailMap[6], D-LMBmap demonstrated higher accuracy in recognizing axons and artifacts under all conditions (Fig. 2e). When segmented axons are visualized in 3D, D-LMBmap naturally reflects the actual distribution and thickness of axons (Extended Data Fig. 1 and Supplementary Video 1). There is no redundancy in the $z$ axis, which is caused by repetitive annotation in adjacent 2D slices (Supplementary Video 2). In quantitative evaluations across ten cubes containing various types of axons (Supplementary Fig. 1c), D-LMBmap consistently outperformed in Dice, ClDice[29,30] and ClPrecision scores (Fig. 2f and Extended Data Fig. 2a,b), irrespective of cube location (Extended Data Fig. 2c,d) or axon density (Extended Data Fig. 3a,b).

Ablation study results support the effectiveness of our data augmentation and axial attention strategies for axon segmentation (Supplementary Fig. 2). In summary, our axon segmentation workflow is effective and applicable to different neuron types at the whole-brain level (Extended Data Fig. 4 and Supplementary Video 3).

### Brain-style transfer implementation

Following whole-brain axon segmentation, the critical next step for whole-brain connectivity mapping is registering the experimental brain to a standard brain atlas (for example, the Allen atlas[18]).

Robust and accurate whole-brain registration should ensure precise alignment of the brain intensity and internal regions. To support region-wise whole-brain registration, we develop a brain region segmentation method for the automated delineation of different brain regions. Due to the differences in sample preparation, imaging settings, etc., brain samples from different batches/laboratories may display varying textures, colors and levels of distortion. Cross-modality registration faces even greater variability and challenges in developing a unified model of deep segmentation. In addition, direct 3D segmentation of a whole mouse brain is currently impractical due to computational limitations and manual annotation of multiple 3D brains is labor intensive; however, converting a 3D brain to 2D slices often results in information loss and accurately identifying boundaries of brain regions in single coronal, horizontal or sagittal views is difficult[31]. To overcome these challenges, we implement brain-style transfer and multiview semi-supervised segmentation modules in D-LMBmap (Fig. 3).

To mitigate the variability of brain samples across different batches and modalities, we developed a style-transfer solution inspired by artistic image techniques[32]. It converts the appearance of 2D brain slices into a reference atlas style, while preserving their original content. D-LMBmap achieves the style transfer based on an unpaired image-to-image translation DNN, CycleGAN[33] (Extended Data Fig. 5a). To maintain consistency in brain shapes, we further developed a deep-segmentation backbone, CEA-Net, to automatically segment brain outlines for each input brain image (Supplementary Fig. 3). Here, we used an LSFM brain sample and the Allen atlas[18] as example inputs (Fig. 3a). The network learns Generator A and B to transfer the style between the LSFM brain and Allen atlas, as well as learning Discriminator A and B to differentiate the original and synthetic brains. CEA-Net introduces brain outline constraints to keep consistency between the input LSFM brain sample and the output synthetic 'Allen-style' brain. The resulting style-transferred brain retains the shape and boundaries of the input LSFM brain at the voxel level but adopts the texture, color and appearance of the Allen atlas (Fig. 3b and Extended Data Fig. 5b). Additionally, other atlases such as the LSFM brain atlas[34] can also be used as references (Extended Data Fig. 5c).

The unsupervised and fully automated style-transfer strategy in D-LMBmap is robust and works with experimental brains and atlases imaged at different resolutions and orientations. Furthermore, it does not require slice-to-slice anatomical correspondence between the sample and the reference images, eliminating the time-consuming manual pairing of the input images. This strategy will also be employed in the subsequent whole-brain registration module.

### Multiview semi-supervised brain region segmentation

D-LMBmap employs a new deep model for accurate and robust brain region segmentation in a multiview and semi-supervised framework (Fig. 3c). We developed Semi-CEA as the semi-supervised brain image segmentation backbone (Extended Data Fig. 6a). The deep model is

**Fig. 3 | Brain-style transfer and automated brain region segmentation.**
**a**, The input consists of LSFM brain slices, while the reference images are from the Allen atlas. The DNN based on CycleGAN consists of two generators and two discriminators. Generator A generates Allen-style images from LSFM-style images, whereas Generator B generates LSFM-style images from Allen-style images. Discriminator A discriminates between original and synthetic LSFM images and Discriminator B does the same for Allen images. The final output is the synthetic Allen images, which are converted to the Allen style with their original content. The brain outlines segmented by CEA-Net are employed as constraints during CycleGAN training. **b**, Example results showing the input LSFM brain sample has been successfully transferred into a synthetic 'Allen-style' brain. **c**, The network architecture for semi-supervised multiview brain region segmentation. The network is trained by one atlas/annotated brain and several unannotated brains. The 2D brain slices from two views, coronal and horizontal, are extracted for training in Semi-CEA. After training, brain region predictions in coronal and

horizontal views are transformed into a unified view and then combined for the computation of consistency loss (MSE loss). The semi-loss and the multiview loss are integrated for the whole deep model training. **d**, Examples of brain region segmentation results in coronal and horizontal views. Six main brain regions are color coded (CTX, green; CP, blue; HPF, light green; BS, pink; CB, orange; and CBX, yellow). The white lines in the amplified images mark the edge of segmented brain regions. **e**, Quantitative evaluation of different brain region segmentation methods on autofluorescence channel LSFM brains using the atlas-trained pipeline ($n = 12$). The brain data and annotations used for training the multiview Semi-CEA deep model (left). Region-wise median Dice score for six brain regions (CP, HPF, CTX, CB, CBX and BS) (middle). Average median Dice score of different methods (right). **f**, Quantitative evaluation of different brain region segmentation methods on autofluorescence channel LSFM brains using the sample-trained pipeline ($n = 8$). Box plot, center line, median; box limits, upper and lower quartiles; whiskers, 1.5 × interquartile range; points, individual data points.

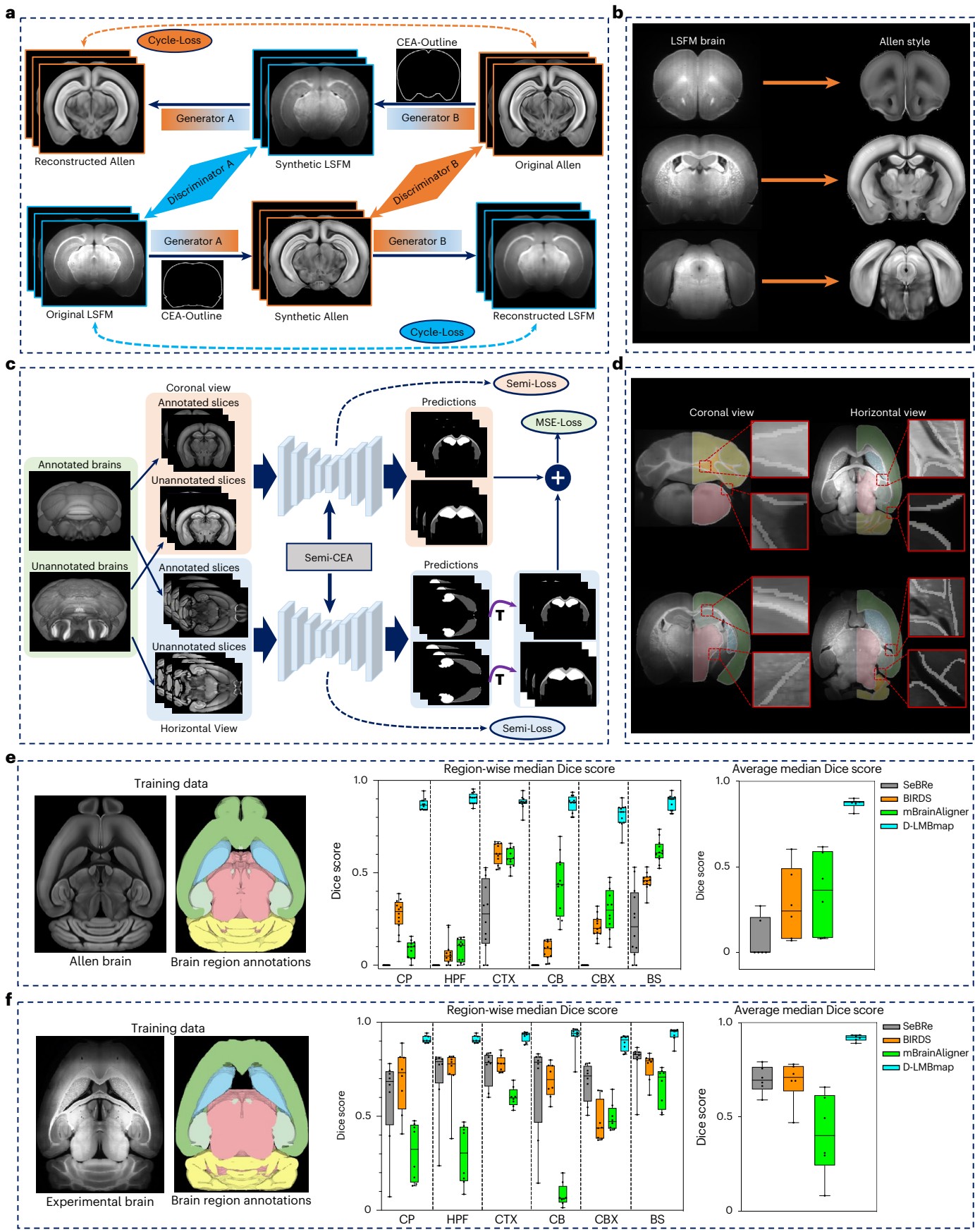

trained by one or several annotated brains and a few unannotated brains, where the method considers 2D brain slices in both coronal and horizontal orientations. The multiview strategy minimizes the information lost during 3D to 2D conversion and successfully improves the segmentation performance. The annotated brain(s) can be a brain atlas (Allen atlas[18]) or generated by the user. The training results of coronal and horizontal slices are combined to compute the loss function based on their consistency and back-propagation for the model training.

To better support the following whole-brain registration step, six major brain regions were selected as default settings for segmentation: cerebral cortex (CTX), caudoputamen (CP), hippocampal formation (HPF), brain stem (BS), cerebellum (CB) and cerebellar cortex (CBX) (Extended Data Fig. 6b). There are two pipelines available; the 'atlas-trained pipeline' uses a public atlas for training and applies style transfer to convert experimental brain samples to atlas style, whereas the 'sample-trained pipeline' involves manual annotation of any detailed brain structures in one or several experimental brains for training, providing high accuracy. Both pipelines yield the segmented brain regions in the modality of the experimental brains (Fig. 3d).

To validate the quantitative efficiency of D-LMBmap on brain region segmentation, we manually annotated 12 LSFM brains generated in different laboratories. We evaluated both pipelines, atlas-trained and sample-trained. Compared to recent mouse brain region segmentation methods (SeBRe[11], BIRDS[9] and mBrainAligner[13]), both D-LMBmap pipelines showed superior results in all six brain regions (Fig. 3e,f). When using the Allen atlas for training, the atlas-trained pipeline of D-LMBmap outperformed other methods with a 30% higher average median Dice score (Fig. 3e). When annotated brain samples were used for training, the sample-trained pipeline of D-LMBmap achieved a 10% higher average median Dice score compared to other methods (Fig. 3f and Extended Data Fig. 6c). Moreover, the sample-trained pipeline also achieved better and more stable performance than the atlas-trained pipeline, with a 5% higher average median Dice score. D-LMBmap also showcased excellent performance on low-resolution magnetic resonance imaging (MRI) brains (Supplementary Fig. 4) and LSFM brains imaged in the stained-specific channel (Supplementary Fig. 5).

The results of the ablation study suggest that our innovative brain-style transfer and multiview semi-supervised segmentation modules are highly efficient for brain region segmentation (Supplementary Fig. 6), enabling D-LMBmap achieves a Dice score of around 0.9 across various modalities and requires minimal or no manual processing.

## Multi-constraint and multiscale whole-brain registration

D-LMBmap achieves meticulous cross-modality registration of 3D whole mouse brains, addressing the challenges of distortion and damage caused by sample preparation. Artifacts bias certain brain areas and contribute to complex imaging variations. Regions near the ventricular system are susceptible to distortion, whereas areas close to the surface are vulnerable to damage during tissue preparation. Existing methods

mainly focus on intensity alignment by large-scale deformation optimization at the whole-brain level; however, the experimental source brains and the reference brain aimed for registration are often from different modalities. Meanwhile, large-scale deformation optimization cannot guarantee an accurate alignment for individual internal brain regions, especially small or easily damaged areas.

To overcome these issues, we develop a multi-constraint and multiscale DNN for whole-brain registration (Fig. 4a). Multiple constraints are introduced at different levels, including the style-transferred source brain, segmented major brain regions and selected small brain structures, including the ventricular system, which can be automatically obtained by the aforementioned modules. The style-transferred source brain has a similar appearance to the reference brain and keeps voxel-level one-to-one correspondence with the source brain. This can alleviate the intensity gap between the reference brain and the source brain. The constraints in each brain region greatly improve the optimization in comparison with the whole brain, which can further enhance the registration of local regions. Subsequently, all the inputs are downsampled twice for training and undergo transformations at each scale. We extend the VoxelMorph[17] to achieve multiscale deformable registration with multi-constraints. We train the model first on downsampled data with fewer optimization parameters and then fix the parameters to train larger-sized data. The model takes the original brain, style-transferred brain and segmented brain regions as inputs in different channels, updating their transformation parameters simultaneously. This unified framework enables the automatic computation of rigid, affine and deformable transformations from the source brain to the reference brain (Fig. 4b).

D-LMBmap outperforms state-of-the-art mouse brain registration methods, including ClearMap[14], aMAP[16] and mBrainAligner[13], across different modalities. To achieve quantitative evaluation, we computed the median Dice score of six large brain regions (CTX, CB, CBX, HPF, CB and CP) and the average median Dice score between the source brains and the Allen atlas after the whole-brain registration, for 12 LSFM whole mouse brains in the autofluorescence channel. D-LMBmap achieved individual brain region scores ranging from 0.87 to 0.95 and a region average median Dice score of 0.93, about 10% higher than other methods (Fig. 4c and Extended Data Fig. 7a). Additionally, D-LMBmap demonstrated superior whole-brain registration results for MRI brains (Extended Data Fig. 7b,d) and LSFM brains in the stained-specific channel (Extended Data Fig. 7c,e).

Our ultimate goal is to achieve accurate registration of hundreds of individual brain regions, including the tiny structures within the thalamus, hypothalamus and brainstem; however, current methods struggle with the registration accuracy of small brain regions. In contrast, the multi-constraint strategy provides more reliable anchors to guide the deformation optimization in D-LMBmap, thereby much more effective in registering small brain structures (Extended Data Fig. 8a,b). We evaluated the median Dice score for five selected small brain structures

**Fig. 4 | Whole-brain 3D registration workflow and results. a,** The multiscale learning-based framework with multi-constraints for the whole-brain 3D registration. The registration input contains multiple sources, including the LSFM experimental brain, the style-transferred experimental brain, the segmented brain regions and the reference atlas and atlas brain regions. All the inputs are downsampled twice and then computed with rigid and affine transformation using convolutional layers. A neural network based on VoxelMorph is used for deformable transformation. **b,** Registration results of representative brains from different modalities, including the LSFM brain in the autofluorescence channel, the MRI brain and the LSFM brain in the stained-specific channel. Orange lines indicate where the brain region boundaries are defined in the Allen atlas overlaid. (Scale bar, $x, y, z = 1$ mm). **c,** Quantitative evaluation and methods comparison of whole-brain registration on 12 LSFM brains in the autofluorescence channel. Median Dice score of six individual major brain regions (CP, HPF, CTX, CB, CBX and BS) (left). Average median Dice

score across six brain regions (Right). Box plot: center line, median; box limits, upper and lower quartiles; whiskers, 1.5× interquartile range; points, individual data points. **d,** Quantitative evaluation and methods comparison of whole-brain registration on small brain structures using nine LSFM brains. Median Dice score of five small brain structures (act, fr, mtt, IPN, Hb) (left). Average median Dice score across five small brain structures (right). Box plot, center line, median; box limits, upper and lower quartiles; whiskers, 1.5 × interquartile range; points, individual data points. **e,** Evaluating the whole-brain registration by the false-positive report of cell distribution in the ventricular system. (i) Representative horizontal and coronal brain slices containing the LV, 3rd V and AQ; (ii) Numbers of cells that are falsely reported to be in LV, 3rd V and AQ by ClearMap, mBrainAligner and D-LMBmap (one-way analysis of variance followed by Dunnett's multiple comparison test. $F(1.003, 2.006) = 30.44$, $P = 0.031$, $n = 3$). *$P < 0.05$. Measure of center, mean; error bars, mean ± s.d.

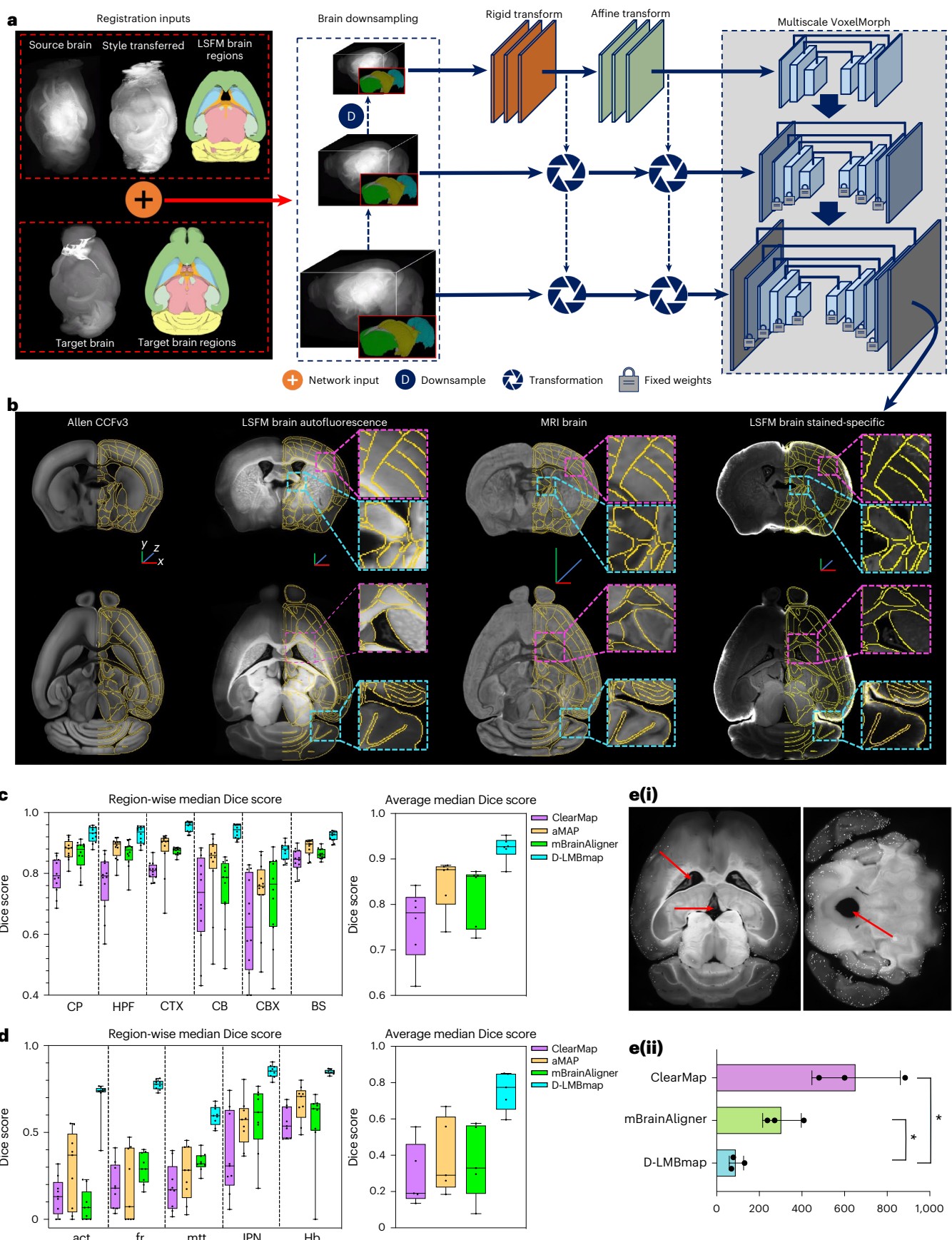

(anterior commissure temporal limb (act), fasciculus retroflexus (fr), mammillothalamic tract (mtt), habenular (Hb) and interpeduncular nucleus (IPN)), as well as the average median Dice score between the source brains and the Allen atlas after the whole-brain registration. When automated small-structure constraints are applied, D-LMBmap achieved a median Dice score from 0.60 to 0.85 for individual small brain regions and an average median Dice score of 0.76, about a 35% higher average Dice score than other methods (Fig. 4d). Users can further improve results by using constraints generated from manually annotated training samples. The superior performance of D-LMBmap in registering fine brain structures is also evident in the evaluation of landmark distance deviations (Extended Data Fig. 9d,e).

The ventricular system is prone to deformation during tissue preparation, resulting in inaccurate registration of adjacent brain regions. For example, we analyzed three mouse brains stained for neuronal activity in response to foot shock. Several activated key brain regions locate near the ventricular system, such as the habenula, paraventricular nucleus of the thalamus, periaqueductal gray and amygdala. Inaccurate registration can misassign signals from these regions to the ventricular system, leading to misinterpretations. To assess D-LMBmap's effectiveness in registering ventricular system-adjacent brain regions, we compared it to ClearMap[14] and mBrainAligner[13]. We counted the falsely detected number of neurons in the lateral ventricle (LV), the third ventricle (3rd V) and the cerebral aqueduct (AQ) after whole-brain registration. D-LMBmap reported fewer false assignments (91 cells per brain in the LV and 3rd V and 2 cells per brain in the AQ) compared to the other two methods, which wrongly assigned around 300–600 cells per brain to the ventricular system (Fig. 4e).

These results demonstrate that D-LMBmap can achieve excellent whole-brain registration by employing multiple constraints to achieve multiregional alignment optimization (Extended Data Fig. 9a,b). Constraints of the style-transferred source brain ensure consistent intensity globally and locally (Extended Data Fig. 9c). This enables registration of source brains with fewer anatomical features and more imaging noise, such as LSFM brains in the stained-specific channel and MRI brains (Fig. 4b and Extended Data Fig. 7b–e). In addition, due to the tissue-clearing processing and light-sheet imaging settings, the majority of our LSFM brains suffer from damage in the olfactory bulb, cerebellum and brain surface. Nevertheless, constraints of different brain regions assure region-wise alignment. D-LMBmap can even perform accurate registration when the brains are damaged by up to 50% (Supplementary Fig. 7).

The multiscale learning strategy reduces computational complexity and efficiently handles large-sized whole brains. The whole registration process is completed in just a few minutes, much faster than traditional optimization-based methods that take hours. For a regular LSFM source brain (320 × 456 × 528 voxels), registration to the Allen atlas takes only about 5 min on a standard laptop (Supplementary Table 5). In summary, D-LMBmap's learning-based framework facilitates effective and efficient whole-brain registration, offering a comprehensive solution.

### Mouse brain circuitry mesoscale automatic profiling

D-LMBmap integrates three modules to create a complete workflow for quantifying axonal projections throughout the whole brain. The software offers five key functions: automated axon segmentation, brain-style transfer, brain region segmentation, whole-brain registration and region-wise axon quantification. We use automated tools for labeling, training, versioning, continuous integration, packaging, distribution and documentation to enable a reliable, reproducible and easy-to-use software package.

Here, we showcase the whole-brain projection heat maps of four different neuronal types generated by D-LMBmap, demonstrating its adaptability to different neuronal types (Supplementary Table 1) and both sparse[27] (Fig. 5a) and dense labeling (Fig. 5b,c and Extended Data Fig. 10). We successfully obtained the first 3D whole-brain projectomes of the dorsal raphe nucleus serotonin neurons (Fig. 5b) and the ventral tegmental area GABAergic neurons (Fig. 5c), which are quite challenging because of the extreme density and complexity of the axons. Additionally, axonal density quantification is provided for each brain region based on the Allen atlas taxonomy (Fig. 5a(ii),b(ii),c(ii)). The software also offers batch processing, whole-brain visualization (Supplementary Videos 4–6) and result exportation functionalities (Supplementary Table 2).

## Discussion

Here, we present D-LMBmap, an end-to-end deep-learning system for mouse whole-brain circuitry profiling. It offers automated axon segmentation, enabling efficient identification and reconstruction of long-range axonal projections across the entire mouse brain within hours, without manual annotation. D-LMBmap quantifies axon densities in hundreds of brain regions using a new 3D registration method that incorporates brain-style transfer and region constraints, ensuring accurate and robust cross-modal registration in minutes. We have packaged D-LMBmap in a user-friendly workflow, making it accessible to neuroscience researchers without extensive computational backgrounds.

Mapping connectomes at single synapse resolution often relies on time-consuming electron microscopy (EM); however, generating whole-organism connectomes using EM-based nanoscale techniques has been limited to organisms such as *Caenorhabditis elegans*[35–37] due to the size and complexity of the mammalian brain[38]. Recently, notable progress has been made in mesoscale connectomics by mapping cell-type specific connections across different mouse brain regions using a combination of viral genetic labeling and block-face imaging techniques[39], such as serial two-photon tomography (STPT)[40,41], fluorescence micro-optical sectioning tomography (fMOST)[42,43], high-definition fMOST (HD-fMOST)[44,45] and volumetric imaging with synchronized on-the-fly-scan and readout (VISoR)[46]. Although algorithms for analyzing block-face imaging data have been developed[41,43,47], automated and efficient methods for tracking single axons at the whole-brain level are still in high demand. To achieve single-neuron tracing, block-face imaging strategies typically use sparse labeling and reconstruct axons from high-resolution 3D brain images of 20,000 × 30,000 × 25,000 voxels. It will be very time-consuming to exhaustively predict small sized cubes (for example, 150 × 150 × 150 voxels) one-by-one, but by limiting the regions for analysis only to the areas relevant to the sparsely labeled fibers by preprocessing may reduce computational time. In the future, we will further extend our automated axon segmentation pipeline for single-neuron tracing based on high-resolution 3D brain images.

Block-face imaging requires specialized instruments that may not be readily accessible to many researchers. Modern bulk-tracing data, generated using viral-genetic strategies and imaged with LSFM at the

---

**Fig. 5 | Whole-brain axonal architecture and regional analysis. a**, Whole-brain circuitry profiling of Sert-Stanford brains. (i) Axon segmentation results on ten horizontal brain slices overlaid and average axon distribution heatmaps in horizontal, coronal and sagittal views (*n* = 3). (ii) Axon density in hierarchical brain regions based on the Allen atlas. Scale bar, *x*, *y*, *z* = 1 mm. **b**, Whole-brain circuitry profiling of Sert-NIBS brains. (i) Axon segmentation results on ten horizontal brain slices overlaid and average axon distribution heatmaps in

horizontal, coronal and sagittal views (*n* = 3). (ii) Axon density in hierarchical brain regions based on the Allen atlas. Scale bar, *x*, *y*, *z* = 1 mm. **c**, Whole-brain circuitry profiling of GABA-NIBS brains. (i) Axon segmentation results on ten horizontal brain slices overlaid and average axon distribution heatmaps in horizontal, coronal and sagittal views (*n* = 3). (ii) Axon density in hierarchical brain regions based on the Allen atlas. Scale bar, *x*, y, *z* = 1 mm.

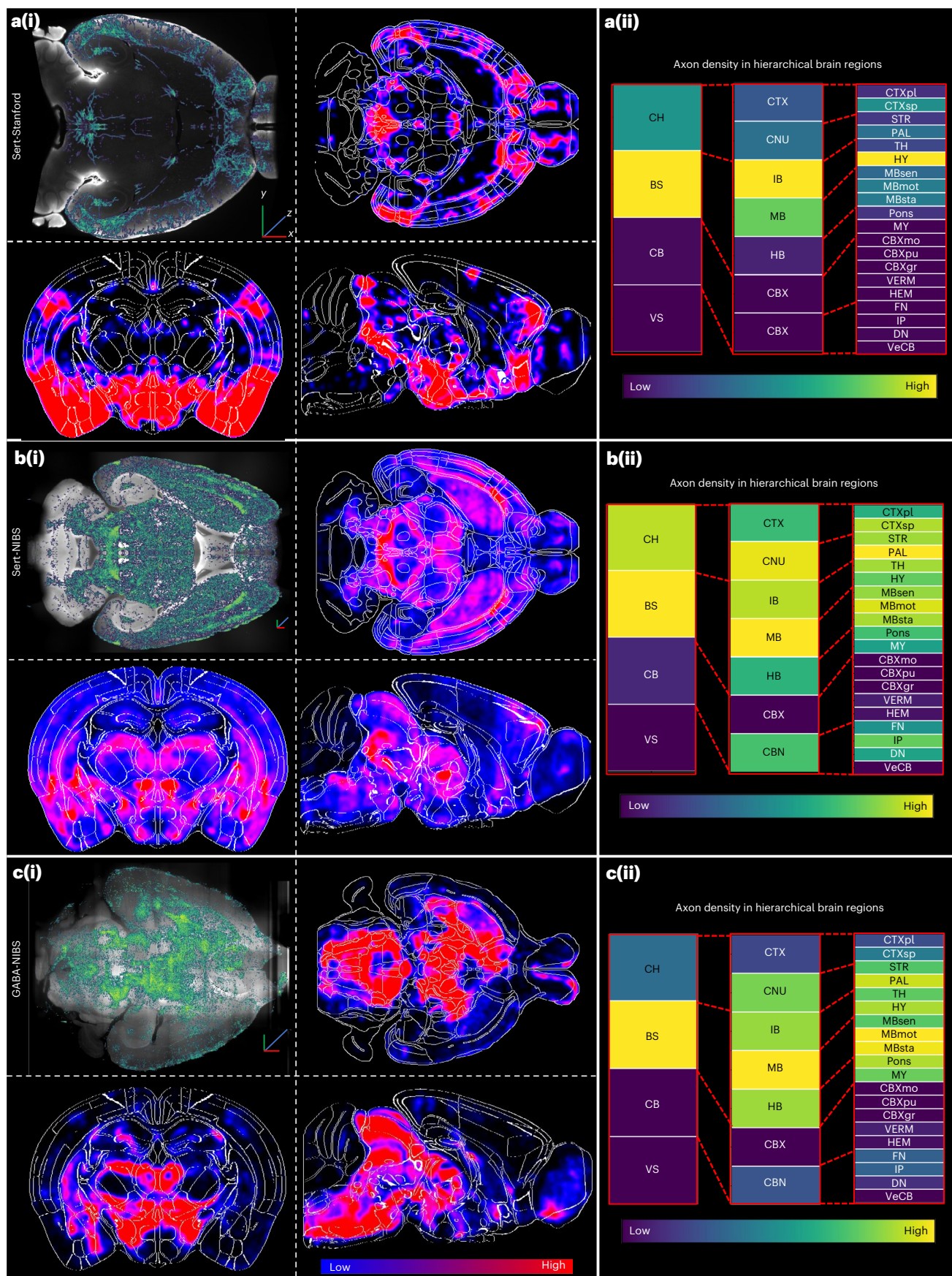

whole-brain level, can reveal the connection relationships of highly specific neuronal types defined by multiple molecular features, as well as other important anatomical information[3,27,48]. Despite the limited resolution of LSFM brains for single-cell level axon tracking, this method enables high-throughput mapping from whole-brain images of 2,000 × 2,500 × 2,000 voxels. The widespread adoption of LSFM facilitated high-throughput mesoscale 3D whole-brain connectivity mapping necessitates software development for data analysis.

Deep learning has gained prominence in image analysis due to their remarkable performance[13,49–55]; however, their application to 3D circuitry profiling of brain samples is complicated by the diverse image variance and complexity of the brain tissue. Deep models rely on accurate and extensive training data and producing comprehensive manual annotation poses a bottleneck in developing robust whole-brain projection mapping algorithms[6,13,52]. D-LMBmap achieves superior accuracy but greatly alleviates the labor-intensive manual annotation. For axon segmentation of LSFM samples, it leverages automated annotation and 3D cube augmentation, eliminating the time-consuming process of manual labeling. Additionally, D-LMBmap utilizes brain-style transfer and semi-supervised learning techniques to enhance whole-brain registration accuracy with minimal manual input. The automated pipeline achieves excellent brain region segmentation with a Dice score exceeding 90%, requiring either no manual input or only one annotated brain per sample batch. For greater accuracy in specific brain structures or when experimental brains differ greatly from the atlas, users can train the registration model with manually segmented regions of interest and perform automated testing (Extended Data Fig. 8b), eliminating the need for manual delineation of regions of interest for each experimental brain[56].

Second, D-LMBmap is easily applicable to various brain samples, providing effective axon segmentation regardless of sample backgrounds and morphological diversity (Extended Data Figs. 2–4 and Supplementary Fig. 1). We have successfully tested D-LMBmap on different axonal projection types and LSFM brain samples from various laboratories, achieving consistent results, including serotonergic, GABAergic, glutamatergic[28] and dopaminergic neurons (Supplementary Table 1). Furthermore, our brain transfer strategy allows D-LMBmap to facilitate cross-modality image registration. This is particularly useful for samples with limited anatomical features or low-resolution boundaries, such as LSFM brains that are imaged in stained-specific channels or those collected via MRI (Fig. 4b, Extended Data Figs. 6 and 7 and Supplementary Figs. 4 and 5).

Third, D-LMBmap offers user-friendly software equipped with well-trained deep models for axon segmentation, brain-style transfer, brain region segmentation and whole-brain registration. This comprehensive toolkit allows neuroscientists to effortlessly conduct whole-brain circuitry mapping. Our straightforward installation process, along with a tutorial video and example data available at https://github.com/lmbneuron/D-LMBmap, ensures accessibility for researchers with varying levels of computational experience. To our knowledge, D-LMBmap is the first software package to provide an end-to-end solution for whole-brain circuitry profiling.

Finally, D-LMBmap is an open-source software with a flexible modular design. For example, whole-brain registration using D-LMBmap excels in algorithmic efficiency and it is cost effective for wet laboratories to train the models using 25-μm resolution images on a regular server (Supplementary Table 5); however, training models for higher resolution using 10-μm images directly is impractical. It can be resolved by extending the pipeline with a module that registers major brain regions instead of the entire brain at 10-μm resolution. This approach combines whole-brain registration at lower resolutions and registration of major brain regions at higher resolution (Supplementary Fig. 8). Although the training time increases for refining high-resolution registration of each major brain region, the overall training time remains acceptable for most wet labs. Most notably, each

deep model in D-LMBmap can be extended to accommodate various signals (for example, axons, somata and nuclei), imaging modalities and animal models.

While this study primarily focuses on the development and validation using LSFM mouse brains, D-LMBmap offers the potential for broader applications. Even though our automated strategy greatly reduces manual input in axon segmentation, developing training modules for specific axon types or sample batches remains time-consuming. Next, we will focus on transfer learning, including domain adaptation and model generalization techniques, to create more generalized deep models that can be effectively applied to various axon types, minimizing the need for specialized training. Meanwhile, our current brain-style transfer algorithm has limitations in preserving fine-grained local structures, due to its unguided, unpaired and unsupervised nature. To address this, we will explore a diffusion model with multi-constraint embedding for fine-grained brain-style transfer, aiming to develop a unified cross-modal brain registration solution.

## Online content

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

## Methods

### Animal care and use

All experiments related to the use of mice in the Medical Research Council Laboratory of Molecular Biology were carried out in accordance with the UK Animals (Scientific Procedures) Act of 1986, with local ethical approval provided by the Medical Research Council Laboratory of Molecular Biology Animal Welfare Ethical Review Board and overseen institutionally by designated animal welfare officers (Animal Project License PP6471806). *TRAP2;Ai14* mice (*TRAP2*, JAX 03032; *Ai14*, JAX 7914) were kindly shared by L. Luo, Stanford University. All experiments related to the use of mice at the National Institute of Biological Sciences (NIBS), Beijing were approved by the Animal Care and Use Committee in accordance with the Regulations for the Administration of Affairs Concerning Experimental Animals of China. *DAT-Cre* (JAX 006660) and *Vgat-Cre* mice (JAX 028862) were obtained from the Jackson Laboratory. *Sert-Cre* mice (031028-UCD) were obtained from the University of California, Davis.

### Data preparation

The average and annotation templates of the Allen atlas were downloaded from the Allen Institute web portal (http://atlas.brain-map.org/). All the brain datasets, processing methods, brain names and imaging resolutions are summarized in Supplementary Table 1. All the samples were collected from adult mice. *TRAP2;Ai14* mice were used to generate somata-stained and nuclei-stained datasets. Mice used for these two datasets received five mild electrical foot shocks delivered through the floor. Each foot shock was up to 0.7 mA for up to 2 s. The brain samples were cleared based on a modified Adipo-Clear protocol[57]. Detailed protocols of TRAP2; Ai14 staining and c-fos staining have been described previously[58]. In brief, we used rabbit anti-Fos (Synaptic Systems, cat. no. 226003, lot 9-95; dilution, 1:500), rabbit anti-RFP (Rockland, 600-401-379, lot 42896, dilution, 1:500) and donkey anti-rabbit Alexa Fluor 647 (Jackson ImmunoResearch, 711-605-152, lot 161533; dilution, 1:500). The Sert-Stanford dataset is from the samples of the *Vglut3-Cre; Sert-Flp* group in the study of Ren et al.[57] and raw data are kindly shared by D. Friedmann. The DCN-Stanford dataset is from the study of Kebschull et al.[28] and the raw data are kindly shared by J. Kebschull.

Brain samples presented as Sert-NIBS, GABA-NIBS and DA-NIBS are generated at NIBS. Samples of Sert-NIBS, GABA-NIBS and DA-NIBS were prepared by an unpublished labeling system, LINCS (label individual neurons with chemical dyes and with controllable sparseness), which introduces chemical dyes (for example, Alexa Fluor 647) as the signaling molecule for photostable and ultrabright labeling. LINCS labeling was performed via a viral-genetic approach and achieved cell-type specificity using the Cre-loxP system and recombinant adeno-associated viral (AAV) vectors. For Sert-NIBS brains, 50 nl AAV was injected in the dorsal raphe nucleus of *Sert-Cre* mice. For GABA-NIBS brains, 50 nl AAV was injected in the ventral tegmental area unilaterally. For the DA-NIBS brain, 50 nl AAV was injected in the ventral tegmental area bilaterally. The brain samples were cleared based on the iDISCO+ protocol[14]. Samples were stored in dibenzyl ether until clear and imaged within 1 week. The cleared mouse brains were imaged in horizontal orientation with the dorsal side up on a light-sheet microscope (Ultramicroscope II, LaVision BioTec) using an sCMOS camera (Andor Neo) and a ×4/0.3 objective lens equipped with a 6-mm working distance dipping cap. Samples were scanned for 640 nm and 488 nm (autofluorescence) channels with dynamic focus using one-sided illumination with a step size of 3 μm.

The MRI dataset was accessed from https://github.com/dmac-lab/mouse-brain-atlas. We used the ex vivo brain MRI dataset for validating the brain region segmentation and whole-brain registration methods. The dataset provides annotations of 21 different brain regions[59]. The image resolution was 150 μm per voxel. As the size and shape of MRI brains are greatly different from the Allen atlas,

we resized and downsampled the Allen atlas to the MRI resolution for whole-brain registration.

In the task of whole-brain axon segmentation, we manually annotated ten large-sized cubes for quantitative evaluation and comparison, with a size of 600 × 600 × 225 voxels, including two cubes from Sert-Stanford, two cubes from DCN-Stanford, two cubes from Sert-NIBS, two cubes from GABA-NIBS and two cubes from DA-NIBS. The number of automatically annotated cubes for deep model training, including 'pure' axon cubes and 'pure' artifact cubes and mixed cubes after data augmentation, is provided in Supplementary Table 2. The size of automatically annotated cubes was 150 × 150 × 150 voxels. In the brain region segmentation task, we manually annotated 12 LSFM brains in the autofluorescence channel with six major brain regions (CTX, CB, CBX, BS, HPF and CP) for quantitative evaluation and comparison, including three somata-stained brains, three nuclei-stained brains, three Sert-Stanford brains and three Sert-NIBS brains. These annotations are also used for the evaluation of whole-brain registration. We also manually annotated nine LSFM brains in the autofluorescence channel with five small brain structures (act, fr, mtt, IPN and Hb) for quantitative evaluation and comparison of whole-brain registration in local areas, including three somata-stained brains, three nuclei-stained brains, three Sert-Stanford brains. We further manually selected 18 landmarks in Allen atlas and the LSFM brains for quantitative evaluation and comparison of whole-brain registration. The data used for brain region segmentation and whole-brain registration are summarized in Supplementary Table. 3. All training and prediction was performed using an NVIDIA GeForce RTX 3090 graphics-processing unit.

### Axon segmentation

**Automated cube annotation and data augmentation.** The manual selection of 'axon' and 'artifact' cubes is generally based on the axon projection patterns. To achieve high-quality training, representative 'axon' cubes were selected for brain regions receiving axonal innervation. For the 'axon' cubes, users were not required to select cubes with 100% pure axons, as a few highlighted noises would not influence the subsequent process. For the 'artifact' cubes, users can traverse the whole brain to select typical noises (for example, blood vessels, highlighted edges and bright points). We binarized the selected axon cubes and extracted the axons with a set of image-processing techniques, including a Gaussian filter with a kernel size of 3 × 3 × 3, following the difference of Gaussians to increase the visibility of axons and thresholding to get binarized axons. The threshold can be set automatically or by users after checking the 3D cube. Subsequently, to keep the tree-topological structure of axons, we employed image dilation to connect the adjacent fragments of the above binarized axons. Then the center lines were extracted to skeletonize the axons and were dilated to generate annotations with unified thickness (the center voxel extends one more voxel in six directions of the 3D space; left and right, top and bottom, front and back). In doing so, the axons are annotated automatically and in 3D (Supplementary Fig. 1b). For an axon cube with the size of 150 × 150 × 150 voxels, the automated annotation can be obtained within 5 min. We summarize the number of automatically annotated cubes in Supplementary Table 3.

We introduced three data augmentation strategies to simulate real scenarios of heterogeneous brain samples (CutMix[20], histogram matching[60] and local contrast enhancement[22]) (Fig. 2c). In the CutMix strategy, we randomly cut part of the axon cubes and the artifact cubes respectively and then mix the two partial cubes as a new cube, which includes both axons and artifacts. In the histogram matching, we randomly selected two cubes and transferred the histogram from one to another, simultaneously preserving the annotated axons. In the local contrast enhancement, we randomly change the intensity of stochastic axons to enhance the diversity of axon morphology and connectivity patterns. Given 100 cubes of axons and artifacts individually, our data

augmentation strategy can generate more than 1,000 cubes with annotations for deep model training (Supplementary Table 3).

**Network for axon segmentation.** We designed the axon segmentation based on the nnU-Net architecture[23]. The self-configuring parameters and settings were automatically computed according to the different training datasets. Similar to the 3D U-Net backbone[8], the network included six layers of encoder and six layers of decoder. Moreover, we introduced the axial attention module[61] in the last decoder layer for integrating self-attention to each axis independently. The kernel size was set as $3 \times 3 \times 3$, with the instance normalization and the activation function of LeakyReLU. The initial learning rate was set as 0.0003. The input image size was $128 \times 128 \times 128$ voxels, which were randomly cropped from the original training cubes with the size of $150 \times 150 \times 150$ voxels. Binary Cross Entropy loss with a Sigmoid layer (BCEWithLogitsLoss) was employed as the loss function during the network training. The loss function $\mathscr{L}_{AS}$ for axon segmentation can be formulated as:

$$\mathscr{L}_{AS} = \frac{1}{N} \sum_{n=1}^{N} \{-w[y_n \times \log \sigma(x_n) + (1 - y_n) \times \log(1 - \sigma(x_n))]\} \quad (1)$$

where $N$ indicates the number of voxels in a 3D cube. $\sigma$ indicates the Sigmoid function to normalize the predictions $x$ in [0,1]. $x$ indicates the predictions. $y$ indicates the ground-truth annotations. We set the pos_weight $w$ as 1 when $y$ equals to 0 (indicating the current voxel annotation is a background) and $w$ as 3 when $y$ equals to 1 (indicating the current voxel annotation is an axon). For the sparse axon cubes, we used the skeletonized annotations for the deep model training to keep the tree-topological structure of axons. For extremely dense axon cubes, the skeletonized annotations are also quite dense and cannot reflect the real structure of axons. Hence, we used binarized annotations for the deep model to recognize axons and artifacts. The network was trained within 550 epochs. The number of cubes used for training is summarized in Supplementary Table 3. In general, based on 1,000 cubes after data augmentation, the network training can be finished within 12 h. The prediction of a whole-brain (for example, with the size of $2,000 \times 2,500 \times 2,000$ voxels) for all stained axons can be finished within 6 h.

In the testing phase, ten large-sized cubes with a size of $600 \times 600 \times 225$ voxels are used for quantitative evaluation, including two cubes from Sert-Stanford, two cubes from Sert-NIBS, two cubes from GABA-NIBS, two cubes from DCN-Stanford and two cubes from DA-NIBS (Supplementary Fig. 1c). For the evaluation metric, we first employed the Dice score, which has been widely used for the evaluation of most image segmentation and registration tasks. As the Dice score indicates the voxel-wise volumetric scores that cannot well evaluate the connectivity of axon tubular structures, we also introduced the ClDice based on the intersection of center lines and axon volumes[30], which has also been employed for the evaluation of mouse brain vasculature segmentation[29]. Additionally, we also reported the ClPrecision and ClRecall to comprehensively reflect whether methods can well identify axons/artifacts and fully explore all stained axons (Extended Data Fig. 2a).

**Brain-style transfer.** We developed a deep model of brain-style transfer based on CycleGAN[33]. Different from the original version of CycleGAN, we introduce a brain outline segmentation subnetwork, namely CEA-Net (Supplementary Fig. 3), in the CycleGAN framework, for the automated segmentation and preservation of the brain outline between the input brain and the output style-transferred brain. As an example, we employed a somata-stained LSFM brain as the experimental brain and the Allen atlas as the atlas brain (Fig. 3a). In the brain-style transfer framework, Generator A included three convolutional layers and several residual blocks, which were trained to generate images from the

LSFM brain style to Allen style. In the meantime, a CEA-Net was also trained for the brain outline segmentation of synthetic Allen images, with ground-truth annotations of the brain outline from the original LSFM outline. This CEA-Net could keep the brain outline consistent between the synthetic Allen and the original LSFM. Generator B had the same architecture, which was trained to generate images from the Allen style to the LSFM brain style. The CEA-Net for brain outline segmentation was also embedded in Generator B, which could keep the brain outline consistent between the synthetic LSFM and the original Allen. Discriminator A included five fully convolutional layers, which were trained to differentiate LSFM brain-style images that were original or synthetic. Discriminator B had the same architecture, which was trained to differentiate Allen-style images that were original or synthetic. Accordingly, the overall loss function $\mathscr{L}_{BST}$ for brain-style transfer included the CycleGAN loss and the CEA-Net segmentation loss, which can be formulated as:

$$\mathscr{L}_{BST} = \mathscr{L}_{GAN}(G_A, D_B, X, Y) + \mathscr{L}_{GAN}(G_B, D_A, Y, X) \\ + \mathscr{L}_{cyc}(G_A, G_B) + \mathscr{L}_{BCE}(X) + \mathscr{L}_{BCE}(Y) \quad (2)$$

$$\mathscr{L}_{GAN}(G_A, D_B, X, Y) = \mathbb{E}_{y \sim p_{data}(y)}[\log(D_B(y))] \\ + \mathbb{E}_{x \sim p_{data}(x)}[\log(1 - D_B(G_A(x)))] \quad (3)$$

$$\mathscr{L}_{GAN}(G_B, D_A, Y, X) = \mathbb{E}_{x \sim p_{data}(x)}[\log(D_A(x))] \\ + \mathbb{E}_{y \sim p_{data}(y)}[\log(1 - D_A(G_B(y)))] \quad (4)$$

$$\mathscr{L}_{cyc}(G_A, G_B) = \mathbb{E}_{x \sim p_{data}(x)}[\|G_B(G_A(x)) - x\|_1] \\ + \mathbb{E}_{y \sim p_{data}(y)}[\|G_A(G_B(y)) - y\|_1] \quad (5)$$

$$\mathscr{L}_{BCE}(X) = -\sum_{n=1}^{N} x_n \times \log(\bar{x}_n) + (1 - x_n) \times \log(1 - \bar{x}_n) \quad (6)$$

$$\mathscr{L}_{BCE}(Y) = -\sum_{n=1}^{N} y_n \times \log(\bar{y}_n) + (1 - y_n) \times \log(1 - \bar{y}_n) \quad (7)$$

where the $\mathscr{L}_{GAN}$ indicates the adversarial losses, $\mathscr{L}_{cyc}$ indicates the cycle consistency loss, following the settings of the CycleGAN. $\mathscr{L}_{BCE}$ indicates the segmentation loss for brain outline segmentation and preservation. $G_A, G_B, D_A, D_B$ indicates Generator A, Generator B, Discriminator A and Discriminator B respectively. $X$ and $Y$ indicate the LSFM brain image datasets and Allen brain image datasets, respectively. $\bar{x}_n$ indicates the predictions X, where $x_n$ indicates the ground truth. $\bar{y}_n$ indicates the predictions Y, where $y_n$ indicates the ground truth.

In our implementation, the brain-style transfer is completed for each 2D image slice, for example, transferring hundreds of brain slices to Allen atlas style. The slices for the style transfer can be either from coronal, horizontal or sagittal views. In our experiments, all style-transferred models were trained on brain images in the coronal view, with a fixed image size of $320 \times 448$ pixels. We trained style-transfer models based on different experimental brain datasets and brain atlas. During the style-transfer training, the learning rate was set as 0.0002, with a batch size of 1. The network was trained within 200 epochs. One experimental brain with around 500 slices in the coronal view was enough to train a style-transfer model with a brain atlas. The deep model training for the brain style transfer can be finished within 12 h.

## Brain region segmentation
**Network for brain region segmentation.** The backbone of the brain region and outline segmentation is CEA-Net (Supplementary Fig. 3). CEA-Net is originally from CE-Net[62], which is a recently developed

semantic segmentation model, including two additional modules in comparison with the U-Net architecture (dense atrous convolution (DAC) and residual multi-kernel pooling (RMP)), which can better investigate local to global context cues for brain region segmentation. Based on the CE-Net, we introduce the attention gates[61] following the RMP module, which can better learn to segment brain regions with varying shapes and sizes. Considering the limited access of brain images with ground-truth manual annotations, we extended CEA-Net in a semi-supervised manner, namely Semi-CEA (Extended Data Fig. 6a), based on the most commonly used semi-supervised benchmarks, Mean Teachers[63]. Semi-CEA includes two models (the student model and the teacher model), where the parameters in the teacher model are first obtained from the student model trained in annotated data, by the exponential moving average weights. Then a consistency loss (mean square error; MSE) is computed based on the original prediction from the student model and the noisy prediction from the teacher model (prediction for the image with π angle rotation). Moreover, as some brain regions cannot be well identified in a single view (for example, HPF, CP and CBX), we further proposed a new multiview Semi-CEA framework for more accurate brain region segmentation (Fig. 3c). In our experiment, we trained the multiview Semi-CEA in coronal and horizontal views. For a specific brain region, the predictions in coronal and horizontal views were combined in 3D for the computation of multiview MSE loss. Accordingly, the overall loss function is the combination of the MSE loss and the semi-supervised segmentation loss (BCEWithLogitsLoss) in two views:

$$\mathcal{L}_{\text{BRS}} = \mathcal{L}_{\text{cor}} + \mathcal{L}_{\text{hor}} + \mathcal{L}_{3\text{D}} \tag{8}$$

$$\mathcal{L}_{\text{cor}} = \mathcal{L}_{\text{hor}} = \mathcal{L}_{\text{sup}} + \mathcal{L}_{\text{qua}} \tag{9}$$

$$\mathcal{L}_{\text{sup}} = -\sum_{i=1}^{N} \left( x_i \log y_i + (1 - x_i) \log(1 - y_i) \right) \tag{10}$$

$$\mathcal{L}_{\text{qua}} = \sum_{i=N}^{N+P} \left( \hat{y}_i - \hat{y}_i' \right)^2 \tag{11}$$

$$\mathcal{L}_{3\text{D}} = \sum_{i=1}^{N} \left( y_i^c - y_i^{\text{ht}} \right)^2 + \sum_{j=1}^{P} \left( \hat{y}_j^c - \hat{y}_j^{\text{ht}} \right)^2 \tag{12}$$

where $\mathcal{L}_{\text{cor}}$ and $\mathcal{L}_{\text{hor}}$ indicate the segmentation loss in coronal and horizontal views, respectively. $\mathcal{L}_{\text{cor}}$ and $\mathcal{L}_{\text{hor}}$ are computed in the same way (Semi-Loss), which includes both the $\mathcal{L}_{\text{sup}}$ and $\mathcal{L}_{\text{qua}}$. The training set consists of $N$ annotated brain image slices and $P$ unannotated brain images. $\mathcal{L}_{\text{sup}}$ indicates the supervised loss for annotated brain image slices (BCEWithLogitsLoss), where $\mathcal{L}_{\text{qua}}$ indicates the quadratic loss function for unannotated brain image slices. $x_i$ indicates the ground-truth annotation, where $y_i$ indicates the prediction of annotated brain region. $\hat{y}_i$ indicates the unannotated image predictions without π angle rotation, where $\hat{y}_i'$ indicates the unannotated image predictions after π angle rotation. Additionally, $\mathcal{L}_{3\text{D}}$ indicates the MSE loss of two views' prediction in 3D for annotated and unannotated images, where $y_i^c$, $\hat{y}_j^c$ indicates the annotated and unannotated image predictions in the coronal view. $y_i^{\text{ht}}$ and $\hat{y}_j^{\text{ht}}$ indicate the annotated and unannotated image predictions in the coronal view, which are transformed from the horizontal view.

When training the multiview Semi-CEA network, the network input image size was unified as 448 × 320 pixels in the coronal view and 512 × 448 pixels in the horizontal view. The initial learning rate was 0.0001, with a batch size of 16. RMSProp was adopted as the optimizer. The network was trained within 100 epochs. To facilitate the training of multiple brain regions, pairs of brain regions were trained together, such as CB and BS, CTX and CBX, CP and HPF. The brain region

segmentation model can be trained within 8 h. After the model training, one brain region with 500 slices in a whole brain can be quickly predicted within 1 min.

In the experiment of LSFM autofluorescence brain region segmentation, the Allen atlas brain slices in the coronal view were first used for training the multiview Semi-CEA model. Then the 12 LSFM brains were used for evaluation, where D-LMBmap transferred the 12 LSFM brains with Allen style for a more accurate prediction (Fig. 3e). We also trained the multiview Semi-CEA model based on four LSFM autofluorescence brains, with one from somata-stained, one from nuclei-stained, one from Sert-Stanford and one from Sert-NIBS. The remaining eight brains were used for evaluation (Fig. 3f and Extended Data Fig. 6c). In the experiment of MRI brain region segmentation, we used only one MRI brain for the multiview Semi-CEA training. Then seven MRI brains were used for evaluation (Supplementary Fig. 4a,b). As the MRI dataset does not annotate the CBX brain region, we only reported the Dice score of the other five brain regions. In the experiment of LSFM stained-specific brain region segmentation, we used the deep model trained on the Allen atlas for evaluation, where three LSFM stained-specific brains were transferred to Allen style before brain region segmentation (Supplementary Fig. 5a,b).

### Whole-brain registration
**Network architecture for whole-brain registration.** We designed a multiscale and multi-constraint DNN for the whole-brain 3D registration (Fig. 4a). As an example, we employed a somata-stained brain as the source brain and the Allen atlas as the reference brain. The network inputs included the original LSFM brain, the LSFM brain with Allen style, the segmented major brain regions of the LSFM brain, the Allen atlas and corresponding brain regions in the Allen atlas. The initial brain size was unified as 320 × 456 × 528 voxels, which is consistent with the original size of Allen atlas. The network downsamples all the inputs twice to 160 × 228 × 264 voxels and 80 × 114 × 132 voxels. Then the inputs in the minimum resolution (80 × 114 × 132 voxels) were first trained with a rigid transformation network, including a nine-layer convolution for feature extraction and a two-layer convolution for rotation and translation matrix computation. Similar to the architecture of rigid networks, the affine transformation network was trained to learn the deformation and translation matrix. The rigid and affine transformations in the minimum resolution inputs were also applied to a higher resolution, including the 160 × 228 × 264-voxel and 320 × 456 × 528-voxel sized inputs. Subsequently, we extended the VoxelMorph network[17] in a multiscale format, for the training of nonrigid deformation from the source brain to the reference brain. The VoxelMorph was first trained on the minimum resolution, with five layers of convolutional encoders for feature extraction of inputs and seven layers of convolutional decoders for the computation of deformation fields. The corresponding layers were connected by the concatenated skip connections. After training on the minimum resolution, the network parameters were fixed and used for the training of inputs with higher resolution. There were 16 layers of both convolution encoders and decoders in the second resolution network (160 × 228 × 264 voxels) and 24 layers of both convolutional encoders and decoders in the third resolution network (320 × 456 × 528 voxels). The loss function in the rigid network is the MSE loss for the similarity measuring between each brain region in source brain and the reference brain, whereas the affine network includes the MSE loss and the regularization loss. The loss function in the VoxelMorph network includes an unsupervised loss and an auxiliary data loss, following the settings from the original VoxelMorph network. The network was trained by integrating the above loss functions, where the overall loss function $\mathcal{L}_{WBR}$ for whole-brain registration can be formatted as:

$$\mathcal{L}_{\text{WBR}} = \mathcal{L}_r + \mathcal{L}_a + \mathcal{L}_v \tag{13}$$

$$\mathscr{L}_r = \mathscr{L}_{\text{sim}} = \begin{cases} \mathscr{L}_{\text{MI}}(X,Y) = -\sum_{x \in V_X} \sum_{y \in V_Y} p(x,y) \log\left(\frac{p(x,y)}{\sum_{x \in V_X} p(x,y) \sum_{y \in V_Y} p(x,y)}\right), \\ \qquad \text{for the original brain} \\ \mathscr{L}_{\text{CC}}(X,Y) = \frac{1}{N} \sum_{j=1}^{N} \frac{\sum_{i \in \delta_j}(X_i - \bar{X}_i)(Y_i - \bar{Y}_i)}{\sqrt{\sum_{i \in \delta_j}(X_i - \bar{X}_i)^2 \sum_{i \in \delta_j}(Y_i - \bar{Y}_i)^2}}, \\ \qquad \text{for style transferred brain} \\ \mathscr{L}_{\text{MSE}}(X,Y) = \sqrt{\frac{1}{N} \sum_{i=1}^{N}(X_i - Y_i)^2}, \\ \qquad \text{for brain regions} \end{cases}$$

(14)

$$\mathscr{L}_a = \mathscr{L}_{\text{sim}} + \mathscr{L}_{\text{ind}} + \mathscr{L}_{\text{rank}} \qquad (15)$$

$$\mathscr{L}_{\text{ind}} = \|A - I\|_F^2 + \|b\|_2^2 \qquad (16)$$

$$\mathscr{L}_{\text{rank}} = |\text{rank}(A) - 1| \qquad (17)$$

$$\mathscr{L}_v = \mathscr{L}_{\text{sim}} + \mathscr{L}_{\text{reg}} + \mathscr{L}_{\text{inv}} \qquad (18)$$

$$\mathscr{L}_{\text{reg}} = \|\nabla \Phi\|_1 \qquad (19)$$

$$\mathscr{L}_{\text{inv}} = \sqrt{\frac{1}{N} \sum_{i=1}^{N}(Y_i - I_i)^2}, \, I = Y \odot \Phi^{-1} \qquad (20)$$

$$\Phi^{-1} = (-\Phi_x \odot \Phi, -\Phi_y \odot \Phi, -\Phi_z \odot \Phi) \qquad (21)$$

where $\mathscr{L}_r, \mathscr{L}_a, \mathscr{L}_v$ indicate the loss functions of rigid, affine, multiscale VoxelMorph network, respectively. In the rigid network, there are three types of similarity loss function ($\mathscr{L}_{\text{sim}}$) in measuring the source brain and the reference brain (the mutual information loss ($\mathscr{L}_{\text{MI}}$), the cross-correlation loss ($\mathscr{L}_{\text{CC}}$) and the MSE loss ($\mathscr{L}_{\text{MSE}}$)). $X$ indicates the reference brain, where $Y$ indicates the registered brain. For a brain location, $p(x,y)$ indicates the probability of voxel values as $x$ in $X$ and $y$ in $Y$ at the same position. $V_X$ and $V_Y$ indicate the two sets of all voxel values in $X$ and $Y$, respectively. $\delta_j$ indicates the cube with the size of $s \times s \times s$ and the center of $j$. $\bar{X}_i$ indicates the average voxel value of $\delta_j$ in $X$. The $\mathscr{L}_{\text{MI}}$ is employed for the similarity measuring between the original source brain and the reference brain. The $\mathscr{L}_{\text{CC}}$ is employed for the similarity measuring between the style-transferred source brain and the reference brain. The $\mathscr{L}_{\text{MSE}}$ is employed for the similarity measuring between each brain region in source brain and the reference brain. In the affine network, besides the similarity loss ($\mathscr{L}_{\text{sim}}$), it also includes two more items ($\mathscr{L}_{\text{ind}}$ for the penalization of the affine transformation from the identity and $\mathscr{L}_{\text{rank}}$ for constraining brain scaling in the affine transformation matrix). $A$ indicates the predicted affine transformation matrix, where $b$ indicates the offset in the affine transformation. In the multiscale VoxelMorph network, besides the similarity loss ($\mathscr{L}_{\text{sim}}$), it also includes two more items ($\mathscr{L}_{\text{reg}}$ for the constrain of the gradient in nonlinear to keep smooth transformation and $\mathscr{L}_{\text{inv}}$ for the constraint of nonlinear transformation space). The $\Phi$ indicates the nonlinear transformation space, where $\Phi^{-1}$ indicates the approximate inverse transformation in the nonlinear transformation space. $\odot$ indicates the interpolation.

For the multi-constraints, the original source brain, style-transferred source brain and the source brain regions, were set as inputs for the deep model training in different channels, which were used for the similarity computation with the corresponding reference brain and reference brain regions, respectively. The original source brain was assigned with higher weights in the loss computation (70%), whereas the style-transferred source brain and brain regions were assigned with lower weights in the loss computation (15%). In each

training batch, the above constraints' losses in different channels were integrated for model updating. $\Phi_x$, $\Phi_y$ and $\Phi_z$ indicate the offset in the $x$, $y$ and $z$ directions, respectively.

During the network training, the initial learning rate in the rigid and affine network was set as 0.0001. The initial learning rate in the VoxelMorph was set as 0.001. Adam was adopted as the optimizer, with a batch size of one. In the training on the minimum resolution inputs ($80 \times 114 \times 132$-voxel-sized brains), the training epoch was set as 1,000, whereas in the training of the higher resolution inputs ($160 \times 228 \times 264$-voxel and $320 \times 456 \times 528$-voxel-sized brains), the training epoch was set as 300.

In the experiment of LSFM autofluorescence brain registration to Allen atlas, ten LSFM brains were used for multiscale and multi-constraint deep registration model training, where the constraints of all brain regions were directly obtained by the automated brain region segmentation. Then the 12 LSFM brains with manual brain region annotations were used for evaluation (Fig. 4c and Extended Data Fig. 7a). In the registration evaluation of small brain structures, nine LSFM brains with manual brain region annotations were used for evaluation (Fig. 4d and Supplementary Table 4). In the registration evaluation of landmark distance, we employed the landmark extraction method – 2.5D corner detection, presented in mBrainAligner[13]. We filtered out 18 landmarks that were automatically detected across all the testing LSFM brains and the Allen atlas (Extended Data Fig. 9d). In the experiment of MRI brain registration, five MRI brains were used for model training, where the constraints of brain regions were obtained from the MRI datasets by manual annotations. Then three MRI brains were used for evaluation (Extended Data Fig. 7b,d). In the experiment of LSFM stained-specific brain registration, we directly used the deep model trained on the LSFM autofluorescence brains for evaluation, where the inputs were the style-transferred LSFM stained-specific brains (Allen style) (Extended Data Fig. 7c,e).

### Software and algorithms
Software resources used are detailed in the following table.

| Software | Developer | Link |
| --- | --- | --- |
| PyTorch v.1.11.0 | The Linux Foundation | https://pytorch.org |
| PyCharm v.2022.3 | JetBrains | https://www.jetbrains.com/ |
| Anaconda v.4.12.0 | Anaconda | https://www.anaconda.com/ |
| Python v.3.8.8 | Python Software Foundation | https://www.python.org/ |
| ImageJ (Fiji) v.1.53q | National Institutes of Health | https://imagej.nih.gov/ij/index.html |
| Elastix v.4.8 | Image Sciences Institute | https://elastix.lumc.nl/ |
| IMARIS v.9.0.1 | Oxford Instruments | https://imaris.oxinst.com/ |
| Vaa3D v.4.001 | Allen Institute | https://github.com/Vaa3D |
| ITK-SNAP v.3.6.0 | Yushkevich et al.[64] | www.itksnap.org |
| Allen Institute's Common Coordinate Framework (CCFv3) | Allen Institute's for Brain Science | http://atlas.brain-map.org |
| GraphPad Prism v.9.4.1 | GraphPad | https://www.graphpad.com/ |

### Reporting summary
Further information on research design is available in the Nature Portfolio Reporting Summary linked to this article.

### Data availability
The datasets generated and analyzed in this study are available on the D-LMBmap's GitHub page (https://github.com/lmbneuron/D-LMBmap). All the automatically annotated and manually annotated

samples are also available on GitHub. All the source data files for each figure are available at https://doi.org/10.5281/zenodo.8123585. The full-resolution LSFM brain images are available on request. MRI brain data are available at https://github.com/dmac-lab/mouse-brain-atlas. The Allen Institute's Common Coordinate Framework (CCFv3) atlas is available at http://atlas.brain-map.org. Source data are provided with this paper.

## Code availability

The source code of all D-LMBmap modules, including automated axon segmentation, brain-style transfer, brain region segmentation, whole-brain 3D registration, along with the executable files of D-LMBmap software and sample data can be found at D-LMBmap's GitHub page (https://github.com/lmbneuron/D-LMBmap).

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

## Acknowledgements

This work was supported by the Medical Research Council, as part of United Kingdom Research and Innovation (UK Research and Innovation) (MC_UP_1201/22). For the purpose of open access, the Medical Research Council Laboratory of Molecular Biology has applied a CC BY public copyright license to any Author Accepted Manuscript version arising. This work was also partially funded by NARSAD Young Investigator Award (2020, BBRF) to J.R. and Ministry of Science and Technology (2022ZD0206700) and the Beijing Municipal Government of P.R.C. to R.L. We thank D. Friedmann for advice on Adipo-Clear, J. Kebschull and D. Friedmann for data sharing, and L. Luo, M. Hastings, A.M.J. Adams and J. Song for critique on the manuscript.

## Author contributions

J.R. conceptualized and supervised the project. Z.L. led the development of D-LMBmap. J.R. and Z.L. designed all the experiments. Z.L., Z.S. and Y.Z. designed and implemented the automated axon segmentation module. Z.L., J.L. and X.H. designed and implemented the brain-style transfer and brain region segmentation module. Z.L., H.Z. and E.Z. designed, implemented and developed the whole-brain registration module and the D-LMBmap software. Z.L., X.Z., J.L., X.H., Y.W., Y.Z., P.L., Z.S. and J.R. co-annotated and checked the testing data. S.Z. and R.L. generated the LSFM imaging data from the NIBS. J.R. generated the LSFM imaging data from the LMB. J.R., Z.L. and R.N.S. wrote the manuscript with input from all co-authors.

## Competing interests

The authors declare no competing interests.

## Additional information

**Extended data** is available for this paper at https://doi.org/10.1038/s41592-023-01998-6.

**Correspondence and requests for materials** should be addressed to Jing Ren.

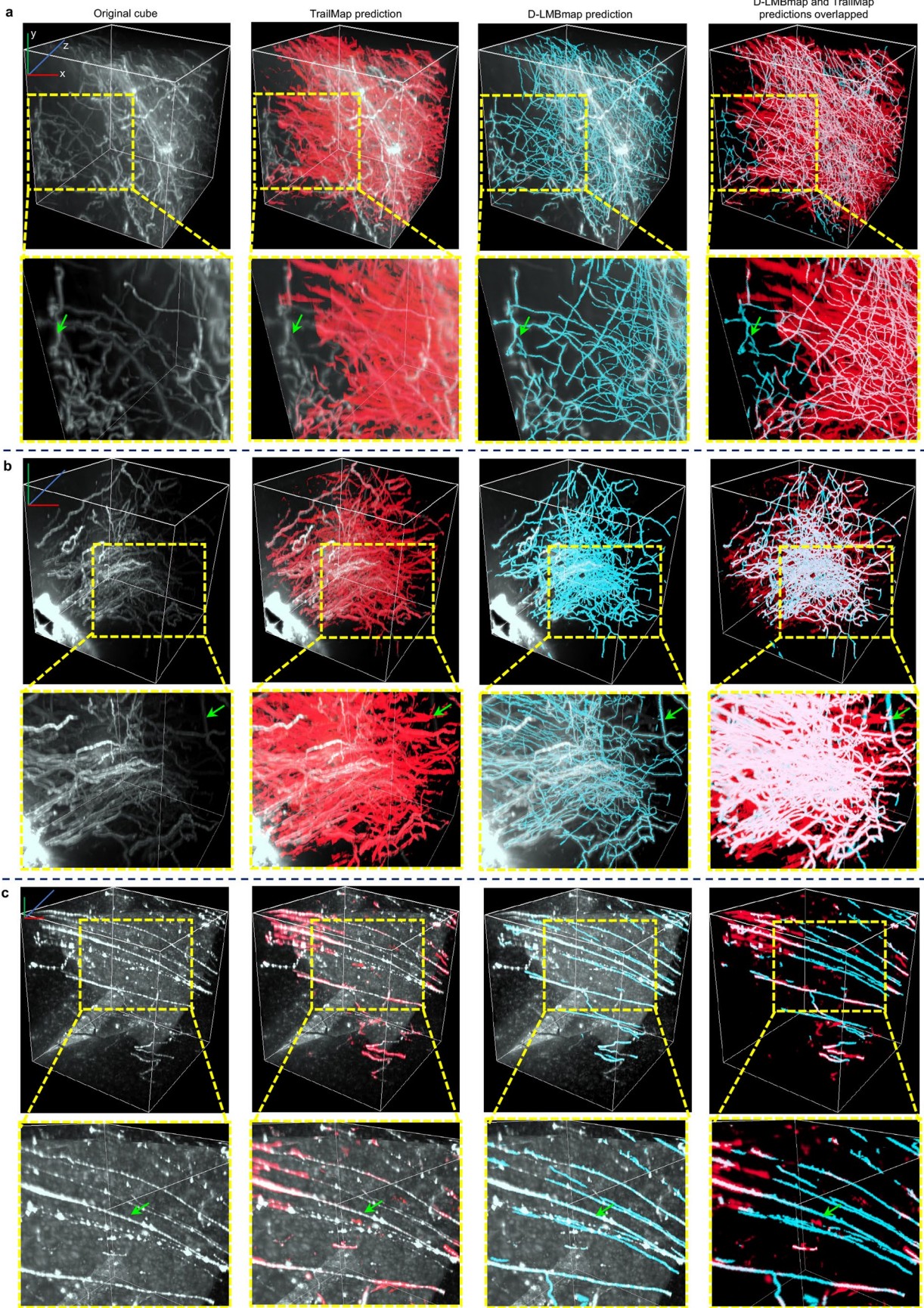

**Extended Data Fig. 1 | Comparison of axon segmentation results with enlarged views between the TrailMap and the D-LMBmap. a, b, c,** Three representative cubes with enlarged views (the first column), as well as the axon segmentation prediction of TrailMap (the second column), D-LMBmap (the third column) and the overlapped prediction between the TrailMap and the D-LMBmap (fourth column). The cube size is 200 × 200 × 225 voxels. (Scale bar, X, Y, Z = 60 μm.).

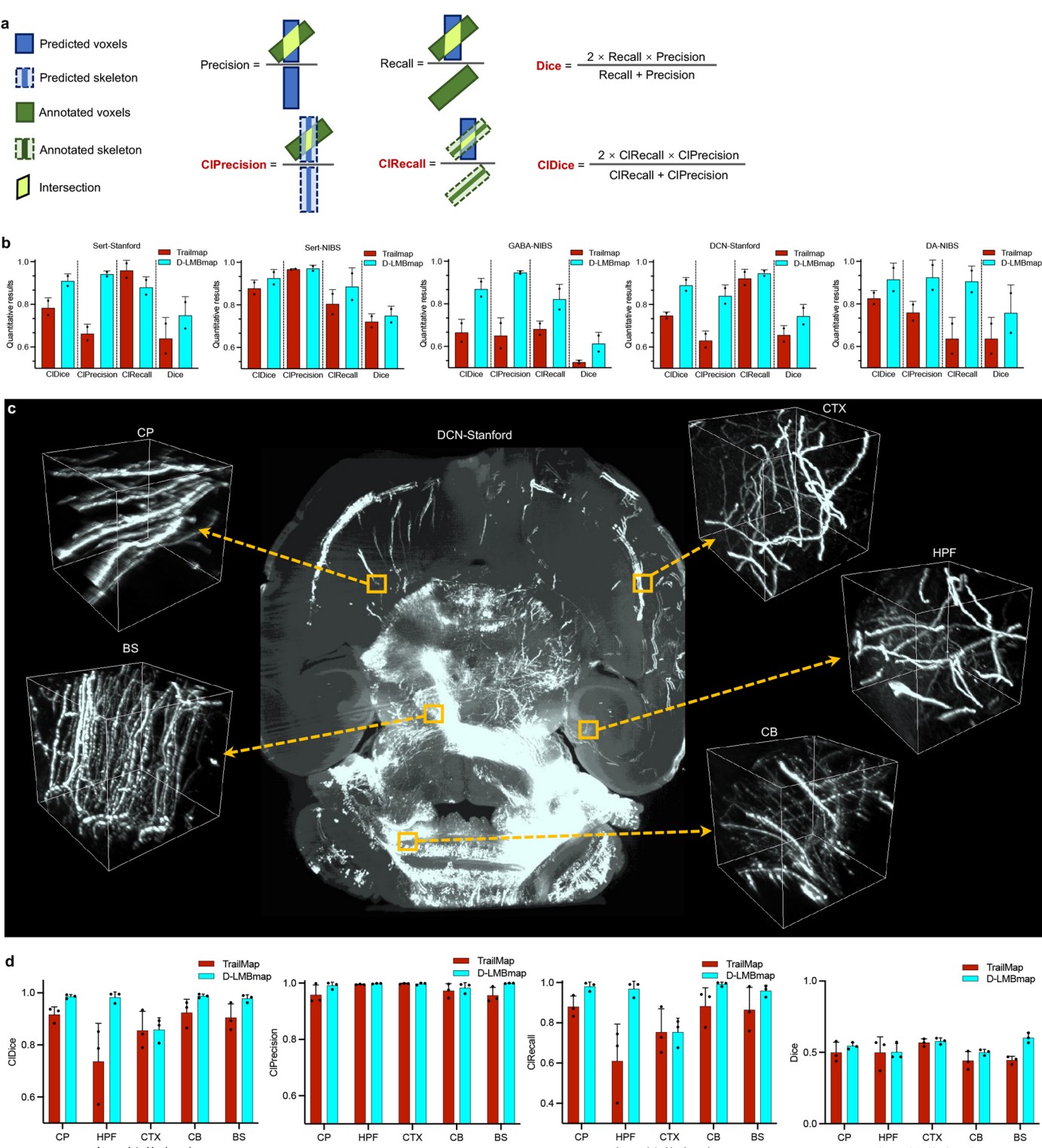

**Extended Data Fig. 2 | Metrics used for axon segmentation and effectiveness evaluation of diverse cubes. a**, Metrics used for suitable and comprehensive evaluations of axon segmentation, including ClDice, ClPrecision, ClRecall and Dice score. **b**, Quantitative comparison between D-LMBmap and TrailMap under the evaluation of ClDice, ClPrecision, ClRecall, and Dice score using the manually annotated cubes in different types of axons respectively, n = 2 cubes in each axon types. Error bars, mean ± SD. **c**, Representative cubes of axons distributed in different major brain regions in a DCN-Stanford brain. Cubes are selected from the CP, BS, CB, HPF, and the CTX. **d**, Quantitative evaluation and comparison between TrailMap and D-LMBmap for the effectiveness of handling diverse axon across different brain regions. For each testing session, testing cubes (150 × 150 × 150) were selected from one out of the five brain regions, including the CP, HPF, CTX, CB, and the BS. Training cubes were selected from the rest four brain regions. ClDice, ClPrecision, ClRecall, and Dice scores were used as evaluation metrics, n = 3 cubes in each testing brain region. Error bars, mean ± SD.

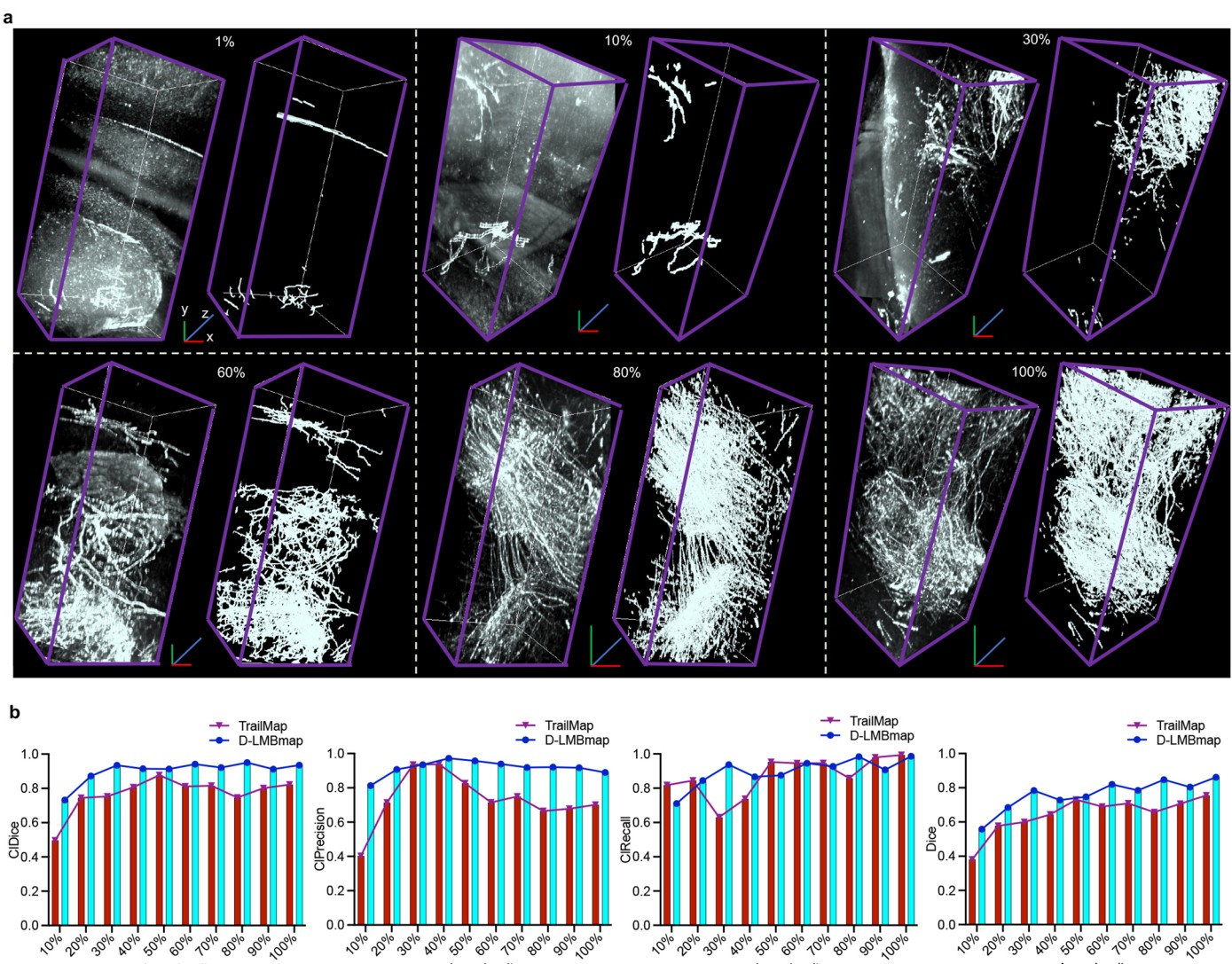

**Extended Data Fig. 3 | Effectiveness evaluation of D-LMBmap for predicting cubes with different axon densities. a**, Representative cubes with different axon densities ranging from 1% to 100%. Here, 100% density is defined as the axons occupying around 800 million voxels in one cube (200 × 200 × 450 voxels, scale bar, X, Y = 240 µm, Z = 90 µm.). **b**, Quantitative evaluation and comparison between the TrailMap and the D-LMBmap when predicting cubes with different axon densities under the metrics of ClDice, ClPrecision, ClRecall, and Dice score.

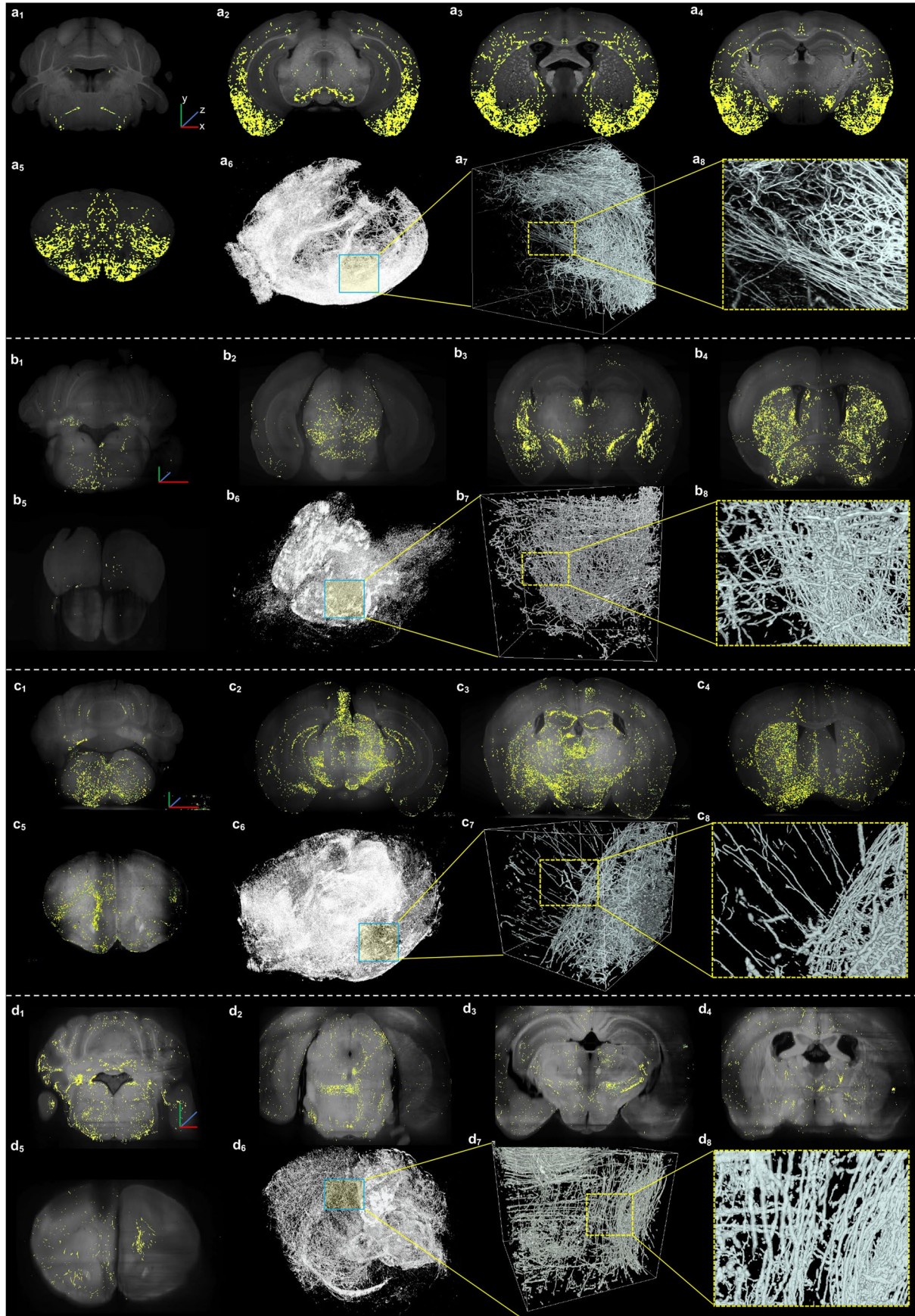

**Extended Data Fig. 4 | See next page for caption.**

**Extended Data Fig. 4 | Whole-brain axon segmentation of different cell types.**
**a**, ($A_1$-$A_5$), coronal slices of segmented axons in a Sert-Stanford brain. ($A_6$), 3D segmentation of whole-brain axons in the Sert-Stanford brain. ($A_7$ - $A_8$), zoom-in views of segmented axons from $A_6$. **b**, ($B_1$-$B_5$), coronal slices of segmented axons in a DA-NIBS brain. ($B_6$), 3D segmentation of whole-brain axons in the DA-NIBS brain. ($B_7$ - $B_8$), zoom-in views of segmented axons from $B_6$. **c**, ($C_1$-$C_5$), coronal slices of segmented axons in a GABA-NIBS brain. ($C_6$), 3D segmentation of whole-brain axons in the GABA-NIBS brain. ($C_7$ - $C_8$), zoom-in views of segmented axons from $C_6$. **d**, ($D_1$-$D_5$), coronal slices of segmented axons in a DCN-Stanford brain. ($D_6$), 3D segmentation of whole-brain axons in the DCN-Stanford brain. ($D_7$ - $D_8$), zoom-in views of segmented axons from $D_6$. (Scale bar, X, Y, Z = 1 mm.).

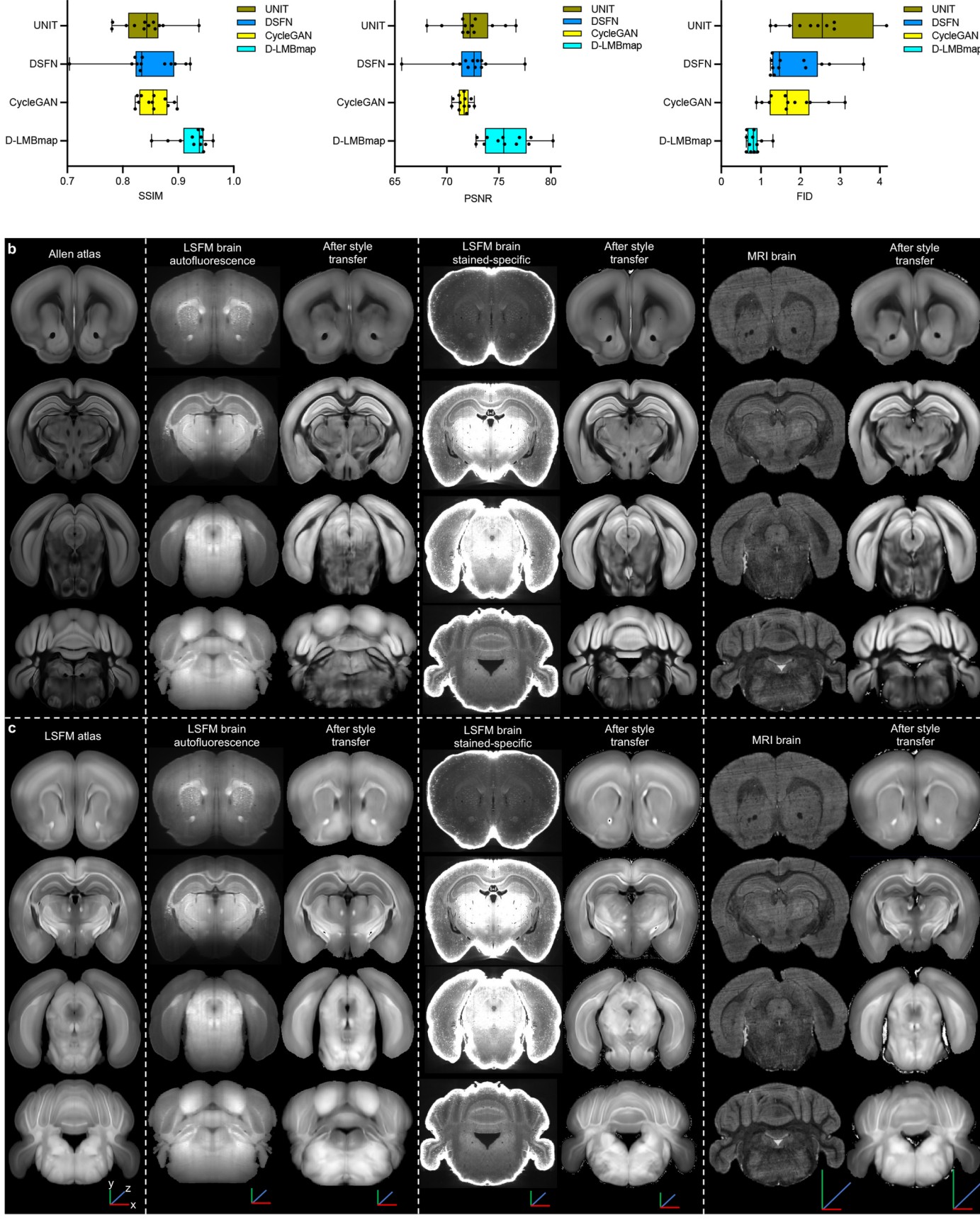

**Extended Data Fig. 5 | See next page for caption.**

**Extended Data Fig. 5 | Style transfer solution employed in D-LMBmap.**
**a**, Quantitative comparison of four style transfer methods, UNIT, DSFN, CycleGAN and D-LMBmap, under the evaluation metrics of SSIM, PSNR, and FID (n = 12 brains) at the whole brain level. The higher the SSIM and PSNR scores, the more similar the reconstructed LSFM brain is to the original LSFM brain. Conversely, the lower the FID score, the more similar the reconstructed LSFM brain is to the original LSFM brain. Box plot: center line, median; box limits, upper and lower quartiles; whiskers, 1.5× interquartile range; points, individual data points. **b**, Sample brains transferred into Allen atlas style. Here are images of a LSFM brain in the autofluorescence channel, an LSFM brain in the stained-specific channel, and an MRI brain which has been transferred into the Allen atlas style. (Scale bar, X, Y, Z = 1 mm.). **c**, Sample brains transferred into LSFM atlas style. The figure shows the LSFM brain in the autofluorescence channel, the LSFM brain in the stained-specific channel, and the MRI brain transferred into the LSFM brain atlas style. (Scale bar, X, Y, Z = 1 mm.).

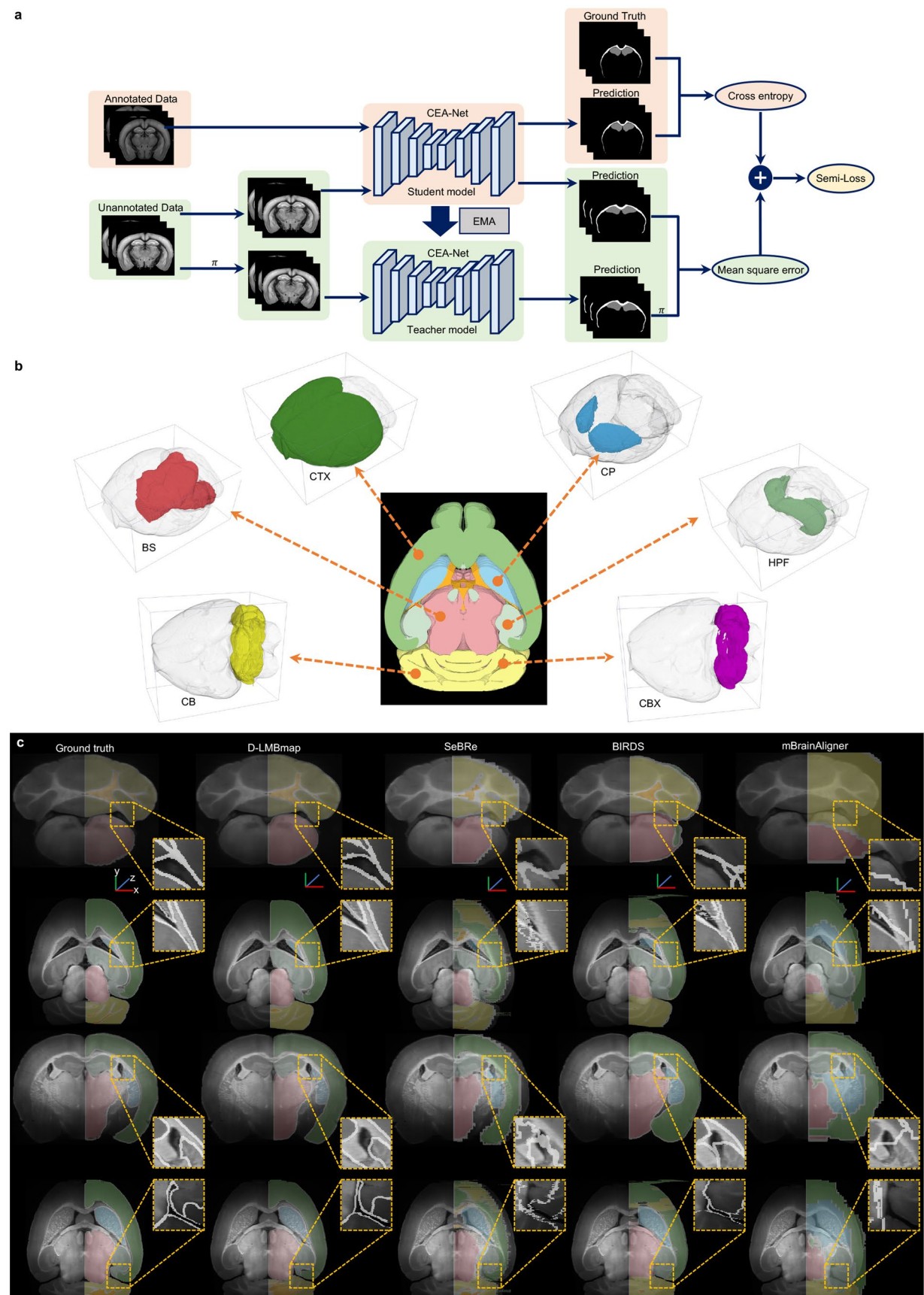

**Extended Data Fig. 6 | See next page for caption.**

**Extended Data Fig. 6 | The architecture of Semi-CEA and the segmentation of major brain regions. a**, Semi-CEA is developed for automated brain outline and brain region segmentation. It is the semi-supervised version of the CEA-Net, where the rotation consistency is applied for the computation of semi-supervised loss. **b**, The six major brain regions used for the evaluation of brain region segmentation. The BS, CP, HPF, and CTX are adjacent to each other, and the CBX is a subregion of the CB. **c**, The comparison of brain region segmentation results of an LSFM autofluorescence brain among D-LMBmap, SeBRe, BIRDS, and mBrainAligner. (Scale bar, X, Y, Z = 1 mm.).

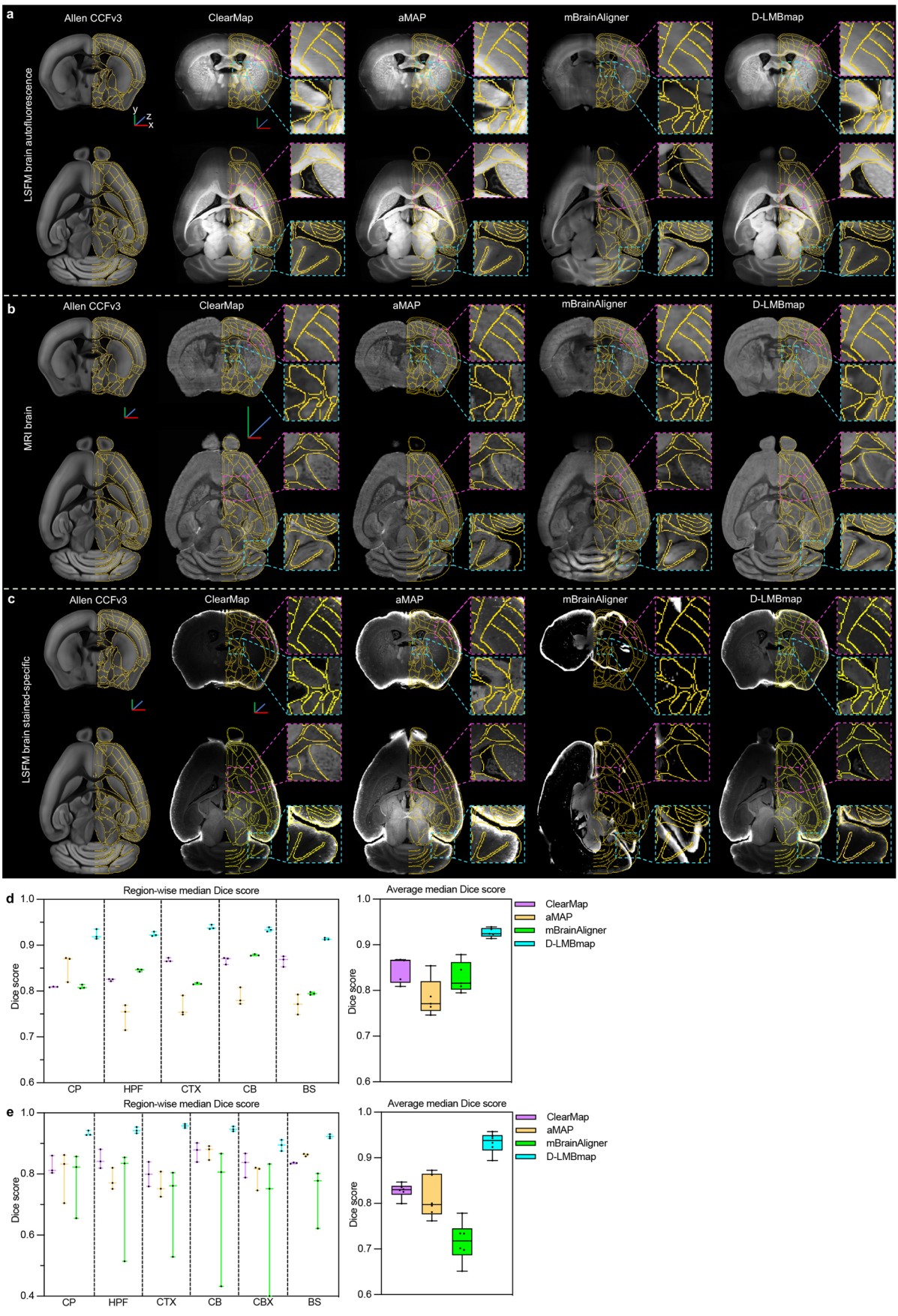

**Extended Data Fig. 7 | See next page for caption.**

**Extended Data Fig. 7 | Comparison of cross modality whole brain registration results using different methods. a**, Representative results of whole-brain registration of an LSFM brain in the autofluorescence channel by ClearMap, aMAP, mBrainAligner, and D-LMBmap. (Scale bar, X, Y, Z = 1 mm.). **b**, Representative results of whole-brain registration of an MRI brain by the four methods. (Scale bar, X, Y, Z = 1 mm.). **c**, Representative results of whole-brain registration of an LSFM brain in the stained-specific channel by the four methods. (Scale bar, X, Y, Z = 1 mm.). **d**, Quantitative comparison of the whole-brain registration results on MRI brains generated by ClearMap, aMAP, mBrainAligner, and D-LMBmap (n = 3). Left: region-wise median Dice score for five brain regions (CP, HPF, CTX, CB, and BS). Right: average median Dice score of different methods. Box plot: center line, median; box limits, upper and lower quartiles; whiskers, 1.5× interquartile range; points, individual data points. **e**, Quantitative comparison of the whole-brain registration results on LSFM brains in the stained-specific channel generated by ClearMap, aMAP, mBrainAligner, and D-LMBmap (n = 3). Left: region-wise median Dice score for six brain regions (CP, HPF, CTX, CB, CBX, and BS). Right: average median Dice score of different methods. Box plot: center line, median; box limits, upper and lower quartiles; whiskers, 1.5× interquartile range; points, individual data points.

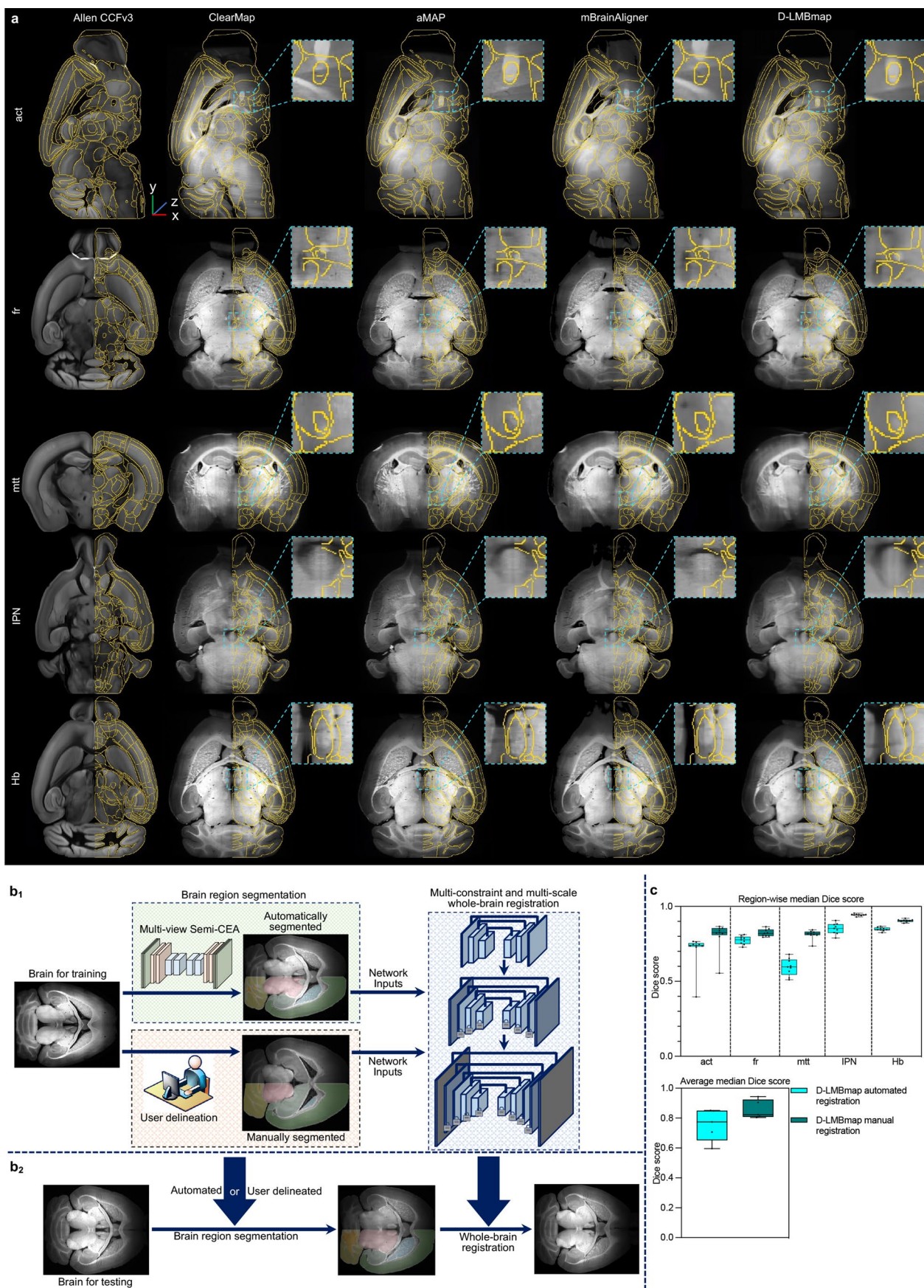

**Extended Data Fig. 8 | See next page for caption.**

**Extended Data Fig. 8 | Evaluation of registration methods for small brain structures. a**, Representative images of whole-brain registration on small brain structures of an LSFM brain in the autofluorescence channel by ClearMap, aMAP, mBrainAligner, and D-LMBmap. (Scale bar, X, Y, Z = 1 mm.). **b**, Schematic drawings of the pipelines employed in the D-LMBmap whole-brain registration based on brain region constraints. $B_1$. In the training pipeline of D-LMBmap for whole-brain registration, brain region constraints can either be automatically obtained using our developed Multi-view Semi-CEA network for automated brain region segmentation or manually delineated by users. These brain region constraints are then used as input for the training of the whole-brain registration model. $B_2$. In the testing pipeline of D-LMBmap for whole-brain registration, the

brain regions of the testing brain can be obtained either automatically using the Multi-view Semi-CEA network or manually delineated by users. Once this is done, registration can be automatically achieved using the trained whole-brain registration deep model. **c**, Registration results of whole-brain registration on small brain structures of 9 LSFM brains in the autofluorescence channel by D-LMBmap using different brain region constraints, where the cyan results indicate the results using the automated segmentation pipeline, and the deep blue results indicate the results using manual delineation pipeline. Box plot: center line, median; box limits, upper and lower quartiles; whiskers, 1.5× interquartile range; points, individual data points.

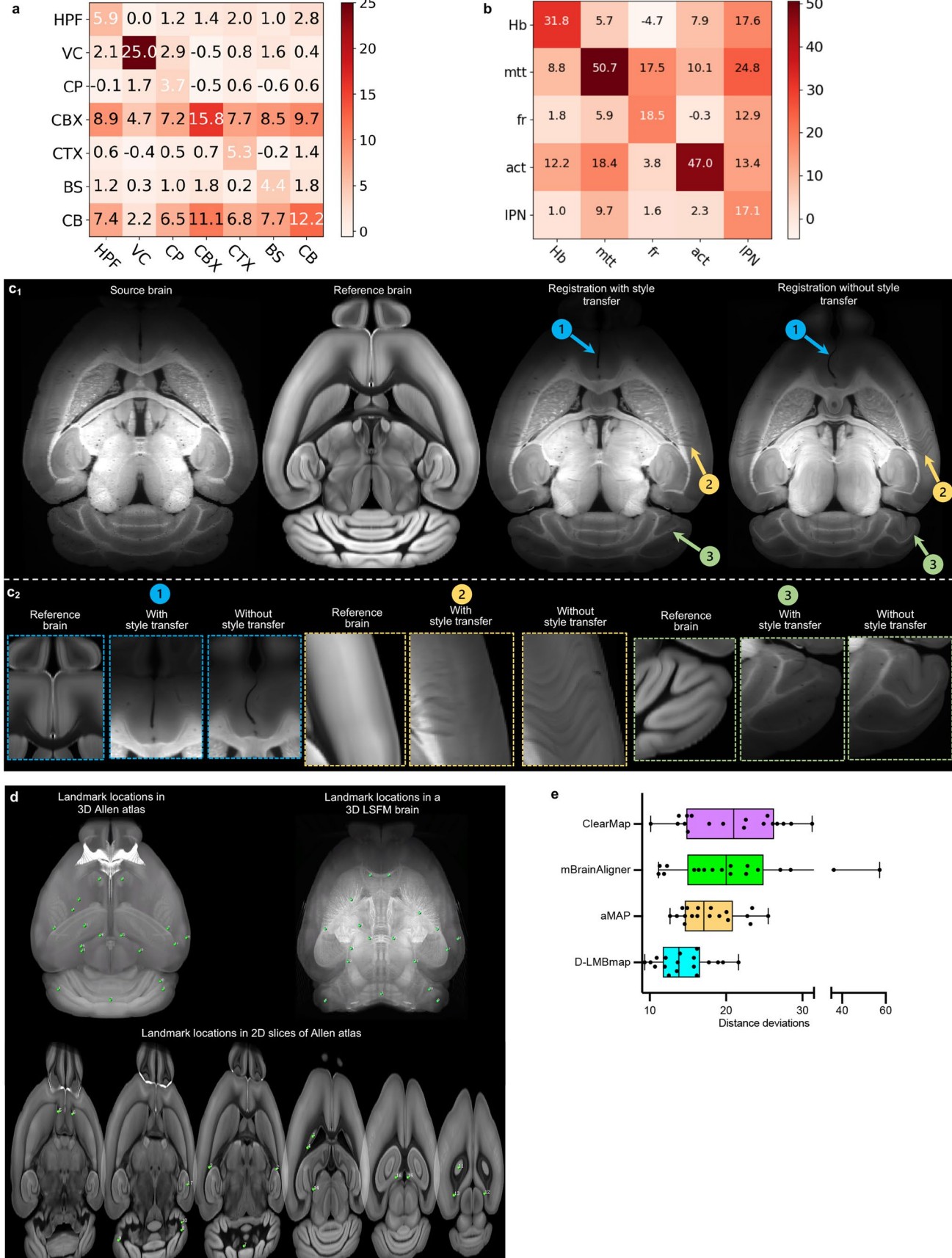

**Extended Data Fig. 9 | See next page for caption.**

**Extended Data Fig. 9 | Evaluation of the multi-constraint strategy in the D-LMBmap whole brain registration module. a**, The confusion matrix for validating the effectiveness of individual major brain region constraints on LSFM brains (n = 12). The X-axis indicates individual brain region constraints, and the Y-axis indicates the change of the brain region Dice score after the single brain region constraint is added. (Ventricles, VC = LV + 3$^{rd}$V + AQ). **b**, The confusion matrix for validating the effectiveness of individual small brain structure constraint on LSFM brains (n = 9). **c**, Ablation study of the effectiveness of brain style transfer in the whole-brain registration. C$_1$. From left to right, the original LSFM brain as the source brain, the Allen atlas as the reference brain, the registered brain based on D-LMBmap and using style transfer, and the registered brain based on D-LMBmap without using style transfer. C$_2$. Comparison across the reference brain, the registered brain with and without brain style transfer in the magnified view of the frontal cortex (in blue squares), the somatosensory cortex (in yellow squares), and CBX (in green squares). **d**, The locations of 18 manually selected landmarks are presented in the 3D Allen atlas, the corresponding 2D slice of the Allen atlas, and a 3D LSFM brain. **e**, Quantitative comparison of the whole-brain registration results on LSFM brains (n = 6) by computing the average landmark distance generated by ClearMap, aMAP, mBrainAligner, and D-LMBmap. Box plot: center line, median; box limits, upper and lower quartiles; whiskers, 1.5× interquartile range; points, individual data points.

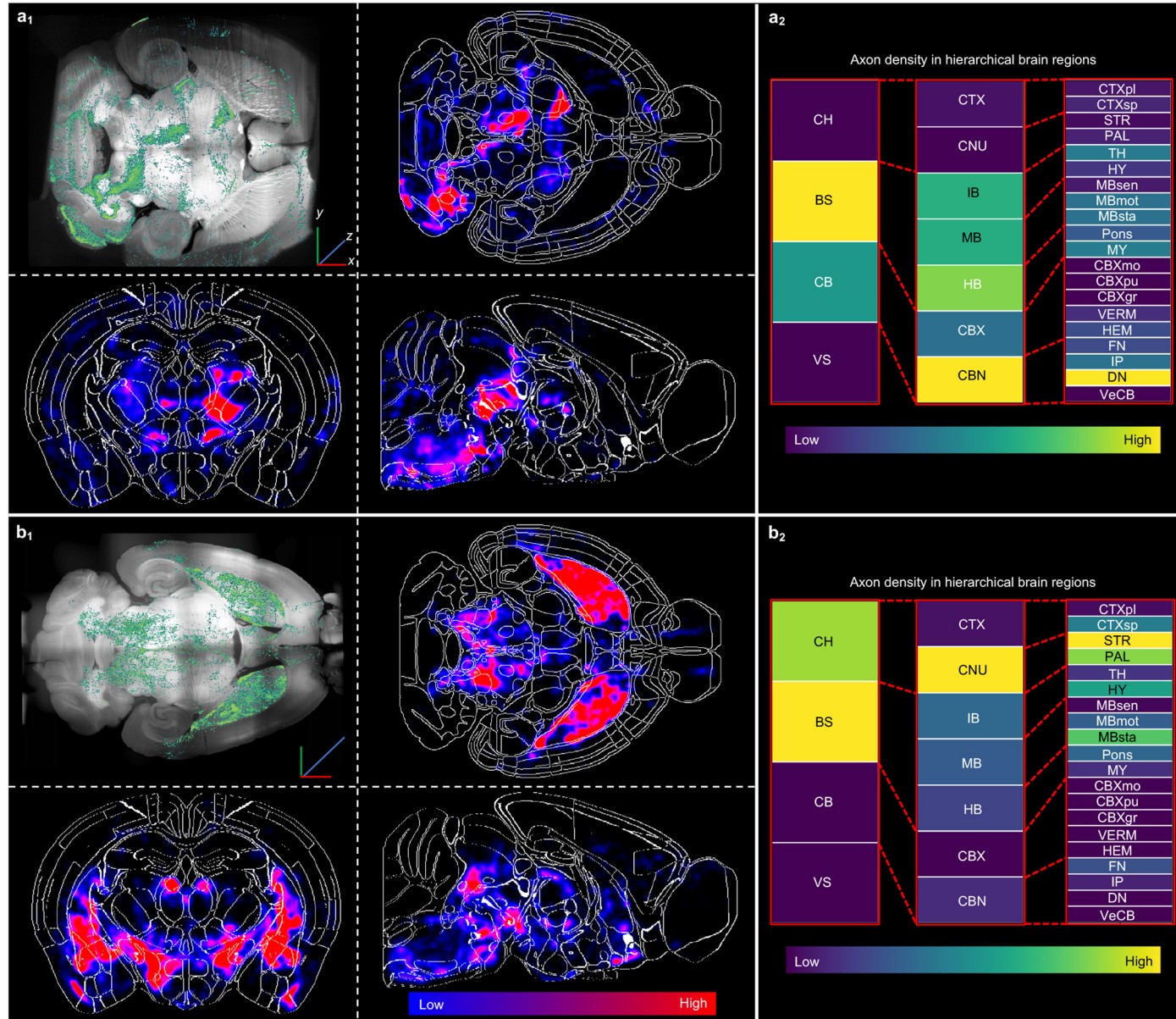

**Extended Data Fig. 10 | Whole-brain circuit projections and regional quantification. a**, Whole-brain circuitry profiling of DCN-Stanford brains. A₁. Axon segmentation results on ten horizontal brain slices overlaid, and average axon distribution heatmaps in horizontal, coronal, and sagittal views (n = 3). A₂. Axon density in hierarchical brain regions based on the Allen atlas.

(Scale bar, X, Y, Z = 1 mm.). **b**, Whole-brain circuitry profiling of DA-NIBS brains. B₁. Axon segmentation results on ten horizontal brain slices overlaid, and axon distribution heatmaps in horizontal, coronal, and sagittal views (n = 1). B₂. Axon density in hierarchical brain regions based on the Allen atlas. (Scale bar, X, Y, Z = 1 mm.).

# Reporting Summary

## Statistics

For all statistical analyses, confirm that the following items are present in the figure legend, table legend, main text, or Methods section.

| n/a | Confirmed | |
|---|---|---|
| ☐ | ☒ | The exact sample size (*n*) for each experimental group/condition, given as a discrete number and unit of measurement |
| ☐ | ☒ | A statement on whether measurements were taken from distinct samples or whether the same sample was measured repeatedly |
| ☐ | ☒ | The statistical test(s) used AND whether they are one- or two-sided<br>*Only common tests should be described solely by name; describe more complex techniques in the Methods section.* |
| ☒ | ☐ | A description of all covariates tested |
| ☐ | ☒ | A description of any assumptions or corrections, such as tests of normality and adjustment for multiple comparisons |
| ☐ | ☒ | A full description of the statistical parameters including central tendency (e.g. means) or other basic estimates (e.g. regression coefficient) AND variation (e.g. standard deviation) or associated estimates of uncertainty (e.g. confidence intervals) |
| ☐ | ☒ | For null hypothesis testing, the test statistic (e.g. *F*, *t*, *r*) with confidence intervals, effect sizes, degrees of freedom and *P* value noted<br>*Give P values as exact values whenever suitable.* |
| ☒ | ☐ | For Bayesian analysis, information on the choice of priors and Markov chain Monte Carlo settings |
| ☒ | ☐ | For hierarchical and complex designs, identification of the appropriate level for tests and full reporting of outcomes |
| ☒ | ☐ | Estimates of effect sizes (e.g. Cohen's *d*, Pearson's *r*), indicating how they were calculated |

*Our web collection on statistics for biologists contains articles on many of the points above.*

## Software and code

Policy information about availability of computer code

| Data collection | We use imageJ (Fiji) (version 1.53q) to select manual annotated cubes. ITK-SNAP (version 3.6.0) was used for manual annotation of brain images. |
|---|---|
| Data analysis | All the softwares used in this study are listed below as "Software, Developer, Link".<br>PyTorch 1.11.0, The Linux Foundation, https://pytorch.org<br>PyCharm 2022.3, JetBrains, https://www.jetbrains.com/<br>Anaconda 4.12.0, Anaconda Inc, https://www.anaconda.com/<br>Python 3.8.8, Python Software Foundation, https://www.python.org/<br>ImageJ (Fiji) 1.53q, National Institutes of Health, https://imagej.nih.gov/ij/index.html<br>Elastix 4.8, Image Sciences Institute, https://elastix.lumc.nl/<br>IMARIS 9.0.1, Oxford Instruments, https://imaris.oxinst.com/<br>Vaa3D 4.001, Allen Institute, https://alleninstitute.org/what-we-do/brain-science/research/products-tools/vaa3d/<br>ITK-SNAP 3.6.0, Paul A. Yushkevich et al. 2006, www.itksnap.org<br>Allen Institute's Common Coordinate Framework (CCFv3), Allen Institute's for Brain Science, http://atlas.brain-map.org<br>GraphPad Prism 9.4.1, GraphPad,  https://www.graphpad.com/ |

For manuscripts utilizing custom algorithms or software that are central to the research but not yet described in published literature, software must be made available to editors and reviewers. We strongly encourage code deposition in a community repository (e.g. GitHub). See the Nature Portfolio guidelines for submitting code & software for further information.

## Data

Policy information about availability of data

All manuscripts must include a data availability statement. This statement should provide the following information, where applicable:

- Accession codes, unique identifiers, or web links for publicly available datasets
- A description of any restrictions on data availability
- For clinical datasets or third party data, please ensure that the statement adheres to our policy

The datasets generated and analysed in this study are available at the D-LMBmap's Github page (https://github.com/lmbneuron/D-LMBmap). All the automatically annotated and manually annotated samples are also available at the Github page. All the source data files for each figure are available at https://doi.org/10.5281/zenodo.8123585. The full resolution LSFM brain images are available on request. All the MRI brain data are available at  https://github.com/dmac-lab/mouse-brain-atlas. The Allen Institute's Common Coordinate Framework (CCFv3) atlas are available at http://atlas.brain-map.org.

## Human research participants

Policy information about studies involving human research participants and Sex and Gender in Research.

| Reporting on sex and gender | N/A |
| --- | --- |
| Population characteristics | N/A |
| Recruitment | N/A |
| Ethics oversight | N/A |

Note that full information on the approval of the study protocol must also be provided in the manuscript.

# Field-specific reporting

Please select the one below that is the best fit for your research. If you are not sure, read the appropriate sections before making your selection.

☒ Life sciences       ☐ Behavioural & social sciences       ☐ Ecological, evolutionary & environmental sciences

For a reference copy of the document with all sections, see nature.com/documents/nr-reporting-summary-flat.pdf

# Life sciences study design

All studies must disclose on these points even when the disclosure is negative.

| Sample size | 34 LSFM mouse brains imaged in 647 channel with 25 manually annotated cubes were used for the evaluation of whole-brain axon segmentation. 54 LSFM brains imaged in 488 channel with 12 brains, 30302 image slices containing manual annotated brain regions were used for the evaluation of brain region segmentation and whole-brain registration. This is so far all the diverse data collected and manual annotated in our project. It's sufficient to lead to a determination that systematic errors surpassed statistical errors. |
| --- | --- |
| Data exclusions | No data were excluded from the analyses. |
| Replication | We successfully tested against 25 cubes on axon segmentation, 12 brains on brain style transfer, brain region segmentation, and whole-brain registration. |
| Randomization | Randomization was not performed as multiple experimental groups across biological samples were not used in this study. |
| Blinding | All annotators worked in isolation and were blinded to group allocation. |

# Reporting for specific materials, systems and methods

We require information from authors about some types of materials, experimental systems and methods used in many studies. Here, indicate whether each material, system or method listed is relevant to your study. If you are not sure if a list item applies to your research, read the appropriate section before selecting a response.

## Materials & experimental systems

| n/a | Involved in the study |
|---|---|
| ☐ | ☒ Antibodies |
| ☒ | ☐ Eukaryotic cell lines |
| ☒ | ☐ Palaeontology and archaeology |
| ☐ | ☒ Animals and other organisms |
| ☒ | ☐ Clinical data |
| ☒ | ☐ Dual use research of concern |

## Methods

| n/a | Involved in the study |
|---|---|
| ☒ | ☐ ChIP-seq |
| ☒ | ☐ Flow cytometry |
| ☒ | ☐ MRI-based neuroimaging |

## Antibodies

| | |
|---|---|
| Antibodies used | Rabbit anti-Fos (Synaptic Systems, CAT 226003, lot 9-95; dilution: 1:500 for iDISCO), rabbit anti-RFP (Rockland, 600-401-379, lot 42896, dilution:1:500), donkey anti-rabbit AlexaFluor 647 (Jackson ImmunoResearch, 711-605-152, lot: 161533; dilution: 1:500) |
| Validation | Fos: synthetic peptide corresponding to rat Fos AA2-17, website states specific to Fos, validated for immunohistochemistry in mouse tissue on the website and in several publications e.g. DeNardo et al. Nature Neuroscience, 2019. RFP: website states antibody is expected to cross-react with RFP variants (e.g. mCherry, tdTomato) and has been validated for immunohistochemistry in mouse in several publications (e.g. Crowther et al, Stem Cell Reports, 2018) and for iDISCO in mouse tissue (DeNardo et al. Nature Neuroscience, 2019) |

## Animals and other research organisms

Policy information about studies involving animals; ARRIVE guidelines recommended for reporting animal research, and Sex and Gender in Research

| | |
|---|---|
| Laboratory animals | TRAP2;Ai14 mice (TRAP2, JAX 03032; Ai14, JAX 7914) were kindly shared by Liqun Luo, Stanford University, CA, US.  DAT-Cre (JAX 006660) and Vgat-Cre mice (JAX 028862) were obtained from the Jackson Laboratory. Sert-Cre mice (031028-UCD) were obtained from the University of California, Davis. All the mice used in this study are adult mice (8-16 weeks old). |
| Wild animals | This study did not involve wild animals. |
| Reporting on sex | The sex of each mouse is irrelevant to our study, which focused on algorithm-development for whole-brain 3D image analysis. |
| Field-collected samples | This study did not involve field-collected samples |
| Ethics oversight | All experiments related to the use of mice in the Medical Research Council Laboratory of Molecular Biology (MRC LMB) were carried out in accordance with the UK Animals (Scientific Procedures) Act of 1986, with local ethical approval provided by the MRC LMB Animal Welfare Ethical Review Board (LMB AWERB) and overseen institutionally by designated animal welfare officers (NACWOs). Animal Project Licence (PPL) is PP6471806. All experiments related to the use of mice at the National Institute of Biological Sciences, Beijing (NIBS) were approved by the Animal Care and Use Committee of NIBS in accordance with the Regulations for the Administration of Affairs Concerning Experimental Animals of China. |

Note that full information on the approval of the study protocol must also be provided in the manuscript.

