## [Peer Review File · Nature Methods]

Peer Review Information

Manuscript Title: D-LMBmap: A fully automated deep learning pipeline for whole-brain profiling of neural circuitry

Corresponding author name(s): Jing Ren

Editorial Notes: n/a

Reviewer Comments & Decisions:

Decision Letter, initial version:

Dear Jing,

Thank you for your letter detailing how you would respond to the reviewer concerns regarding your Article, "D-LMBmap: A fully automated deep learning pipeline for whole-brain profiling of neural circuitry". We have decided to invite you to revise your manuscript as you have outlined, before we reach a final decision on publication.

Please make sure documented code and test data are available for referee testing upon revision.

- * include a point-by-point response to the reviewers and to any editorial suggestions
- * please underline/highlight any additions to the text or areas with other significant changes to facilitate review of the revised manuscript
- * address the points listed described below to conform to our open science requirements

* ensure it complies with our general format requirements as set out in our guide to authors at www.nature.com/naturemethods

* resubmit all the necessary files electronically by using the link below to access your home page

[Redacted] This URL links to your confidential home page and associated information about manuscripts you may have submitted, or that you are reviewing for us. If you wish to forward this email to co-authors, please delete the link to your homepage.

We hope to receive your revised paper within three months. If you cannot send it within this time, please let us know. In this event, we will still be happy to reconsider your paper at a later date so long as nothing similar has been accepted for publication at Nature Methods or published elsewhere.

OPEN SCIENCE REQUIREMENTS

REPORTING SUMMARY AND EDITORIAL POLICY CHECKLISTS

DATA AVAILABILITY

Please include a “Data availability” subsection in the Online Methods. This section should inform readers about the availability of the data used to support the conclusions of your study, including accession codes to public repositories, references to source data that may be published alongside the paper, unique identifiers such as URLs to data repository entries, or data set DOIs, and any other statement about data availability. At a minimum, you should include the following statement: “The data that support the findings of this study are available from the corresponding author upon request”, describing which data is available upon request and mentioning any restrictions on availability. If DOIs are provided, please include these in the Reference list (authors, title, publisher (repository name), identifier, year). For more guidance on how to write this section please see: <http://www.nature.com/authors/policies/data/data-availability-statements-data-citations.pdf>

CODE AVAILABILITY

Please include a “Code Availability” subsection in the Online Methods which details how your custom code is made available. Only in rare cases (where code is not central to the main conclusions of the paper) is the statement “available upon request” allowed (and reasons should be specified).

MATERIALS AVAILABILITY

ORCID

Sincerely,
Rita

Rita Strack, Ph.D.
Senior Editor
Nature Methods

Reviewers' Comments:

Reviewer #1:

Remarks to the Author:

In this manuscript, Li et al. report a fully automated deep learning pipeline called D-LMBmap, which is used for computational processing of light sheet-based 3D microscopic images and generating whole-brain axonal projection maps. This work breaks new ground and reports an impressive single workflow to achieve accurate brain registration, structural segmentation, axonal segmentation, and quantification. The authors present solid and convincing evidence demonstrating the advantages of D-LMBmap compared to several other current brain mapping tools, such as SeBRe, BIRDS, mBrainAligner, etc. Based on my experience of brain mapping for over a decade, the D-LMPmap may be one of the most powerful informatics tools to accelerate the generation of a whole-brain 3D connectome. Because lightsheet and other 3D microscopic imaging technologies have been adopted by numerous laboratories, the D-LMPmap will be an extremely valuable tool for these labs to map anatomical and

behavioral data. Overall, the manuscript is very well written and figures carefully constructed. With all of these considerations, I strongly support its publication in Nature Methods.

I have only one major comment about the registration component. I am very impressed by the registration accuracy as described in the manuscript. However, based on my personal experience and knowledge, all current automated registration algorithms (including mBrainAligner described in the manuscript and the Allen Institute's registration algorithm) suffer a major shortcoming. Their registration for large brain structures, such as the cerebral cortex, hippocampus, striatum, etc, is reasonable, but their accuracy for registration of smaller structures, such as individual thalamic or hypothalamic nuclei, is not ideal. This issue has been a long-standing problem affecting accuracy of large-scale anatomical data annotation and analysis. In comparison with classic neuroanatomical studies using Nissl or other cytoarchitectonic features for registration, all of these automated registration programs, including Allen CCF3, mBrainAligner and the D-LMBmap described here, use autofluorescence channel for registration. In my view, without cytoarchitectonic features, the registration accuracy for small structures will not be sufficient. I hope the authors can address this question in their revision. For example, in Figure 5, can the authors provide ground truth to demonstrate registration accuracy for small structures in thalamus, hypothalamus, and brainstem?

Additionally, because I anticipate that this program will be adopted by numerous labs, please make sure the code deposited in Github is useable.

Reviewer #2:

Remarks to the Author:

Zhongyu Li et al. developed D-LMBmap, an end-to-end package providing an integrated workflow containing three learning-based modules for whole-brain connectivity mapping. This paper is refreshing for researchers in this field and provides a new tool for projection connectivity research. However, as described in the paper, most of them are the transfer and application of existing AI methods, which lack enough originality. It would be a good job if the author could continue to polish the paper and add more rich results to demonstrate the practicability of the method.

1. It is suggested that authors adopt more careful and rigorous comments on some "first" or similar statements in the manuscript.

i. The authors claimed that "To the best of our knowledge, this is the first learning-based whole mouse brain registration framework which can achieve rigid, affine and deformation transformation in an end-to-end deep neural network." [Line 378-380] Different groups have studied learning-based methods for whole brain registration. The following literature is recommended :

☐ Qu, L., Li, Y., Xie, P. et al. Cross-modal coherent registration of whole mouse brains. Nat Methods 19, 111–118 (2022). <https://doi.org/10.1038/s41592-021-01334-w>

☐ Ni, H., Feng, Z., Guan, Y. et al. DeepMapi: a Fully Automatic Registration Method for Mesoscopic Optical Brain Images Using Convolutional Neural Networks. *Neuroinform* 19, 267–284 (2021). <https://doi.org/10.1007/s12021-020-09483-7>

ii. The authors claimed that “current 3D brain registration methods heavily rely on the brain outline without a specific focus on the alignment of individual brain regions”. [Line 69-71] Brain region registration technology has been developed for many years, and it has long been a consensus in the field that only the outer contour of the brain region can not achieve accurate registration. So, there are a lot of approaches that are focused on the registration of individual brain areas.

☐ Qu, L., Li, Y., Xie, P. et al. Cross-modal coherent registration of whole mouse brains. *Nat Methods* 19, 111–118 (2022).

☐ Ni, H., Tan, C., Feng, Z. et al. A Robust Image Registration Interface for Large Volume Brain Atlas. *Sci Rep* 10, 2139 (2020).

☐ Jiang, X., Ma, J., Xiao, G., et al. A review of multimodal image matching: Methods and applications, *Information Fusion*, 73, 22-71 (2021).

iii. In Line 105, “The three main modules—axon segmentation, brain region segmentation, and whole brain registration—are based on novel deep-learning neural network algorithms”. Does the term “Novel” refer to the author’s novel approach, or does it mean the introduction of the latest algorithm? If it is the latter, it is recommended to add relevant references. If it is the former, the idea of using automatically generated labels to train neural networks for axon segmentation has been previously reported. The segmentation framework used by the author is based on the existing nn-Unet without any innovation or improvement in the architecture. Therefore, this module is not innovative for deep learning algorithms.

☐ Huang Q, Chen Y, Liu S, et al. Weakly supervised learning of 3d deep network for neuron reconstruction[J]. *Frontiers in Neuroanatomy*, 2020, 14: 38.

☐ Chen W, Liu M, Du H, et al. Deep-Learning-Based Automated Neuron Reconstruction From 3D Microscopy Images Using Synthetic Training Images[J]. *IEEE Transactions on Medical Imaging*, 2021, 41(5): 1031-1042.

☐ Chen X, Zhang C, Zhao J, et al. Weakly Supervised Neuron Reconstruction From Optical Microscopy Images With Morphological Priors[J]. *IEEE Transactions on Medical Imaging*, 2021, 40(11): 3205-3216.

2. Axon segmentation

a) Is the purpose of axon segmentation for projection axon density calculation or further axon tracing? At present, due to the rapid development of high-resolution whole-brain imaging technology, the research of neuroscience has not only focused on finding out which are axon signals (segmentation), but also needs to skeletonize them, judge their connection relationship, and finally realize the axon tracing. In this case, segmentation is the easiest first step. In particular, the latest HD-fMOST (Zhong et al. 2021, *Nat Methods*) had shown that a large range of complex neural fibers was accurately identified by

traditional algorithms due to the high-definition image quality. As the most difficult analytical task, axon tracing has not yet been solved, and it is also the focus of most current axon analysis methods.

b) The axons are distributed in the whole brain. The labeling effect of virus labels, autofluorescence background of local brain tissue, and distribution characteristics of axons themselves may affect the quality and features of the final axon images. Can the expanded training set in this paper cover all the axon features in the whole brain? Can simple axon training sets be used to achieve accurate segmentation of complex and dense axons? -- This will affect the application of the method, and I suggest that the author add relevant validation experiments to demonstrate it.

c) It is recommended that the author supplement the evaluation of the integrity of the segmented axon. The Dice coefficient in Fig2 was only about 0.7, indicating that the degree of signal extraction was limited. Although cIDice reached 0.9, as an index to measure topological similarity, it is not appropriate to use cIDice for evaluation on the premise that the integrity of extracted axons cannot be guaranteed. In addition, according to "the quantitative evaluation of six sample cubes" [Line 200-202], the precision rate was 0.8 and the Dice coefficient was 0.7, then the recall rate can be calculated as only about 0.6. Does it mean that about 40% of the axons were not recognized?

d) The main difficulty in axon segmentation or identification is the low signal intensity and the signal-to-noise ratio of the target object, which may further lead to specific problems such as discontinuity of the segmented axon. I suggest adding enlarged views to show the segmented axon details. Extended Data Movie 2 appears to show some unrecognized axons.

e) The Dice index and Precision index of this paper are far better than the TrailMap method. TrailMap used expert annotated data to conduct model training and prediction based on, while D-LMBMap used automatically generated training data. I suggest that the authors supplement the corresponding ablation experiments to reveal the mechanism behind the improvement.

f) In extended data table 1, the author gives the data set used for axon segmentation. There are 3 data sets with low resolution, it is difficult to ensure that a single axon can be distinguished -- does it limit the application scenarios of the method? Adding a discussion about it will be good.

g) During the automatic generation of training samples, about 40 axon cubes were selected for some data and about 90 for others. In addition, the number of artifact blocks is also not fixed. How to determine these numbers? Is it a random selection within the whole brain? If not, on what basis?

3. Brain region segmentation and registration

a) The idea of style transfer is very clever, and it is indeed promising to solve the problem due to the big style difference between the data to be registered and the reference map like CCF, which affects the

registration quality. As stated in the paper, the premise of the smooth application is that "in which the image appearance is converted into a reference image style without losing its original content" [Line 226-227]. However, it can be seen from Extended data fig.4 that the structure information and gray value information of the transferred brain and the original brain is very different. The gray value difference in the original brain has specific biological significance, while some gray values were directly reversed after the transfer. Is there a biological or mathematical explanation behind this mapping mechanism, and how to verify it without losing the original content?

b) Before registration, the author down-sampled the whole brain data set to 320*456*528 pixels (extended data table 4), which belonged to the test data with low resolution. Allen CCF currently provides the data with 10-micron resolution. I wonder how well the method has been tested on this level of data.

c) As far as we know, it works well to calculate Dice and so on for large brain regions. Can you supplement registration results of smaller nuclei (lateral habenular, etc.), and even consider using reference sites to calculate distance deviations directly?

4. Software

a) The software is difficult to install and lacks dependency packages. Could you please provide more detailed instructions for the convenience of users?

b) The software is not stable enough and sometimes encounters crashes. It is suggested to further optimize the software for better promotion and use.

Reviewer #3:

Remarks to the Author:

A. The authors present an integrated pipeline for automated analysis of mesoscale projection mapping from whole-brain light microscopic imaging data of different modalities. The work is based on putting together deep net based modules for axon segmentation and brain/atlas registration.

B. The primary claim to originality is that this is the first time such an integrated pipeline has been presented, and also that individual modules in this pipeline are new. This claim is difficult to sustain.

First, the primary focus of the paper is on the brain/atlas registration problem, but the methods presented by the authors are neither of very high quality (they only segment the brain into 6 compartments - compare this with the 500-1000 compartments in current mouse brain atlases). Leaving aside the comparisons with other methods, such a segmentation is of very little value for modern

mesoscale projection analysis from a biological significance perspective. Besides, the multimodal/cross contrast brain/atlas registration problem addressed by the authors through the style-transfer and GANS approaches has already been addressed satisfactorily in previous publications (e.g. both learned contrast maps and treatment of damaged/distorted sections of the brain are addressed in https://doi.org/10.1007/978-3-030-33226-6_18).

Second, segmentation of axons in brain images is a large subject and goes well beyond the few references considered by the authors (consider the DIADEM challenge and many associated papers). Even in the context of mesoscale projection mapping there have been multiple relevant works, including the original AIBS publications on mesoscale connectivity maps of the mouse brain. The authors posit that the primary difficulty lies in the laborious process of manually segmenting of proofreading individual axons under a sufficient variety of circumstances. This is true, but the solution offered by the authors (reliance on human labeling of image cubes containing/not containing axons, followed by a set of classical machine vision approaches including thresholding, morphological operations etc) fails to reach the biologically relevant bar of providing ground truth guarantees on the individually traced neurite fragments. While the specific technique presented by the authors for axon segmentation may or may not correspond to previous approaches (it is difficult to judge because the authors do not comprehensively review this large literature), there is no demonstration of biological ground truthing (which necessarily requires a validation set of manually labelled axons across a spectrum of brain regions). Thus it is difficult to conclude that there has been a true methodological advance here, as opposed to yet another technique added to a large variety of techniques already brought to bear on this problem.

Third the authors do not show recognition of one of the primary issues that plague the analysis of mesoscale projections, namely the analysis of the injection sites themselves. The key problem with this kind of projection mapping and associated analysis is that injections are difficult to control (thus they have widely varying sizes and seldom cover the brain in a uniform manner), thus the projection maps derived from such injections have necessarily to be subjected to careful deconvolution analysis. This is still an open problem without satisfactory solutions, perhaps only to be addressed by combining with single-neuron data (of which there is a growing volume today). Genetic constructs seldom label a spatially well-localized set of somata, so any automated pipeline for mesoscale projection analysis worth its salt must address thorny issue. The manuscript shows no evidence that this has been considered.

As such, the novelty of both the modules and the integrated approach is modest, and of uncertain biological relevance. If the authors consider that the software pipeline integration is the key story here this paper might better be directed towards a software journal.

Author Rebuttal to Initial comments

NMETH-A50696A

Response to Reviewers

We thank reviewers for their appreciation about the significance and quality of our work, and for their helpful suggestions. We have performed a number of new experiments, data analyses, and amended the manuscript to address their critiques. As a result, the paper and the whole D-LMBmap pipeline have been improved beyond the original submission.

All revised sections in the text are indicated by red-coloured font. The detailed point-by-point responses are provided below (*reviewers' critiques are copied in blue*).

Reviewers' Comments:

Reviewer #1:

Remarks to the Author:

In this manuscript, Li et al. report a fully automated deep learning pipeline called D-LMBmap, which is used for computational processing of light sheet-based 3D microscopic images and generating whole-brain axonal projection maps. This work breaks new ground and reports an impressive single workflow to achieve accurate brain registration, structural segmentation, axonal segmentation, and quantification. The authors present solid and convincing evidence demonstrating the advantages of D-LMBmap compared to several other current brain mapping tools, such as SeBRe, BIRDS, mBrainAligner, etc. Based on my experience of brain mapping for over a decade, the D-LMPmap may be one of the most powerful informatics tools to accelerate the generation of a whole-brain 3D connectome. Because lightsheet and other 3D microscopic imaging technologies have been adopted by numerous laboratories, the D-LMPmap will be an extremely valuable tool for these labs to map anatomical and behavioral data. Overall, the manuscript is very well written and figures carefully constructed. With all of these considerations, I strongly support its publication in Nature Methods.

I have only one major comment about the registration component. I am very impressed by the registration accuracy as described in the manuscript. However, based on my personal experience and knowledge, all current automated registration algorithms (including mBrainAligner described in the manuscript and the Allen Institute's registration algorithm) suffer a major shortcoming. Their registration for large brain structures, such as the cerebral cortex, hippocampus, striatum, etc, is reasonable, but their accuracy for registration of smaller structures, such as individual thalamic or hypothalamic nuclei, is not ideal. This issue has been a long-standing problem affecting accuracy of large-scale anatomical data annotation and analysis. In comparison with classic neuroanatomical studies using Nissl or other cytoarchitectonic features for registration, all of these automated registration programs, including Allen CCF3, mBrainAligner and the D-LMBmap described here, use autofluorescence channel for registration. In my view, without cytoarchitectonic features, the registration accuracy for small structures will not be sufficient. I hope the authors can address this question in their revision. For example, in Figure 5, can the authors provide ground truth to demonstrate registration accuracy for small structures in thalamus, hypothalamus, and brainstem?

10

NMETH-A50696A

Response to Reviewers

We sincerely thank the reviewer for his/her great appreciation of the novelty, significance and quality of our study, and their helpful suggestions.

We agree with the reviewer that accurate registration of small brain regions is challenging and there is great room for improvement on this issue. Indeed, for many small brain structures, images taken by autofluorescence channels without cytoarchitectonic labelling may not provide sufficient information because the anatomical definitions of these structures are based on their cytoarchitectonic features. Nevertheless, we took on this challenge and first tested how well D-LMBmap performs on the whole brain registration of small brain structures that have detectable features under autofluorescence imaging. We generated the ground truth of small region registration by manually annotating several small brain structures in the thalamus, hypothalamus, and brainstem to facilitate the validation, including habenular (Hb), mammillothalamic tract (mtt), anterior commissure temporal limb (act), fasciculus retroflexus (fr) and interpeduncular nucleus (IPN).

We generated the quantitative whole-brain registration results of these five small brain regions using the automated pipeline in D-LMBmap, and compared them with related registration solutions. D-LMBmap can outperform other methods in all five small brain regions. As presented in the new Figure 4D, D-LMBmap achieves a median Dice score from 0.60 to 0.85 for individual small brain regions, and an average median Dice score of 0.76. Compared with other methods, D-LMBmap achieves about 35% higher Dice score in small brain region registration.

Next, we asked, when doing small structure registration, how close D-LMBmap automated pipeline was to achieving results comparable to its pipeline that incorporated training data from manual segmentation (Extended Data Fig. 16B). As presented in the new Extended Data Fig. 16C, when trained with data generated by user delineation, D-LMBmap achieves a median Dice score from 0.80 to 0.94 for individual small brain regions, and an average median Dice score of 0.86. This data demonstrates that users can also further improve the results by using manually segmented regions of interest for training constraints. More importantly, unlike existed method using manual delineated brain regions directly on the experimental (testing) brains¹, once D-LMBmap finished the training of the deep registration model from the manually delineated brain regions, it can still achieve accurate small structure whole-brain registration on the testing brains without further manual inputs (Figure R1). Thus, D-LMBmap stands out as the optimal method for registering small brain structures with detectable features under the autofluorescence channel.

But for small brain structures without detectable features under autofluorescence channel, it is extremely difficult, if not impossible to validate their registration results simply because even experts cannot generate manually annotated ground truth without visible features. And as the reviewer pointed out, this is the limitation faced by all the registration methods using autofluorescence channels. However, our multi-constraint strategy achieves great multi-regional alignment optimization, which was shown in the Extended Data Fig. 17A, that adding individual regional constraints enhances the registration of other major brain regions. Due to its ability to achieve robust

11

NMETH-A50696A

Response to Reviewers

deformation optimization and multi-regional alignment optimization, we propose that D-LMBmap can enhance the registration results of small brain structures without detectable features by leveraging regional constraints from adjacent structures with detectable features or user input based on other channels. We tested this hypothesis by generating a confusion matrix, which is presented in the new Extended Data Fig. 17B, showing that adding individual constraints of small brain structures greatly improves the registration of other small brain structures.

In summary, our newly added data demonstrate that D-LMBmap is the best method for accurately and efficiently registering small brain structures with detectable features under the autofluorescence channel. D-LMBmap also provides a new strategy and the pipeline for optimising the accuracy of small brain structures without detectable features under the autofluorescence. Besides our automated pipeline, users could also either add a scant amount of manually segmented constraints or constraints from channels with cytoarchitectonic features during training to generate whole-brain registration of small structures more precisely.

Figure R1. Schematic drawings of the pipelines employed in the D-LMBmap whole-brain registration based on brain regions constraints and other methods.

A. In the training pipeline of D-LMBmap for whole-brain registration, brain region constraints can either be automatically obtained using our developed

12

NMETH-A50696A

Response to Reviewers

Multi-view Semi-CEA network for automated brain region segmentation or manually delineated by users. These brain region constraints are then used as input for the training of the whole-brain registration model. **B.** In the testing pipeline of D-LMBmap for whole-brain registration, the brain regions of the testing brain can be obtained either automatically using the Multi-view Semi-CEA network or manually delineated by users. Once this is done, registration can be automatically achieved using the trained whole-brain registration deep model. **C.** The testing pipeline used by other methods requiring manual delineation.

Additionally, because I anticipate that this program will be adopted by numerous labs, please make sure the code deposited in Github is useable.

We sincerely thank the reviewer for his/her great appreciation of our software. We have invited 12 researchers to test the updated software for ease of use. Of this group, 8 were running the software on Windows 10 X64 and the remaining 4 were using Windows 11 X64. Neither system tested had prior installation of dependency packages. As a result, we have implemented several improvements to expediate the installation process. For example, the software can now be installed directly by double clicking the "D-LMBmap.exe" file.

In addition, we have also provided more detailed instructions consisting of documents, a tutorial movie and example data, accessible from our project's Github page (Link: <https://github.com/lmbneuron/D-LMBmap>). Users can follow the movie as a guide to achieve each module in whole brain projection mapping. Furthermore, on our Github page we have provided our code with detailed annotations consisting of three files: 'axon segmentation', 'brain region segmentation' and 'whole brain registration'. By running this code, users can train their own data and accomplish whole-brain projection mapping.

Reviewer #2:

Remarks to the Author:

Zhongyu Li et al. developed D-LMBmap, an end-to-end package providing an integrated workflow containing three learning-based modules for whole-brain connectivity mapping. This paper is refreshing for researchers in this field and provides a new tool for projection connectivity research. However, as described in the paper, most of them are the transfer and application of existing AI methods, which lack enough originality. It would be a good job if the author could continue to polish the paper and add more rich results to demonstrate the practicability of the method.

We thank the reviewer for his/her positive comments. We have polished the paper and added new results to address each concern. Our responses to the specific comments are provided below.

1. It is suggested that authors adopt more careful and rigorous comments on some "first" or similar statements in the manuscript.

i. The authors claimed that "To the best of our knowledge, this is the first learning-based whole mouse brain registration framework which can achieve

13

NMETH-A50696A

Response to Reviewers

rigid, affine and deformation transformation in an end-to-end deep neural network. [Line 378-380] Different groups have studied learning-based methods for whole brain registration. The following literature is recommended :

Qu, L., Li, Y., Xie, P. et al. Cross-modal coherent registration of whole mouse brains. *Nat Methods* 19, 111–118 (2022). <https://doi.org/10.1038/s41592-021-01334-w>

Ni, H., Feng, Z., Guan, Y. et al. DeepMapi: a Fully Automatic Registration Method for Mesoscopic Optical Brain Images Using Convolutional Neural Networks. *Neuroinform* 19, 267–284 (2021). <https://doi.org/10.1007/s12021-020-09483-7>

We thank the reviewer's advice revised the sentence as follows: "*With this learning-based whole mouse brain registration framework, D-LMBmap achieves rigid, affine and deformation transformation in a comprehensive deep neural network.*" [Line 408-410]

Thanks very much for the references suggested by the reviewer. We incorporated them into the "Discussion" [Line444-464]. Meanwhile, we would also like to clarify that mBrainAligner developed by Qu et al applied the learning-based method in 3D segmentation of main brain regions but not whole brain registration². Their registration step was achieved by landmark mapping and optimization. Additionally, DeepMapi developed by Ni et al relies on manual reference image assignment due to the difference between the experimental brains and the brain atlas³. In comparison, our pipeline can achieve accurate registration in an end-to-end deep neural network automatically, even for brains from different modalities, without manual inputs.

We also found a mistake we made when we cited the paper by Qu et al, thanks to this comment made by the reviewer (should be "Qu et al" instead of "Peng et al"), and have it corrected in the revised manuscript.

ii. The authors claimed that "current 3D brain registration methods heavily rely on the brain outline without a specific focus on the alignment of individual brain regions". [Line 69-71] Brain region registration technology has been developed for many years, and it has long been a consensus in the field that only the outer contour of the brain region can not achieve accurate registration. So, there are a lot of approaches that are focused on the registration of individual brain areas. Qu, L., Li, Y., Xie, P. et al. Cross-modal coherent registration of whole mouse brains. *Nat Methods* 19, 111–118 (2022).
Ni, H., Tan, C., Feng, Z. et al. A Robust Image Registration Interface for Large Volume Brain Atlas. *Sci Rep* 10, 2139 (2020).
Jiang, X., Ma, J., Xiao, G., et al. A review of multimodal image matching: Methods and applications, *Information Fusion*, 73, 22-71 (2021).

Thank you for the suggestion. We followed the reviewer's advice and revised the sentence as "*Secondly, the existing 3D brain registration methods primarily rely on either whole-brain intensity or specified brain regions to carry out the registration, which cannot coordinate multi-regional alignment optimisation and whole brain registration.*" [Line 69-72]. Here we mean to emphasize that, unlike existing methods, ours embeds brain regions in a learning-based registration

NMETH-A50696A

Response to Reviewers

framework and generates multiple brain region constraints that optimize both individual brain regions and whole brain registration simultaneously.

We thank the reviewer for the suggested reference. Qu et al used brain region segmentation results for computing landmarks². Ni et al performed brain region segmentation manually by Amira and processed whole brain registration by SyN separately¹. Jiang et al reviewed general image registration methods, some of which used intensity for image registration but didn't generate brain region constraints⁴. We recognised prior achievements in the field of brain region registration and incorporate comparisons with these methods in the "Introduction" [Line 56-62] and also added them in "Discussion" during revision [Line 443-464].

iii. In Line 105, "The three main modules—axon segmentation, brain region segmentation, and whole brain registration—are based on novel deep-learning neural network algorithms". Does the term "Novel" refer to the author's novel approach, or does it mean the introduction of the latest algorithm? If it is the latter, it is recommended to add relevant references. If it is the former, the idea of using automatically generated labels to train neural networks for axon segmentation has been previously reported. The segmentation framework used by the author is based on the existing nn-Unet without any innovation or improvement in the architecture. Therefore, this module is not innovative for deep learning algorithms.

Huang Q, Chen Y, Liu S, et al. Weakly supervised learning of 3d deep network for neuron reconstruction[J]. Frontiers in Neuroanatomy, 2020, 14: 38.

Chen W, Liu M, Du H, et al. Deep-Learning-Based Automated Neuron Reconstruction From 3D Microscopy Images Using Synthetic Training Images[J]. IEEE Transactions on Medical Imaging, 2021, 41(5): 1031-1042.

Chen X, Zhang C, Zhao J, et al. Weakly Supervised Neuron Reconstruction From Optical Microscopy Images With Morphological Priors[J]. IEEE Transactions on Medical Imaging, 2021, 40(11): 3205-3216.

Thanks for the suggestion. We revised the sentence as "*The three main modules—axon segmentation, brain region segmentation, and whole brain registration—are based on advanced deep-learning neural network algorithms*" [Line 108-109].

We would like to further clarify that even though nnU-Net, CycleGAN and VoxelMorph are existing deep learning algorithms, in order to adapt them for whole brain projectome mapping, we developed new network structures, modules and frameworks to achieve automated processing with minimum labour input so that it can be easily adopted by various labs to accelerate the mesoscale connectomic mapping at the whole brain level.

For example, as the reviewer has attentively pointed out in the comments listed under "3. Brain region segmentation and registration", this is the first time style transfer is proposed for brain region registration and CycleGAN is applied for this function. When we directly applied CycleGAN for brain style transfer, the outlines of the style transferred brain and the original brain cannot be aligned. Therefore, we designed a brain outline segmentation sub-network (i.e., CEA-Net) to integrate with the CycleGAN model. This integrated module can simultaneously achieve automated brain outline

15

NMETH-A50696A

Response to Reviewers

segmentation, brain style transfer and brain outline consistency. Even though style transfer method employed in D-LMBmap is based on CycleGAN, as shown in the newly added Extended Data Fig. 8A and Extended Data Fig. 14A, our improved style transfer method outperforms all other method at both the whole-brain and major brain regions level. Ablation studies also showed that the proposed Multi-view Semi-CEA is essential for the improvement of automated brain region segmentation (Extended Data Fig. 14B).

We share the reviewer's enthusiasm and believe that this strategy can tackle the brain style transfer problem.

We employed VoxelMorph as our basic AI model in whole-brain registration. However, the original VoxelMorph algorithm is not only incapable of handling multi-modality brain data, but also failed to coordinate the registration of individual brain regions and whole brain alignment. To solve these issues, we first extended the VoxelMorph model with multi-constraints to include constraints generated by each of the six major brain regions and the style transferred brain in different channels and final loss functions. The brain region constraints can maintain the whole-brain registration consistency in each major brain region, and the style transferred brain constraints can ensure the whole-brain registration consistency between two brains in different modalities. Furthermore, we extended the VoxelMorph model in multi-scale. The deep model was trained with images from low-resolution to high-resolution, so that the learning-based 3D registration can process big whole-brain data.

Indeed, the three studies suggested by the reviewer employed automatically generated labels to train neural networks for axon segmentation. However, these studies focused on the single neuron tracing/reconstruction in high-resolution images (whole brain size of 20,000×30,000×25,000) generated by block-face imaging (e.g. fMOST, two-photon or confocal microscopy). Conversely, our strategy aims to map bulk tracing data generated by light-sheet microscopy (whole brain size of 2,000×2,500×2,000). We expressed our appreciation for the achievements in the axon segmentation methods applied in block-face imaging and incorporate discussions in our revised manuscript [Line 501-513].

We developed automated annotation, data augmentation, nnU-Net training, and whole-brain prediction modules in our axon segmentation pipeline. The automated annotation module is specifically designed to eliminate laborious manual annotation in axon segmentation, which is a practical and challenging problem for existing solutions (e.g. TrailMap). The data augmentation module is specifically designed to alleviate the limitations of training datasets in the automated annotation module. In the nnU-Net training step, we added the axial attention module to improve feature learning of tree-topological axons. Due to these distinctive designs, we can effectively achieve annotation-free mesoscale whole-brain axon segmentation.

To further validate the effectiveness of our modules, we added new ablation experiments in the revised manuscript. As presented in the newly added Extended Data Fig. 6, even without data augmentation and axial attention, D-LMBmap is more effective than Trailmap. Nevertheless, our data

16

NMETH-A50696A

Response to Reviewers

augmentation and axial attention are essential for the superior performance of our automated annotation and axon segmentation solution and can further markedly improve the results and achieved a CIDice score around 0.9.

The concerns regarding the difference between block-face imaging/single neuron tracing (e.g. fMOST) and light-sheet imaging/bulk tracing are further addressed under the comments listed in "2. Axon segmentation".

2. Axon segmentation

a) Is the purpose of axon segmentation for projection axon density calculation or further axon tracing? At present, due to the rapid development of high-resolution whole-brain imaging technology, the research of neuroscience has not only focused on finding out which are axon signals (segmentation), but also needs to skeletonize them, judge their connection relationship, and finally realize the axon tracing. In this case, segmentation is the easiest first step. In particular, the latest HD-fMOST (Zhong et al. 2021, Nat Methods) had shown that a large range of complex neural fibers was accurately identified by traditional algorithms due to the high-definition image quality. As the most difficult analytical task, axon tracing has not yet been solved, and it is also the focus of most current axon analysis methods.

We agree with the reviewer that the rapid development of whole-brain imaging technology is pushing the limit for judging connection relationship and fine axon tracing. HD-fMOST is a great example for the block-face imaging/single neuron tracing technology which has advanced our knowledge on mapping brain-wide connectivity with fine-scale spatial organization information. However, there is still a steep tradeoff between throughput and resolution in anatomical approaches to mapping long-range connections. Due to the extensive intertwining of neural processes, to achieve long-range projection axonal tracing, studies using block-face imaging strategy usually use sparse labelling and reconstruct fewer than 40 neurons per brain^{5,6}. The combination of tissue-clearing and block-face imaging can handle maximumly up to 180 neurons per brain, and 25 brains were used to reconstruct 1000 neurons in the study of Winnubst et al⁷.

Here, we use serotonin neurons as an example to address the limitation of the throughput generated by block-face imaging. There are roughly ~ 20,000 serotonin neurons projecting to the entire forebrain in a mouse brain. These neurons are highly heterogenous, containing different sub-populations defined by the projection sites and various collateralization pattern, which was revealed by LSF⁸. In our previous study, sparsely labelled serotonin neurons were traced and reconstructed by the fMOST pipeline and the results confirmed the subpopulations we revealed by using bulk-tracing imaging⁹. Fine-scale spatial organization and further detailed heterogeneity were uncovered by single-neuron tracing, but the cell number is insufficient for analyzing the statistics of their branching patterns across multiple target regions.

Block-face imaging requires specialized instruments that may not be readily accessible to many researchers. As a result, there are further limitations to the accumulation of cell numbers for a specific neuronal type across different labs.

17

NMETH-A50696A

Response to Reviewers

Traditional bulk-tracing has been criticized for sampling the aggregate architecture of neurons at the injection site and missing the diversity of projection patterns. However, modern bulk-tracing data produced by viral-genetic strategy and imaged by light-sheet at the whole brain level can reveal the connection relationship of a very specific neuronal type defined by multiple molecular features as well as other anatomical information⁸⁻¹⁰.

We did not aim for single axon tracing because of the densely labelled data by bulk-tracing and the limited imaging resolution produced by light-sheet microscopy. Nevertheless, our axon segmentation pipeline can achieve not only axon density calculation but also segmentation, skeleton, and connection relationship judgement efficiently.

b) The axons are distributed in the whole brain. The labeling effect of virus labels, autofluorescence background of local brain tissue, and distribution characteristics of axons themselves may affect the quality and features of the final axon images. Can the expanded training set in this paper cover all the axon features in the whole brain? Can simple axon training sets be used to achieve accurate segmentation of complex and dense axons? – This will affect the application of the method, and I suggest that the author add relevant validation experiments to demonstrate it.

We agree with the reviewer that the complexity of the axons, the various artefacts and the diversity of image backgrounds pose great challenges for mesoscale whole-brain projection mapping. This is also the main reason that previous machine-learning methods (e.g. TrailMap) heavily rely on manually annotated training data, which is extremely time-consuming.

To evaluate the efficacy of the automated axon segmentation module in D-LMBmap in capturing axon features throughout the whole brain, we applied it to brain samples generated in different labs by using various viral labelling strategies, including two experimental batches from Stanford University (the U.S.) and three experimental batches from NIBS (China). Long-range projecting axons labelled in these samples are from various brain regions and neuronal types. In addition to serotonergic neurons (midbrain), glutamatergic neurons (cerebellar nuclei) and GABAergic neurons (VTA), we also tested samples containing dopaminergic neurons (VTA) during revision. These axons contain various branching patterns and project to hundreds of brain regions distributed across the whole brain¹¹. Details are also updated in Extended Data Table 3-1.

As presented in the newly added Extended Data Fig.3, D-LMBmap achieves effective axon segmentation in all of the four axonal types across, with a CiDice score of more than 0.85 for all the samples tested, and outperforms Trailmap. Detailed and zoomed-in examples of whole-brain axon prediction results are presented in the updated Extended Data Fig.7 and Extended Data Movie 1, 2, 3.

We also evaluated and compared the efficacy of D-LMBmap in axon segmentation across different brain regions with Trailmap. As presented in the newly added Extended Data Fig. 4, unlike Trailmap, the performance of D-

18

NMETH-A50696A

Response to Reviewers

LMBmap in axon segmentation is consistent and robust across all of the five major brain regions tested.

Furthermore, to ensure that our pipeline is compatible for both sparsely and densely labelled axons, we included brain samples containing a wide spectrum of axon density. Brains from the Sert-Stanford group contains axons with low to medium density, whilst DCN-Stanford, GABA-NIBS and DA-NIBS contain axons with low to high density, and Sert-NIBS contains axons with medium to high density (Extended Data Fig.7, and Fig. 2F). To further validate that our pipeline is well suited for axons across different density and complexity, we tested the axon segmentation performance on selected cubes with different axon densities ranging from 1% to 100%. Here 100% density is defined as the axons occupying around 800 millions of voxels in one cube (200x200x450 voxels). As shown in newly added Extended Data Fig. 5, the performance of D-LMBmap in axon segmentation is consistent and robust across all of the cubes with different axon density.

c) It is recommended that the author supplement the evaluation of the integrity of the segmented axon. The Dice coefficient in Fig2 was only about 0.7, indicating that the degree of signal extraction was limited. Although cDice reached 0.9, as an index to measure topological similarity, it is not appropriate to use cDice for evaluation on the premise that the integrity of extracted axons cannot be guaranteed. In addition, according to "the quantitative evaluation of six sample cubes" [Line 200-202], the precision rate was 0.8 and the Dice coefficient was 0.7, then the recall rate can be calculated as only about 0.6. Does it mean that about 40% of the axons were not recognized?

Thank you for this suggestion. Indeed, CIDice is used to measure the topological similarity but not for integrity evaluation. According to the recent vasculature segmentation works^{12,13}, we believe that measurements such as CIPrecision, CIPrecision and CIDice that are based on the centerline can better evaluate the segmentation of tubular structure comparing to Recall, Precision and Dice.

In the revised manuscript, we use CIPrecision for integrity evaluation and we presented CIDice, CIPrecision, CIPrecision and Dice for total evaluation of axon segmentation. Please see the schematic definition of CIPrecision, CIPrecision and CIDice in the updated Extended Data Fig. 3B. Please also find the detailed data of using CIDice, CIPrecision, CIPrecision and Dice to assess the performance of TrailMap and D-LMBmap on each experimental group in Extended Data Fig.3C.

Comparing to TrailMap, D-LMBmap shows superior performance across every comparison except the CIPrecision in the Sert-Stanford dataset. There are two reasons contribute to this:

- 1). The Sert-Stanford dataset is one of the original datasets used as manually annotated training dataset for TrailMap (The first author of TrailMap, Drew Friedmann, is also one of the co-first authors in Ren et al⁹).
- 2). TrailMap tends to predict axons with redundancy in the Z axis, which is caused by repetitive annotation in adjacent 2D slices (Fig. 2E, Extended Data Movie 2). Therefore, TrailMap tends to predict more positive voxels (both true

19

and false positives), which can result in high CIREcall and low CIPrecision scores (Extended Data Fig.3B). In contrast, our method can reflect the actual distribution and thickness of axons (Extended Data Fig. 2, and Extended Data Movie 1). In fact, if we were to consider an extreme scenario, where an entire cube is predicted by a given method to be covered by axons, the CIREcall generated by this method would reach a maximum of 1. Because of this, considering CIDice can comprehensively reflect the axon segmentation performance judging by both CIPrecision and CIREcall, we stopped using CIREcall for ablation studies but focus on CIDice in the rest of the rebuttal letter (all of the scores are still reported in the manuscript for relevant parts).

d) The main difficulty in axon segmentation or identification is the low signal intensity and the signal-to-noise ratio of the target object, which may further lead to specific problems such as discontinuity of the segmented axon. I suggest adding enlarged views to show the segmented axon details. Extended Data Movie 2 appears to show some unrecognized axons.

Thank you very much for the suggestion. We followed the reviewer's advice and added enlarged views to show the segmented axon details in Extended Data Fig. 2, Extended Data Fig. 7 and Extended Data Movie 1. There are still a few broken axons captured, but, unless all the axons are labelled well and segmented by the software, the discontinuity of the segmented axon cannot be entirely eradicated. Compared to Trailmap, D-LMBmap has drastically reduced this issue. For example, as shown in Extended Data Fig. 2C, all the axons segmented by Trailmap in this cube are broken, and D-LMBmap rescued most of them, but not all.

e) The Dice index and Precision index of this paper are far better than the TrailMap method. TrailMap used expert annotated data to conduct model training and prediction based on, while D-LMBMap used automatically generated training data. I suggest that the authors supplement the corresponding ablation experiments to reveal the mechanism behind the improvement.

Thank you very much for the reviewer's appreciation of the efficiency of our pipeline. Following the reviewer's advice, we did ablation experiments to demonstrate the mechanisms behind the improvements, including data augmentation, axial attention, and the deep neural networks used in training modules.

As presented in the newly added Extended Data Fig. 6, even without data augmentation and axial attention, D-LMBmap is more effective than Trailmap. Nevertheless, our data augmentation and axial attention are essential for the superior performance of our automated annotation and axon segmentation solution and can further remarkably improve the results and achieved a CIDice score around 0.9. Moreover, we also compared the effectiveness of nnU-Net backbone with other most widely used 3D segmentation backbone networks. As shown in Figure R2, based on the same training samples after our data augmentation strategy, nnU-Net performs the best comparing to other network architectures.

Figure R2. Quantitative evaluation for the incorporation of different network architectures for axon prediction. Two manually annotated large-sized cubes (600×600×225 voxels) from Sert-Stanford are used for evaluation (included in Extended Data Table. 3-1)

f) In extended data table 1, the author gives the data set used for axon segmentation. There are 3 data sets with low resolution, it is difficult to ensure that a single axon can be distinguished -- does it limit the application scenarios of the method? Adding a discussion about it will be good.

We thank the reviewer for their suggestion and we incorporated the discussion about the application scenarios [Line 495-511]. Indeed, compared with the resolution of images generated by fMOST (20,000×30,000×25,000), the resolution of images generated by light-sheet microscopy is relatively low (Extended Data Table. 1 and 3). Lower resolution reduces the data size significantly and in this study our data is around 20 – 200 GB per brain sample. However, it is enough for us to achieve all the tasks needed for mesoscale projectome mapping at the whole-brain level, and the processing time using our D-LMBmap pipeline is only 12 hours per brain. Whilst we did not aim for single axon tracing, the pipeline can achieve segmentation, skeleton, and connection relationship judgement based on bulk tracing.

In contrast, fMOST brain data is generally imaged at 40X magnification and has the size around 5TB to 10TB per brain. The D-LMBmap pipeline is compatible with higher resolution images, but because of the limited computational power it may take 500 days to finish projection mapping for a whole brain. If the axonal projections are limited to a few brain regions due to sparse labelling and whole-brain analysis is not necessary (like most fMOST cases), the processing time can be reduced.

g) During the automatic generation of training samples, about 40 axon cubes were selected for some data and about 90 for others. In addition, the number of artifact blocks is also not fixed. How to determine these numbers? Is it a random selection within the whole brain? If not, on what basis?

We have put more details and clarify cube selection in the “Methods” in revised manuscript. [Methods Line 74-80]

The manual selection of “axon” and “artefact” cubes is not random. The reason why the number of cubes manually selected for each tested group is different, is because these groups have different axon projection patterns. To achieve high-quality training, representative “axon” cubes are selected for brain regions receiving axonal innervation. Only 40 “axon” cubes were selected for Sert-Stanford and DCN-Stanford because only subpopulations of serotonin or DCN neurons are labelled and there are many brain regions that do not receive axon inputs. Around 90 “axon” cubes were selected for the Sert-NIBS and GABA-NIBS groups, because they have more brain regions receiving axon inputs and more cubes are selected to represent these brain regions. The number of artefact cubes also depends on the complexity of artefacts for each group.

To describe the relationship between the number of selected training cubes and how well the deep model can predict axons, we did quantitative evaluation by using DCN-Stanford and the newly added brain of DA-NIBS as examples. As shown in Figure R3, we analysed three groups where a differing number of randomly selected training cubes were used. In both of the first groups we used 25 axon and 25 artefact cubes, the second ones 45 axon and 45 artefact cubes, and the last with either 84 axon and 100 artefact cubes or 85 axon and 84 artefact cubes. We find that even though it appears that more training samples can generally achieve better performance for axon segmentation, around 45 axon and 45 artefact cubes can already get satisfactory results for the training of axon segmentation model.

Figure R3. Quantitative evaluation of D-LMBmap for axon segmentation when using different number of training cubes. CIDice and Dice scores were used for evaluation. (a) Axon segmentation performance of D-LMBmap with different number of training cubes selected from DA-NIBS brain. The number of training cubes (axon cube number / artefact cube number) varied from 25/25 to 84/100, and the evaluation was performed using 2 manually annotated large-sized cubes (see Extended Data Table 3-1 for details); (b) Axon segmentation performance of D-LMBmap with different number of training cubes selected from DCN-Stanford brains. The number of training cubes (axon cube number / artefact cube number) varied from 25/25 to 85/84, and the evaluation was performed using 15 manually annotated small-sized cubes that covered different major brain regions (see Extended Data Table 3-2 for details).

3. Brain region segmentation and registration

NMETH-A50696A

Response to Reviewers

a) The idea of style transfer is very clever, and it is indeed promising to solve the problem due to the big style difference between the data to be registered and the reference map like CCF, which affects the registration quality. As stated in the paper, the premise of the smooth application is that "in which the image appearance is converted into a reference image style without losing its original content" [Line 226-227]. However, it can be seen from Extended data fig.4 that the structure information and gray value information of the transferred brain and the original brain is very different. The gray value difference in the original brain has specific biological significance, while some gray values were directly reversed after the transfer. Is there a biological or mathematical explanation behind this mapping mechanism, and how to verify it without losing the original content?

We thank the reviewer very much for their appreciation of our implementation of style transfer to solve the image difference across different modality, especially the difference between samples and atlas.

After style transfer, as shown in Extended Data Fig. 9A, indeed that *"the gray value difference in the original brain has specific biological significance, while some gray values were directly reversed after the transfer."* This is because the images generated by light-sheet microscopy are transferred into Allen atlas style, and Allen atlas is constructed by interpolating high resolution in-plane serial two-photon tomography images¹⁴. This is cross-modality style transfer achieved by our style transfer module based on CycleGAN, which can preserve the original brain structures and region outlines.

CycleGAN is originally designed for keeping image content consistency between the original and the style transferred images, judged by the Cycle-Loss¹⁵. As shown in Fig. 3A, our module first transferred the original LSFM brain images into "Synthetic Allen" images based on the Generator A. Then "Synthetic Allen" images are transferred to "Reconstructed LSFM" images based on the Generator B. Subsequently, the Cycle-Loss is computed between the "Reconstructed LSFM" and the original LSFM images.

The brain structure consistency can also be validated by the brain region segmentation on the style transferred brains (Fig.3E). We generated "ground-truth" data by manually annotating major brain regions on the original samples. Validation results showed that our brain region segmentation on style transferred brain can achieve high Dice scores, indicating that brain structures have not been changed.

Although the style transfer method employed in D-LMBmap is based on CycleGAN, our improved style transfer method outperforms all other method in major brain regions and at the whole brain level. In the newly added Extended Data Fig. 8A and Extended Data Fig. 14A, we showed that for whole brain data, the evaluation metrics of SSIM, PSNR, and FID, the style transfer method employed in D-LMBmap introduces the least information loss between the original LSFM brain and the reconstructed LSFM brain, when compared with the other three methods.

b) Before registration, the author down-sampled the whole brain data set to 320 x 456 x 528 pixels (extended data table 4), which belonged to the test data with

23

NMETH-A50696A

Response to Reviewers

low resolution. Allen CCF currently provides the data with 10-micron resolution. I wonder how well the method has been tested on this level of data.

Thank you for the interesting question. Indeed, as the reviewer pointed out, the latest CCF published by Allen is at 10-micron resolution. The brain size of this latest CCF is 800×1140×1320. We used Allen CCF which is at 25-micron resolution and the brain size is 320×456×528, and that is why we down-sampled the raw data to 320×456×528 for whole brain registration.

Whole-brain 3D registration is a computationally expensive process, especially for the learning-based 3D registration. The following table summarizes the size of the brain and the correspondent theoretical GPU memory required:

Brain size	GPU memory required theoretically
(80,114,132)	6.5G
(160,228,264)	8.9G
(320,456,528)	28.5G
(800,1140,1320)	360.6G

Currently, our learning-based 3D registration is trained on NVIDIA GeForce RTX 3090 GPU with 24GB of memory. Our Concern is that such a highly advanced GPU sufficient to support whole brain resolution using 10-micron resolution CCF may not be accessible to many researchers.

c) As far as we know, it works well to calculate Dice and so on for large brain regions. Can you supplement registration results of smaller nuclei (lateral habenular, etc.), and even consider using reference sites to calculate distance deviations directly?

Thank you for the suggestion. As Reviewer 1 also asked the same question on the registration of small brain structures, we have provided detailed answers on the Page 2 and 3 of this letter. Please find the summary blow.

We generated the ground truth of small region registration by manually annotating several small brain structures in the thalamus, hypothalamus, and brainstem to facilitate the validation, including habenular (Hb), mammillothalamic tract (mtt), anterior commissure temporal limb (act), fasciculus retroflexus (fr) and interpeduncular nucleus (IPN). We quantified the whole-brain registration results of these five small brain regions using the automated pipeline in D-LMBmap, and compared them with related registration solutions. D-LMBmap can outperform other methods in all five small brain regions. As presented in the new Fig. 4D, D-LMBmap achieves a median Dice score from 0.60 to 0.85 for individual small brain regions, and an average median Dice score of 0.76. Compared with other methods, D-LMBmap achieves about 35% higher Dice score in small brain region registration. Next, we asked, when doing small structure registration, how close D-LMBmap

24

automated pipeline was to achieving results comparable to its pipeline that incorporated training data from manual segmentation (Extended Data Fig. 16B). As presented in the new Extended Data Fig. 16C, trained with data generated by user delineation, D-LMBmap achieves a median Dice score from 0.80 to 0.94 for individual small brain regions, and an average median Dice score of 0.86. Thus, our newly added data demonstrate that D-LMBmap is the best method for accurately and efficiently registering small brain structures with detectable features under the autofluorescence channel. D-LMBmap also provides a new strategy and the pipeline for optimising the accuracy of small brain structures without detectable features under the autofluorescence. Besides our automated pipeline, users could also either add a scant amount of manually segmented constraints or constraints from channels with cytoarchitectonic features during training to generate whole-brain registration of small structures more precisely.

Additionally, following the reviewer's advice, we included landmark distance as an additional evaluation metric, along with the Dice score. To compute landmarks in both the Allen atlas and LSFM brains, we employed the landmark extraction method – 2.5D corner detection, presented in mBrainAligner². We filtered out 18 landmarks that were automatically detected across all the testing LSFM brains and the Allen atlas (Extended Data Fig. 17D). As shown in the newly added Extended Data Fig. 17E, D-LMBmap achieves better landmark alignment than other methods.

4. Software

a) The software is difficult to install and lacks dependency packages. Could you please provide more detailed instructions for the convenience of users?

We recognise the difficulties encountered and should have provided more detailed instructions. To remedy this, we have invited 12 researchers to test the updated software for ease of use. Of this group, 8 were running the software on Windows 10 X64 and the remaining 4 were using Windows 11 X64. Neither system tested had prior installation of dependency packages. As a result, we have implemented several improvements to expediate the installation process. For example, the software can now be installed directly by double clicking the "D-LMBmap.exe" file.

In addition, we have also provided more detailed instructions consisting of documents, a tutorial movie and example data, accessible from our project's Github page (Link: <https://github.com/lmbneuron/D-LMBmap>). Users can follow the movie as a guide to achieve each module in whole brain projection mapping. Furthermore, on our Github page we have provided our code with detailed annotations consisting of three files: 'axon segmentation', 'brain region segmentation' and 'whole brain registration'. By running this code, users can train their own data and accomplish whole-brain projection mapping.

b) The software is not stable enough and sometimes encounters crashes. It is suggested to further optimize the software for better promotion and use.

We acknowledge the problems the reviewer encountered while running the software. We have tested the software on a greater number of users and

NMETH-A50696A

Response to Reviewers

incorporated their feedback. The revised version has been optimised and updated to provide a more stable performance and a more user-friendly experience. In the future, as well as frequently supplying updates to maintain the software, we will encourage users to report any issues or bugs encountered. Ultimately, we are confident that this software will be an invaluable resource to those in the field and greatly accelerate studies in mesoscale whole brain mapping.

Reviewer #3:

Remarks to the Author:

A. The authors present an integrated pipeline for automated analysis of mesoscale projection mapping from whole-brain light microscopic imaging data of different modalities. The work is based on putting together deep net based modules for axon segmentation and brain/atlas registration.

B. The primary claim to originality is that this is the first time such an integrated pipeline has been presented, and also that individual modules in this pipeline are new. This claim is difficult to sustain.

We respectfully disagree with the characterizations of our study commented here. Please see below for more detailed responses.

First, the primary focus of the paper is on the brain/atlas registration problem, but the methods presented by the authors are neither of very high quality (they only segment the brain into 6 compartments - compare this with the 500-1000 compartments in current mouse brain atlases). Leaving aside the comparisons with other methods, such a segmentation is of very little value for modern mesoscale projection analysis from a biological significance perspective.

D-LMBmap contains three modules: axon segmentation, brain region segmentation and whole brain registration (Fig. 1). We developed new methods for each of the modules and presented superior data analysis results compared with other state-of-art methods (Fig. 2, 3, 4, &5). High-quality mapping of mesoscale whole brain connectivity requires all of the three modules to be efficient and powerful, and D-LMBmap not only is the first pipeline integrating all the three modules as end-to-end software but a tool that significantly improved the performance on all the three aspects.

It appears from the above comments that Reviewer #3 may have confused "brain region segmentation" with "whole brain registration". In our pipeline, the ~1000 compartments in the mouse brain atlases are mapped and registered after "whole brain registration", based on the crucial results and constraints generated by the "brain region segmentation" step (Fig. 4 & 5). The purpose of image segmentation is to partition an image to different regions based on given criteria for future processes. Brain region segmentation is a major application of atlas-based registration^{16,17}, and whole brain registration task can also benefit from brain region segmentation².

It is not necessary nor computationally effective to have all the 1000 compartments segmented by learning-based methods, because the final goal of mesoscale whole brain connectivity mapping is to achieve brain-to-atlas or/and atlas-to-brain registration of the tracing/labelling data.

26

We developed a novel semi-supervised multi-view pipeline for accurate segmentation of 6 major brain regions. In the brain region segmentation field, three state-of-art methods^{2,16,17} have achieved great results and they also focused on the segmentation of major brain regions but not each of the individual small compartments listed on the atlas. Comparing with these methods, D-LMBmap showed superior performance (Fig. 3E, F, Extended Data Fig. 12, Extended Data Fig. 13). Most importantly, our highly effective brain region segmentation process provides criteria and constraints to greatly improve the performance of whole-brain 3D registration (Fig. 4C, D, E, Extended Data Fig. 15, Extended Data Fig. 16, Extended Data Fig. 17).

It is possible that the reviewer's primary concern is about the accuracy of the registration of small brain regions, which is also the major concern raised by the other two reviewers. We provided detailed answers on the Page 2 and 3 of this letter. Please find the summary blow.

We generated ground truth of small region registration by manually annotating several small brain structures in the thalamus, hypothalamus, and brainstem to facilitate the validation, including habenular (Hb), mammillothalamic tract (mtt), anterior commissure temporal limb (act), fasciculus retroflexus (fr) and interpeduncular nucleus (IPN). We quantified the whole-brain registration results of these five small brain regions using the automated pipeline in D-LMBmap, and compared them with related registration solutions. D-LMBmap can outperform other methods in all five small brain regions. As presented in the new Figure 4D, D-LMBmap achieves a median Dice score from 0.60 to 0.85 for individual small brain regions, and an average median Dice score of 0.76. Compared with other methods, D-LMBmap achieves about 35% higher Dice score in small brain region registration. Next, we asked, when doing small structure registration, how close D-LMBmap automated pipeline was to achieving results comparable to its pipeline that incorporated training data from manual segmentation (Extended Data Fig. 16B). As presented in the new Extended Data Fig. 16C, trained with data generated by user delineation, D-LMBmap achieves a median Dice score from 0.80 to 0.94 for individual small brain regions, and an average median Dice score of 0.86. Thus, our newly added data demonstrate that D-LMBmap is the best method for accurately and efficiently registering small brain structures with detectable features under the autofluorescence channel. D-LMBmap also provides a new strategy and the pipeline for optimising the accuracy of small brain structures without detectable features under the autofluorescence. Besides our automated pipeline, users could also either add a scant amount of manually segmented constraints or constraints from channels with cytoarchitectonic features during training to generate whole-brain registration of small structures more precisely.

Additionally, we included landmark distance as an additional evaluation metric, along with the Dice score. To compute landmarks in both the Allen atlas and LSFM brains, we employed the landmark extraction method – 2.5D corner detection, presented in mBrainAligner². We filtered out 18 landmarks that were automatically detected across all the testing LSFM brains and the Allen atlas (Extended Data Fig. 17D). As shown in the newly added Extended Data Fig. 17E, D-LMBmap achieves better landmark alignment than other methods.

NMETH-A50696A

Response to Reviewers

Besides, the multimodal/cross contrast brain/atlas registration problem addressed by the authors through the style-transfer and GANS approaches has already been addressed satisfactorily in previous publications (e.g. both learned contrast maps and treatment of damaged/distorted sections of the brain are addressed in https://doi.org/10.1007/978-3-030-33226-6_18).

D-LMBmap is the first work applied style transfer method to the learning-based multimodal brain style alignment, and this application greatly improved the atlas based registration. As *Reviewer 2* pointed out, "*The idea of style transfer is very clever, and it is indeed promising to solve the problem due to the big style difference between the data to be registered and the reference map like CCF, which affects the registration quality.*"

We thank the reviewer for suggesting the reference of Tward Daniel et al.¹⁸. However, they did not make use of GANs (Generative Adversarial Nets) but rather the traditional Gaussian Mixture Modeling (GMM). Furthermore, they did not perform 3D whole-brain registration but rather 3D CCF to 2D brain slice registration.

In the work of Tward Daniel et al.¹⁹, they transformed the intensity based on the traditional cubic polynomial intensity transformation (solving Cubic Polynomial, i.e., RGB three channels), which is completely distinct from our learning-based style transfer solution. For their Generative Model, it is based on the traditional GMM, which is used to predict the shape of 2D slices. Such GMM method is not GANs approaches.

Beyond this, they did not register damaged/distorted sections but rather employed the GMM methods to estimate the abnormality locations in 2D slices. D-LMBmap, however, can demonstrably achieve high-quality registration of damage/distorted 3D brain samples (Extended Data Fig. 18).

Second, segmentation of axons in brain images is a large subject and goes well beyond the few references considered by the authors (consider the DIADEM challenge and many associated papers). Even in the context of mesoscale projection mapping there have been multiple relevant works, including the original AIBS publications on mesoscale connectivity maps of the mouse brain.

We agree that segmentation of axons in brain images is a large subject and there are multiple works focused on mesoscale projection mapping.

There are three primary subjects relevant to axon segmentation of light microscopy, 1) digital reconstruction of neuronal morphology in 2D slices, 2) single neuron tracing in high-resolution images generated by block-face imaging (e.g. fMOST), and 3) axon segmentation of 3D whole-brain images generated by the integration of tissue-clearing methods and light-sheet fluorescence microscopy (LSFM).

The axon segmentation method in D-LMBmap is developed for the third subject - high-throughput mesoscale 3D whole-brain connectivity mapping achieved by LSFM and tissue-clearing. Considering the importance and significance of whole brain mesoscale connectivity mapping, there is a

NMETH-A50696A

Response to Reviewers

surprisingly limited number of studies focused on axon segmentation analysis of whole brain LSM data^{19,20}.

Studies/activities mentioned by the reviewers have a different subject comparing with our axon segmentation method.

The DIADEM challenge was a neuron tracing competition held in 2010, which focused on digital reconstruction of neuronal morphology in 2D slices (<https://link.springer.com/article/10.1007/s12021-010-9095-5>). These images are a series of 2D slices taken by 40X to 100X objective lens and each image contains sparsely labelled neurons that often are not long-range (whole-brain level) projecting neurons.

In the series of AIBS (Allen Institute for Brain Science) studies on mesoscale connectivity maps of the mouse brain, there are two types of analysis applied to the whole brain level projection analysis. For the datasets they generated by serial two-photon tomography (STPT)^{21,22}, AIBS did not perform learning-based axon segmentation but projection density analysis based on a combination of adaptive edge/line detection and morphological processing. They undertook single neuron reconstruction with the data generated by fMOST by manual annotation through Vaa3D, without learning-based axon segmentation²³.

We acknowledged the achievements in the axon segmentation methods applied in block-face imaging and incorporate discussions in our revised manuscript. [Line 495-511]

The authors posit that the primary difficulty lies in the laborious process of manually segmenting of proofreading individual axons under a sufficient variety of circumstances. This is true, but the solution offered by the authors (reliance on human labeling of image cubes containing/not containing axons, followed by a set of classical machine vision approaches including thresholding, morphological operations etc) fails to reach the biologically relevant bar of providing ground truth guarantees on the individually traced neurite fragments. While the specific technique presented by the authors for axon segmentation may or may not correspond to previous approaches (it is difficult to judge because the authors do not comprehensively review this large literature), there is no demonstration of biological ground truthing (which necessarily requires a validation set of manually labelled axons across a spectrum of brain regions). Thus it is difficult to conclude that there has been a true methodological advance here, as opposed to yet another technique added to a large variety of techniques already brought to bear on this problem.

We are happy to know that the reviewer also thinks that “*the laborious process of manually segmenting of proofreading individual axons under a sufficient variety of circumstances*” is a major difficulty lies in axon segmentation. Especially relevant to this is the series of work carried out by AIBS on mesoscale connectivity mapping. The reason for this is that they did not use a learning-based axon segmentation method but relied heavily on manual annotation and proofreading.

We must clarify that we are handling distinct datasets and employing dissimilar methodologies. Currently, the only learning-based method designed for high-throughput analysis of whole brain projectomes captured from LFSM is TrailMap²⁰. The primary problem for this strategy is the laborious process of generating training data by human annotation. We hope that since the reviewer appreciates that laborious manual proofreading can be a primary difficulty for single neuron reconstruction, he/she will also acknowledge that generating high quality and quantity training data manually is the bottleneck for learning-based segmentation methods.

To break this bottleneck, we developed an automated pipeline for axon segmentation of LFSM data. The “*human labeling of image cubes containing/not containing axons*” is just the starting point for generating expanded training data. We described our automated pipeline in the main manuscript (lines 129 to 191) and the detailed designs in the methods section (lines 64 to 126). Following the selection of cubes having axons and those containing artefacts, we can generate and expand the training data through subsequent data processing and data augmentation.

To ensure the accuracy and reliability of our axon segmentation results, we follow the standard practice of validating our training outcomes using ground truth generated by manual annotation, as shown in Figure 2. It is important to note that the manual annotation performed in this context is solely for validation purposes and not used for training. This approach provides an objective and rigorous evaluation of the performance of our method and ensures that our segmentation results are consistent with the ground truth. By using manual annotation solely for validation purposes, we can prevent any potential biases from being introduced into the training process, leading to more reliable and unbiased results. Comparing our validation results with the only existing learning-based axon segmentation method for LFSM images – TrailMap²⁰, D-LMBmap showed superior performance (Fig. 2). As Reviewer 2 pointed out, “*The Dice index and Precision index of this paper are far better than the TrailMap method.*”

Third the authors do not show recognition of one of the primary issues that plague the analysis of mesoscale projections, namely the analysis of the injection sites themselves. The key problem with this kind of projection mapping and associated analysis is that injections are difficult to control (thus they have widely varying sizes and seldom cover the brain in a uniform manner), thus the projection maps derived from such injections have necessarily to be subjected to careful deconvolution analysis. This is still an open problem without satisfactory solutions, perhaps only to be addressed by combining with single-neuron data (of which there is a growing volume today). Genetic constructs seldom label a spatially well-localized set of somata, so any automated pipeline for mesoscale projection analysis worth its salt must address thorny issue. The manuscript shows no evidence that this has been considered.

We did not address the analysis of injection sites in our description of D-LMBmap because we believe that this is not a problem that requires solving through the development of computational methods. Instead, the Allen Institute provides tools for analyzing injection sites as part of their mesoscale

NMETH-A50696A

Response to Reviewers

connectivity mapping work, which we discussed earlier (<https://allensdk.readthedocs.io/en/latest/connectivity.html#structure-level-projection-data>). Furthermore, in our previous work, we have demonstrated that modern viral-genetic methods can provide reliable spatially well-localized sets of somata⁸⁻¹⁰.

If the reviewer wishes to integrate and quantify both neurons labelled at the injection site and their long-range projections, an additional channel for somata labelling during viral injection and light-sheet imaging can easily accomplish this. As the current D-LMBmap software already includes a soma detection function (see Fig. 4E), this end-to-end pipeline can perform both soma/nuclei quantification and projection mapping for the same brain. Thus, D-LMBmap can accommodate both somata labelling signals and projection mapping signals, which can be registered to the atlas.

As such, the novelty of both the modules and the integrated approach is modest, and of uncertain biological relevance. If the authors consider that the software pipeline integration is the key story here this paper might better be directed towards a software journal.

Multi-function integration is only one of the great features D-LMBmap processes. It integrates three modules including axon segmentation, brain region segmentation and whole brain registration. We developed new methods for each of the modules and compared them with other state-of-the-art methods. The axon segmentation results generated by D-LMBmap “are far better than the TrailMap method” (Reviewer 2), requiring no manual annotation. Our brain region segmentation and whole brain registration results show “the advantages of D-LMBmap compared to several other current brain mapping tools, such as SeBRe, BIRDS, mBrainAligner” (Reviewer 1). We completely agree with Reviewer 1 that, “because lightsheet and other 3D microscopic imaging technologies have been adopted by numerous laboratories, the D-LMPmap will be an extremely valuable tool for these labs to map anatomical and behavioral data”.

Reference

1. Ni, H., Tan, C., Feng, Z., Chen, S., Zhang, Z., Li, W., Guan, Y., Gong, H., Luo, Q., and Li, A. (2020). A Robust Image Registration Interface for Large Volume Brain Atlas. *Sci Rep* 10, 2139. [10.1038/s41598-020-59042-y](https://doi.org/10.1038/s41598-020-59042-y).
2. Qu, L., Li, Y., Xie, P., Liu, L., Wang, Y., Wu, J., Liu, Y., Wang, T., Li, L., Guo, K., et al. (2022). Cross-modal coherent registration of whole mouse brains. *Nature Methods* 19, 111–118. [10.1038/s41592-021-01334-w](https://doi.org/10.1038/s41592-021-01334-w).
3. Ni, H., Feng, Z., Guan, Y., Jia, X., Chen, W., Jiang, T., Zhong, Q., Yuan, J., Ren, M., Li, X., et al. (2021). DeepMapi: a Fully Automatic Registration Method for Mesoscopic Optical Brain Images Using Convolutional Neural Networks. *Neuroinformatics* 19, 267–284. [10.1007/s12021-020-09483-7](https://doi.org/10.1007/s12021-020-09483-7).
4. Jiang, X., Ma, J., Xiao, G., Shao, Z., and Guo, X. (2021). A review of multimodal image matching: Methods and applications. *Information Fusion* 73, 22–71. [10.1016/j.inffus.2021.02.012](https://doi.org/10.1016/j.inffus.2021.02.012).

31

NMETH-A50696A

Response to Reviewers

5. Zhong, Q., Li, A., Jin, R., Zhang, D., Li, X., Jia, X., Ding, Z., Luo, P., Zhou, C., Jiang, C., et al. (2021). High-definition imaging using line-illumination modulation microscopy. *Nat Methods* 18, 309–315. 10.1038/s41592-021-01074-x.
6. Wang, X., Xiong, H., Liu, Y., Yang, T., Li, A., Huang, F., Yin, F., Su, L., Liu, L., Li, N., et al. (2021). Chemical sectioning fluorescence tomography: high-throughput, high-contrast, multicolor, whole-brain imaging at subcellular resolution. *Cell Reports* 34, 108709. 10.1016/j.celrep.2021.108709.
7. Winnubst, J., Bas, E., Ferreira, T.A., Wu, Z., Economo, M.N., Edson, P., Arthur, B.J., Bruns, C., Rokicki, K., Schauder, D., et al. (2019). Reconstruction of 1,000 Projection Neurons Reveals New Cell Types and Organization of Long-Range Connectivity in the Mouse Brain. *Cell* 179, 268–281.e13. 10.1016/j.cell.2019.07.042.
8. Ren, J., Friedmann, D., Xiong, J., Liu, C.D., Ferguson, B.R., Weerakkody, T., DeLoach, K.E., Ran, C., Pun, A., Sun, Y., et al. (2018). Anatomically Defined and Functionally Distinct Dorsal Raphe Serotonin Sub-systems. *Cell* 175, 472–487.e20. 10.1016/j.cell.2018.07.043.
9. Ren, J., Isakova, A., Friedmann, D., Zeng, J., Grutzner, S.M., Pun, A., Zhao, G.Q., Kolluru, S.S., Wang, R., Lin, R., et al. (2019). Single-cell transcriptomes and whole-brain projections of serotonin neurons in the mouse dorsal and median raphe nuclei. *eLife* 8, 1–36. 10.7554/eLife.49424.
10. Luo, L., Callaway, E.M., and Svoboda, K. (2018). Genetic Dissection of Neural Circuits: A Decade of Progress. *Neuron* 98, 256–281. 10.1016/j.neuron.2018.03.040.
11. Taylor, S.R., Badurek, S., Dileone, R.J., Nashmi, R., Minichiello, L., and Picciotto, M.R. (2014). GABAergic and glutamatergic efferents of the mouse ventral tegmental area. *Journal of Comparative Neurology* 522, 3308–3334. 10.1002/cne.23603.
12. Shit, S., Paetzold, J.C., Sekuboyina, A., Ezhov, I., Unger, A., Zhylyka, A., Plum, J.P.W., Bauer, U., and Menze, B.H. (2021). CLDICE - A novel topology-preserving loss function for tubular structure segmentation. *Proceedings of the IEEE Computer Society Conference on Computer Vision and Pattern Recognition*, 16555–16564. 10.1109/CVPR46437.2021.01629.
13. Todorov, M.I., Paetzold, J.C., Schoppe, O., Tetteh, G., Shit, S., Efremov, V., Todorov-Völgyi, K., Düring, M., Dichgans, M., Piraud, M., et al. (2020). Machine learning analysis of whole mouse brain vasculature. *Nature Methods* 17, 442–449. 10.1038/s41592-020-0792-1.
14. Wang, Q., Ding, S.L., Li, Y., Royall, J., Feng, D., Lesnar, P., Graddis, N., Naeemi, M., Facer, B., Ho, A., et al. (2020). The Allen Mouse Brain Common Coordinate Framework: A 3D Reference Atlas. *Cell* 181, 936–953.e20. 10.1016/j.cell.2020.04.007.
15. Zhu, J.Y., Park, T., Isola, P., and Efros, A.A. (2017). Unpaired Image-to-Image Translation Using Cycle-Consistent Adversarial Networks.

32

NMETH-A50696A

Response to Reviewers

- Proceedings of the IEEE International Conference on Computer Vision 2017-October, 2242–2251. 10.1109/ICCV.2017.244.
16. Wang, X., Zeng, W., Yang, X., Fang, C., Han, Y., and Fei, P. (2021). Bi-channel image registration and deep-learning segmentation (Birds) for efficient, versatile 3d mapping of mouse brain. *eLife* 10, 1–20. 10.7554/eLife.63455.
 17. Iqbal, A., Khan, R., and Karayannis, T. (2019). Developing a brain atlas through deep learning. *Nature Machine Intelligence* 1, 277–287. 10.1038/s42256-019-0058-8.
 18. Tward, D., Li, X., Huo, B., Lee, B., Mitra, P., and Miller, M. (2019). 3D Mapping of Serial Histology Sections with Anomalies Using a Novel Robust Deformable Registration Algorithm. In *Lecture Notes in Computer Science (including subseries Lecture Notes in Artificial Intelligence and Lecture Notes in Bioinformatics) Lecture Notes in Computer Science.*, D. Zhu, J. Yan, H. Huang, L. Shen, P. M. Thompson, C.-F. Westin, X. Pennec, S. Joshi, M. Nielsen, T. Fletcher, et al., eds. (Springer International Publishing), pp. 162–173. 10.1007/978-3-030-33226-6_18.
 19. Tyson, A.L., and Margrie, T.W. (2022). Mesoscale microscopy and image analysis tools for understanding the brain. *Progress in Biophysics and Molecular Biology* 168, 81–93. 10.1016/j.pbiomolbio.2021.06.013.
 20. Friedmann, D., Pun, A., Adams, E.L., Lui, J.H., Kebschull, J.M., Grutzner, S.M., Castagnola, C., Tessier-Lavigne, M., and Luo, L. (2020). Mapping mesoscale axonal projections in the mouse brain using a 3D convolutional network. *Proc. Natl. Acad. Sci. U.S.A.* 117, 11068–11075. 10.1073/pnas.1918465117.
 21. Harris, J.A., Mihalas, S., Hirokawa, K.E., Whitesell, J.D., Choi, H., Bernard, A., Bohn, P., Caldejon, S., Casal, L., Cho, A., et al. (2019). Hierarchical organization of cortical and thalamic connectivity. *Nature* 575, 195–202. 10.1038/s41586-019-1716-z.
 22. Oh, S.W., Harris, J.A., Ng, L., Winslow, B., Cain, N., Mihalas, S., Wang, Q., Lau, C., Kuan, L., Henry, A.M., et al. (2014). A mesoscale connectome of the mouse brain. *Nature* 508, 207–214. 10.1038/nature13186.
 23. Peng, H., Xie, P., Liu, L., Kuang, X., Wang, Y., Qu, L., Gong, H., Jiang, S., Li, A., Ruan, Z., et al. (2021). Morphological diversity of single neurons in molecularly defined cell types. *Nature* 598, 174–181. 10.1038/s41586-021-03941-1.

Decision Letter, first revision:

Dear Jing,

Thank you for submitting your revised manuscript "D-LMBmap: A fully automated deep learning pipeline for whole-brain profiling of neural circuitry" (NMEMH-A50696A). It has now been seen by the original referees and their comments are below. The reviewers find that the paper has improved in revision, and therefore we'll be happy in principle to publish it in Nature Methods, pending minor revisions to satisfy the referees' final requests and to comply with our editorial and formatting guidelines.

In response to the remaining comments from reviewer 2, we ask that you include an honest and detailed discussion of computational costs/data size.

TRANSPARENT PEER REVIEW

Nature Methods offers a transparent peer review option for new original research manuscripts submitted from 17th February 2021. We encourage increased transparency in peer review by publishing the reviewer comments, author rebuttal letters and editorial decision letters if the authors agree. Such peer review material is made available as a supplementary peer review file. Please state in the cover letter 'I wish to participate in transparent peer review' if you want to opt in, or 'I do not wish to participate in transparent peer review' if you don't. Failure to state your preference will result in delays in accepting your manuscript for publication.

ORCID

IMPORTANT: Non-corresponding authors do not have to link their ORCIDs but are encouraged to do so. Please note that it will not be possible to add/modify ORCIDs at proof. Thus, please let your co-authors know that if they wish to have their ORCID added to the paper they must follow the procedure

described in the following link prior to acceptance:

Sincerely,

Rita

Rita Strack, Ph.D.

Senior Editor

Nature Methods

Reviewer #1 (Remarks to the Author):

The authors have carefully and thoroughly addressed my concerns, as well as critiques from other reviewers. I have no more comments.

Reviewer #2 (Remarks to the Author):

The authors responded positively to almost all the comments, and the revised manuscript has improved dramatically. I believe these improvements will be more helpful for readers to understand the main contributions of the study and guide everyone to use the technical ideas and tools suggested by the authors.

It is also because I am happy to see new methods can be applied that I think that the difficulties related to the calculation cannot be ignored. The limitation of computing resources is a constraint to applying your method on 10 μm image data. However, as we know, the down-sampling of CCF to 25 μm per pixel will cause information loss of brain parcellation, especially for subtle brain nuclei. This may also be why CCF needs to provide a 10 μm version. Possibly worse, many biology labs may struggle with the computing resources required for even a basic 25 μm solution. Therefore, I strongly recommend that the authors address this issue more comprehensively, assess the application limitations imposed by this technical bottleneck, or at least provide necessary clarifications in the discussion section or elsewhere where appropriate to mitigate potential concerns.

Author Rebuttal, first revision:

Reviewers' Comments:

Reviewer #1:

Remarks to the Author:

The authors have carefully and thoroughly addressed my concerns, as well as critiques from other reviewers. I have no more comments.

We sincerely thank the reviewer for his/her great appreciation of our study.

Reviewer #2:

Remarks to the Author:

The authors responded positively to almost all the comments, and the revised manuscript has improved dramatically. I believe these improvements will be more helpful for readers to understand the main contributions of the study and guide everyone to use the technical ideas and tools suggested by the authors.

We thank the reviewer for his/her approval of our study, and their helpful suggestions.

1. It is also because I am happy to see new methods can be applied that I think that the difficulties related to the calculation cannot be ignored. The limitation of computing resources is a constraint to applying your method on 10 μm image data. However, as we know, the down-sampling of CCF to 25 μm per pixel will cause information loss of brain parcellation, especially for subtle brain nuclei. This may also be why CCF needs to provide a 10 μm version. Possibly worse, many biology labs may struggle with the computing resources required for even a basic 25 μm solution. Therefore, I strongly recommend that the authors address this issue more comprehensively, assess the application limitations imposed by this technical bottleneck, or at least provide necessary clarifications in the discussion section or elsewhere where appropriate to mitigate potential concerns.

In response to the reviewer's comments on computing resources and imaging resolution in relation to brain registration, we greatly value their input. To address this concern, we have included a new table in the manuscript (Supplementary Table 5), which provides comprehensive information regarding computing resources, training time, and registration time associated with different imaging resolutions. A new figure (Supplementary Fig. 8) is added to illustrate the proposed solutions for whole-brain registration at higher resolution. Furthermore, we have incorporated these discussions into the "Discussion" section of the paper (Line 437-446).

As shown in Supplementary Table x, we provided the required GPU memory and actual running time of D-LMBmap for training the deep registration model from scratch using 100 μm , 50 μm , and 25 μm resolutions, respectively. Most wet labs can achieve training using whole-brain images with a resolution of 25 μm per pixel by an ordinary

server equipped with NVIDIA GeForce RTX 3090 GPU. In D-LMBmap, we also provide well-trained deep models for whole-brain registration, and users can complete registration in just a few minutes using an ordinary desktop or laptop.

Based on theoretical calculations, it is not practical to train whole-brain registration models using images captured at a resolution of 10 μ m per pixel using our current pipeline (Supplementary Table x). Most existing methods primarily focus on resolutions of 25 μ m or even lower, and training whole-brain registration models using high-resolution images remains an open area for further investigation. However, we believe that performing registration with images at a resolution of 10 μ m directly at the whole-brain level may not be the most efficient approach from an algorithmic perspective. As we use the multi-scale and multi-constraint strategy for the registration model, we propose to solve this problem by extending our current pipeline with an extra module that registers major brain regions (e.g., CTX, CNU, IB, MB, HB, CBX, CBN) instead of the entire brain with a resolution of 10 μ m. The initial registration parameters and deformation space can be obtained by the whole-brain registration in lower resolutions (e.g., 25 μ m) first by our current pipeline, and each major brain region can be further registered to Allen CCFv3 atlas at a resolution of 10 μ m and integrated together to archive whole-brain registration at higher resolution (Supplementary Figure x). By doing this, even though the total training time is prolonged while performing further refined high resolution registration for each major brain regions, the total training time for whole-brain registration is still acceptable for most wet labs.

There are also other potential solutions we will explore in the future. One option is to use control points [1] instead of brain images for whole-brain registration. Deformation computation for each voxel in a large-sized 3D whole brain is computationally expensive, requiring significant GPU memory and computational time. To address this, during the training of the deep registration neural networks, the deformation matrix can be computed and updated using a subset of control points/voxels instead of all the 3D voxels in the whole brain. With the deformation matrix of control points, other voxels in the 3D whole brain can be computed using interpolation algorithms [2]. We have conducted preliminary tests on this approach, which significantly reduces computational resource requirements. We need to work on this strategy further to minimize the impact on registration performance. In addition, we can explore the use of distributed systems for training large registration models. With the advancements in technologies like ChatGPT and other large deep models, distributed training techniques have been extensively studied and leveraged through cloud computing platforms such as Google and Amazon. This enables the possibility of training registration models directly on 10 μ m and higher resolution at the whole-brain level.

[1] Modat, Marc, et al. "Fast free-form deformation using graphics processing units." *Computer methods and programs in biomedicine* 98.3 (2010): 278-284.

[2] Aumann, Günter. "A simple algorithm for designing developable Bézier surfaces." *Computer Aided Geometric Design* 20.8-9 (2003): 601-619.

Final Decision Letter:

Dear Jing,

I am pleased to inform you that your Article, "D-LMBmap: A fully automated deep learning pipeline for whole-brain profiling of neural circuitry", has now been accepted for publication in Nature Methods. Your paper is tentatively scheduled for publication in our October print issue, and will be published online prior to that. The received and accepted dates will be Oct 20, 2022 and August 2, 2023. This note is intended to let you know what to expect from us over the next month or so, and to let you know where to address any further questions.

Over the next few weeks, your paper will be copyedited to ensure that it conforms to Nature Methods style. Once your paper is typeset, you will receive an email with a link to choose the appropriate publishing options for your paper and our Author Services team will be in touch regarding any additional information that may be required.

You will receive a link to your electronic proof via email with a request to make any corrections within 48 hours. If, when you receive your proof, you cannot meet this deadline, please inform us at rjsproduction@springernature.com immediately.

Please note that *Nature Methods* is a Transformative Journal (TJ). Authors may publish their research with us through the traditional subscription access route or make their paper immediately open access through payment of an article-processing charge (APC). Authors will not be required to make a final decision about access to their article until it has been accepted. [Find out more about Transformative Journals](https://www.springernature.com/gp/open-research/transformative-journals)

Authors may need to take specific actions to achieve [compliance](https://www.springernature.com/gp/open-research/funding/policy-compliance-faqs) with funder and institutional open access mandates. If your research is supported by a funder that requires immediate open access (e.g. according to [Plan S principles](https://www.springernature.com/gp/open-research/plan-s-compliance))

then you should select the gold OA route, and we will direct you to the compliant route where possible. For authors selecting the subscription publication route, the journal's standard licensing terms will need to be accepted, including [self-archiving policies](https://www.springernature.com/gp/open-research/policies/journal-policies). Those licensing terms will supersede any other terms that the author or any third party may assert apply to any version of the manuscript.

Your paper will now be copyedited to ensure that it conforms to Nature Methods style. Once proofs are generated, they will be sent to you electronically and you will be asked to send a corrected version within 24 hours. It is extremely important that you let us know now whether you will be difficult to contact over the next month. If this is the case, we ask that you send us the contact information (email, phone and fax) of someone who will be able to check the proofs and deal with any last-minute problems.

If, when you receive your proof, you cannot meet the deadline, please inform us at rjsproduction@springernature.com immediately.

Once your manuscript is typeset and you have completed the appropriate grant of rights, you will receive a link to your electronic proof via email with a request to make any corrections within 48 hours. If, when you receive your proof, you cannot meet this deadline, please inform us at rjsproduction@springernature.com immediately.

Once your paper has been scheduled for online publication, the Nature press office will be in touch to confirm the details.

Once your paper has been scheduled for online publication, the Nature press office will be in touch to confirm the details.

Content is published online weekly on Mondays and Thursdays, and the embargo is set at 16:00 London time (GMT)/11:00 am US Eastern time (EST) on the day of publication. If you need to know the exact publication date or when the news embargo will be lifted, please contact our press office after you have submitted your proof corrections. Now is the time to inform your Public Relations or Press Office about your paper, as they might be interested in promoting its publication. This will allow them time to

prepare an accurate and satisfactory press release. Include your manuscript tracking number NMETH-A50696B and the name of the journal, which they will need when they contact our office.

About one week before your paper is published online, we shall be distributing a press release to news organizations worldwide, which may include details of your work. We are happy for your institution or funding agency to prepare its own press release, but it must mention the embargo date and Nature Methods. Our Press Office will contact you closer to the time of publication, but if you or your Press Office have any inquiries in the meantime, please contact press@nature.com.

Nature Portfolio journals [encourage authors to share their step-by-step experimental protocols](https://www.nature.com/nature-research/editorial-policies/reporting-standards#protocols) on a protocol sharing platform of their choice. Nature Portfolio 's Protocol Exchange is a free-to-use and open resource for protocols; protocols deposited in Protocol Exchange are citable and can be linked from the published article. More details can found at www.nature.com/protocolexchange/about.

Best regards,
Rita

Rita Strack, Ph.D.